# Facile induction of immune tolerance by an interleukin-2–TGFβ surrogate agonist

Qinli Sun[1], Alison K. Barrett[2], Masato Ogishi[1], Huiyun Lyu[3,4], Hua Jiang[1], Honghui Liu[1], Yang Zhao[1], Grayson E. Rodriguez[1,5,6], Pingdong Tao[1], Matthias Obenaus[1], Karsten D. Householder[1,5], Qizhi Tang[3,4,7], Tobias V. Lanz[2,8] & K. Christopher Garcia[1,6,9,10] ✉

CD4[+] regulatory T cells (T$_{reg}$ cells) are essential for immune tolerance[1]. Peripherally induced T$_{reg}$ cells (pT$_{reg}$ cells) complement thymic T$_{reg}$ cells by broadening T$_{reg}$ cell reactivity in response to a changing antigenic landscape[2]. Although both TGFβ and IL-2 synergistically promote functional pT$_{reg}$ cell development in vitro[3–6], their combined roles in inducing pT$_{reg}$ cell generation in vivo have not been exploited for tolerizing immunotherapy. Here we designed an IL-2–TGFβ 'surrogate' co-agonist by creating a single-chain fusion protein between IL-2 and a low-affinity TGFβ mimic agonist derived from a helminth parasite[7]. This IL-2–TGFβ surrogate functions as an AND-gated co-agonist and enabled simultaneous cis-activation of IL-2–STAT5 and TGFβ–SMAD2/3 signalling specifically in T cells that express IL-2 receptors. The IL-2–TGFβ surrogate agonist robustly induced antigen-specific, functional and stable pT$_{reg}$ cells in vivo within peripheral lymphoid organs in mice immunized with ovalbumin (OVA) and myelin oligodendrocyte glycoprotein (MOG)$_{35–55}$. The induced pT$_{reg}$ cells display an effector-like, actively expanding state with high RORγt expression, enabling efficient migration and suppression of intestinal inflammation. Treatment with this agonist effectively quelled immune activation in mouse models of allergen-induced allergic inflammation and self-antigen-driven autoimmune neuroinflammation, suggesting a strategy for the induction of antigen-specific pT$_{reg}$ cells in vivo to establish immune tolerance in inflammatory, allergic and autoimmune diseases.

Naive CD4[+] T cells require both STAT and SMAD signalling, induced by JAK/STAT cytokines and TGFβ agonists, respectively, to differentiate into specialized subsets[8,9]. Regulatory T cells (T$_{reg}$ cells) are a specialized CD4[+] T cell subset that maintain immune homeostasis and tolerance[1]. Defined by the transcription factor FOXP3 (refs. 10,11), T$_{reg}$ cells include thymic T$_{reg}$ cells (tT$_{reg}$ cells) that develop in the thymus to enforce self-tolerance, and peripherally induced T$_{reg}$ cells (pT$_{reg}$ cells) that arise from mature CD4[+] T cells after encountering exogenous or altered self antigens[2,12,13]. Cytokine 'third signals' govern pT$_{reg}$ cell differentiation. TGFβ drives FOXP3 induction during pT$_{reg}$ cell differentiation in vitro through SMAD2 and SMAD3 (SMAD2/3) signalling[3]. IL-2 promotes pT$_{reg}$ cell differentiation, functional maturation and expansion through STAT5 activation[4,6,14]. Together, these two pathways form the central axis for pT$_{reg}$ cell generation. Loss of *Itgav* or *Itgb8* in RORγt[+] antigen-presenting cells (APCs) impairs intestinal pT$_{reg}$ cell differentiation[15–19], highlighting the importance of active TGFβ signalling in vivo. However, the synergistic harnessing of STAT5 and SMAD2/3 signalling for pT$_{reg}$ cell generation in vivo remains largely unexplored.

In vivo pT$_{reg}$ cell generation has gained attention because pT$_{reg}$ cells enforce tolerance to diverse non-self antigens, including commensal microbiota, dietary components, allergens, neoantigens and certain pathogens, making their induction therapeutically promising[12,20]. However, this remains challenging because inflammatory cues promote pro-inflammatory CD4[+] T cell differentiation while impairing pT$_{reg}$ cell commitment and stability[21,22]. Moreover, leveraging TGFβ in vivo remains impractical owing to its pleiotropy (fibrosis and tumour promotion) and unfavourable pharmacokinetic properties[23], unlike IL-2, which has been engineered to preferentially target T$_{reg}$ cells[24,25].

To address these challenges, we designed a bi-specific molecule that induces IL-2 and TGFβ signals in cis to the same cell by conjugating a low-affinity TGFβ mimic agonist that largely bypasses other cell types, to IL-2, which acts as both a targeting arm and a STAT5 activator. This molecule selectively and simultaneously activates both pathways on IL-2 receptor (IL-2R)-expressing T cells, enabling localized and cell-specific induction of pT$_{reg}$ cells while minimizing off-target effects. In vivo, it converted up to 80% of ovalbumin (OVA)-specific OT-II and myelin oligodendrocyte glycoprotein (MOG)$_{35–55}$-specific 2D2 CD4[+] T cells into FOXP3[+] pT$_{reg}$ cells in secondary lymphoid organs after intraperitoneal administration of OVA and MOG$_{35–55}$, respectively. The induced pT$_{reg}$ cells are highly functional and stable, exhibit high RORγt expression, establish tolerance that protect against allergic and neuroautoimmune inflammation, and attenuate intestinal inflammation.

[1]Department of Molecular and Cellular Physiology, Stanford University School of Medicine, Stanford, CA, USA. [2]Institute for Immunity, Transplantation, and Infection, Stanford University School of Medicine, Stanford, CA, USA. [3]Diabetes Center, University of California, San Francisco, CA, USA. [4]Gladstone Institute-UCSF of Genomic Immunology, University of California, San Francisco, CA, USA. [5]Program in Immunology, Stanford University School of Medicine, Stanford, CA, USA. [6]Parker Institute for Cancer Immunotherapy, San Francisco, CA, USA. [7]Department of Surgery, University of California, San Francisco, CA, USA. [8]Division of Immunology and Rheumatology, Stanford University School of Medicine, Stanford, CA, USA. [9]Department of Structural Biology, Stanford University School of Medicine, Stanford, CA, USA. [10]Howard Hughes Medical Institute, Stanford University School of Medicine, Stanford, CA, USA. ✉e-mail: kcgarcia@stanford.edu

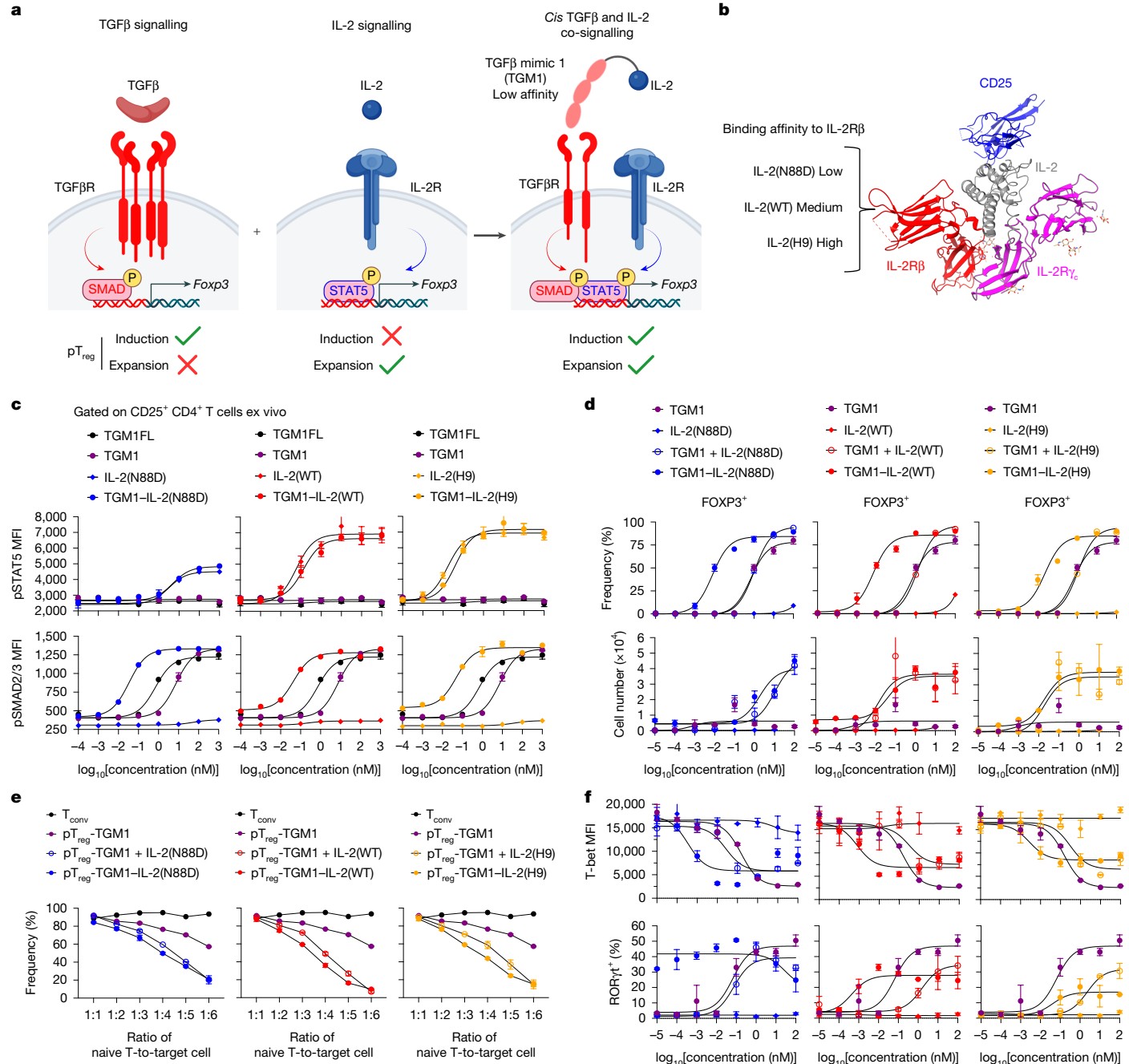

**Fig. 1 | Design and functional validation of IL-2–TGFβ co-agonists. a**, Schematic illustration of the IL-2–TGFβ surrogate agonists. Created in BioRender; Sun, Q. https://BioRender.com/q97sfjt (2026). **b**, Three IL-2 variants with differential IL-2Rβ binding affinities. CD25 is also known as IL-2Rα. **c**, Ex vivo dose–response curves of pSTAT5 and pSMAD2/3 in primary CD25⁺CD4⁺ T cells (*n* = 2 biological replicates). MFI, mean fluorescence intensity. **d**, In vitro dose–response curves

for FOXP3⁺ cell frequency and number in cultured mouse naive CD4⁺ T cells (*n* = 2 biological replicates). **e**, CTV dilution of naive T cells co-cultured in vitro with the indicated cells (*n* = 2 biological replicates). **f**, In vitro dose–response curves for T-bet and RORγt expression in cultured mouse naive CD4⁺ T cells (*n* = 2 biological replicates). Data are presented as mean ± s.e.m. Data in **c**–**f** are representative of two or three independent experiments.

These findings highlight an antigen-specific pT$_{reg}$ cell induction strategy with therapeutic potential across diverse immune diseases.

## Design and characterization of TGM1–IL-2

Our design goal was to activate simultaneous STAT5 and SMAD2/3 signalling in IL-2R-expressing cells. We therefore fused IL-2 to a low-affinity version of TGFβ that is largely functionally inactive on cells that express TGFβ receptor (TGFβR) alone: the fusion protein preferentially acts on IL-2R-expressing T cells, giving the molecule AND-gated activity

to activate both pSTAT5 and SMAD2/3 pathways (Fig. 1a). We were inspired by a helminth-derived TGFβ mimic, termed TGFβ mimic 1 (TGM1), that activates TGFβR–SMAD2/3 signalling[7,26,27]. TGM1 is an immunoglobulin-domain protein[27] that is structurally distinct from mammalian TGFβ (which is a Cys-knot fold[28]), and has ideal drug-like properties. Full-length TGM1 (TGM1FL) comprises five domains: domains 1–3 bind TGFβR with low affinity, whereas domains 4 and 5 engage the co-receptor CD44 with high affinity[26,27]. To generate an IL-2–TGFβ surrogate co-agonist, we fused the low-affinity TGFβ agonist domains 1–3 of TGM1 (referred to hereafter as TGM1) to IL-2 via a

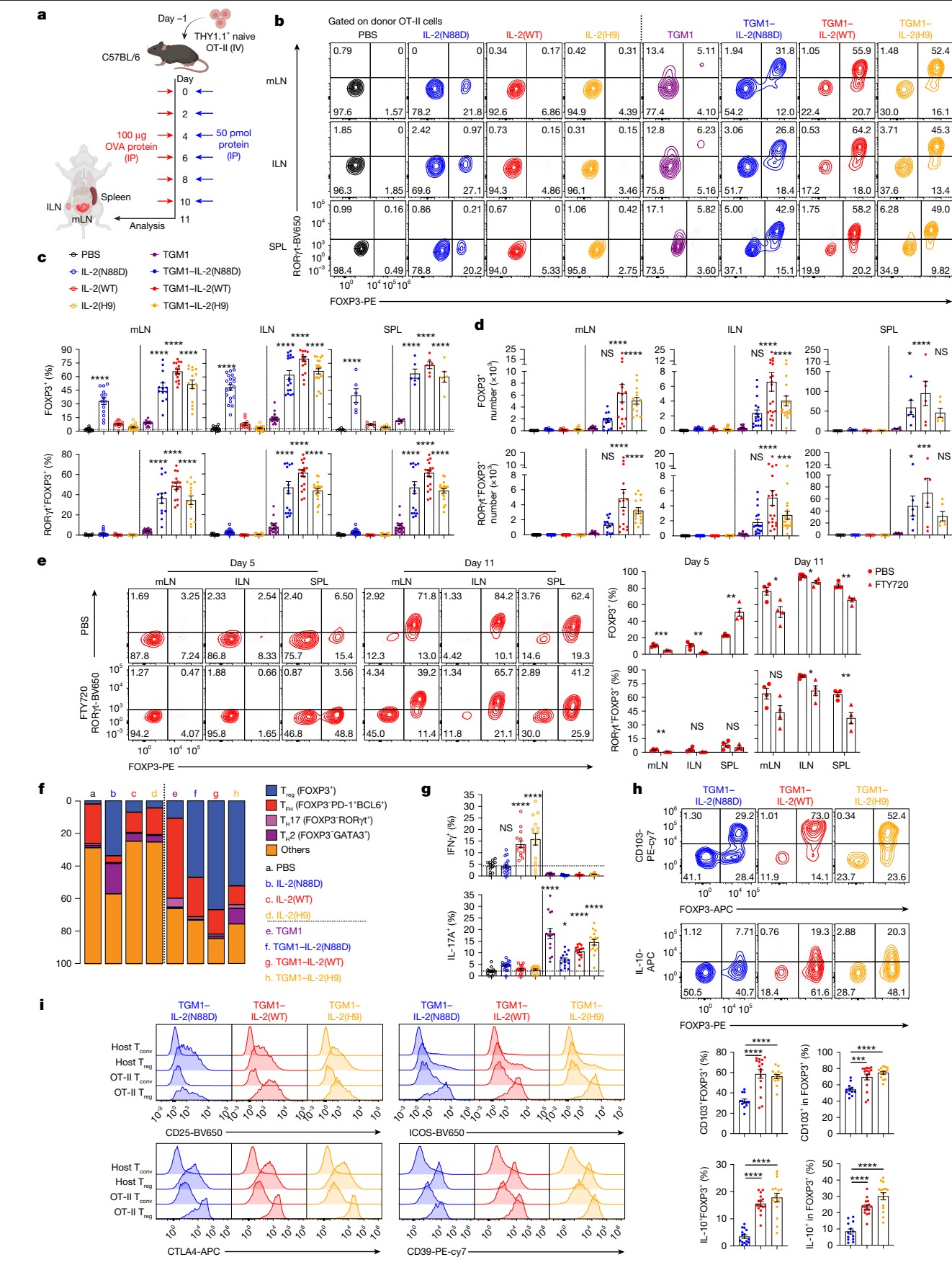

**Fig. 2 | See next page for caption.**

**Fig. 2 | Effect of IL-2–TGFβ co-agonists on antigen-specific pT_reg cell induction and phenotype in OVA-immunized mice. a**, Schematic illustration of OVA and protein administration. Created in BioRender; Sun, Q. https://BioRender.com/xfy9d60 (2026). IP, intraperitoneal; IV, intravenous. **b–d**, Representative flow cytometry plots (**b**) and quantification of FOXP3⁺ and RORγt⁺FOXP3⁺ OT-II cell frequencies (**c**) and absolute cell numbers (**d**) in the indicated lymphoid organs (mLN, $n = 16$ samples; ILN, $n = 20$ samples; spleen (SPL), $n = 6$ samples). **e**, Representative flow cytometry plots and quantification of FOXP3⁺ and RORγt⁺FOXP3⁺ OT-II cell frequencies in the indicated lymphoid organs ($n = 4$ mice per group). **f**, Histogram showing the distribution of distinct subsets (a–h; colour code used in the rest of the figure) among donor OT-II cells in the lymph nodes. **g**, Quantification of IFNγ⁺ and IL-17A⁺ OT-II cell frequencies in the lymph nodes ($n = 16$ samples per group). **h**, Representative flow cytometry plots and quantification of CD103⁺FOXP3⁺ and IL-10⁺FOXP3⁺ OT-II cell frequencies in the lymph nodes ($n = 16$ samples per group). **i**, Representative flow cytometry plots showing expression of the indicated molecules in the indicated CD4⁺ T cell subsets from lymph nodes. Data are presented as mean ± s.e.m. Data in **b–d,f** are pooled from two independent experiments. Data in **e,g–i** are representative of two independent experiments. Statistics were obtained by one-way ANOVA coupled with Dunnett's multiple-comparisons test (**c,d,g,h**) or unpaired Welch's *t*-test (two-tailed) (**e**). *$P < 0.05$, **$P < 0.01$, ***$P < 0.001$, ****$P < 0.0001$; NS, not significant ($P \geq 0.05$).

flexible GSG linker and added mouse serum albumin (MSA) to improve in vivo half-life (Fig. 1a and Extended Data Fig. 1a). We tested three IL-2 variants: wild-type IL-2 (IL-2(WT)), IL-2(N88D) and IL-2(H9) (Fig. 1b). Relative to IL-2(WT), IL-2(N88D) has reduced affinity for the IL-2 receptor β chain (IL-2Rβ), thereby preferentially stimulating T_reg cells with high CD25 expression[24], whereas IL-2(H9) displays enhanced IL-2Rβ affinity and more efficiently activates IL-2Rβ-expressing immune cells[29] (Fig. 1b). Incorporation of low-affinity TGM1 enables the fusion proteins to co-activate SMAD2/3 signalling in IL-2R-expressing cells while providing tunable STAT5 activation potency. These molecules displayed ideal biochemical properties and high expression levels in Expi293 cells, and purified by size-exclusion chromatography as monodisperse proteins (Extended Data Fig. 1b and Supplementary Fig. 1).

Signalling assays in primary mouse CD4⁺ T cells demonstrated that TGM1–IL-2 induced phosphorylated STAT5 (pSTAT5) in CD25⁺ cells at levels resembling those induced by the respective IL-2 counterparts, with the N88D variant exhibiting a reduced maximal response (Fig. 1c). Notably, TGM1–IL-2 markedly enhanced SMAD2/3 activation, shifting the phosphorylated SMAD2/3 (pSMAD2/3) dose–response curve leftward by approximately 2.5 logs compared with low-affinity TGM1 alone (Fig. 1c), reflecting increased apparent TGFβR-binding affinity driven by IL-2–IL-2R engagement. Additionally, pSTAT5 and pSMAD2/3 responses were reduced in CD25⁻ CD4⁺ T cells and CD8⁺ T cells (Extended Data Fig. 1c,d), consistent with their low CD25 expression. Moreover, truncation of any of the three domains in TGM1 abolished pSMAD2/3 activation (Extended Data Fig. 1e). These results show that TGM1–IL-2 preserves IL-2 signalling while robustly inducing pSMAD2/3 in CD25⁺ CD4⁺ T cells.

Next, in vitro pT_reg cell differentiation assays using mouse naive CD4⁺ T cells activated with anti-CD3 and anti-CD28 antibodies showed that TGM1–IL-2 markedly enhanced FOXP3⁺ pT_reg cell generation, shifting the FOXP3⁺ dose–response curve leftward by about 2 logs compared with either low-affinity TGM1 alone or the combination of TGM1 and IL-2 (Fig. 1d). TGM1–IL-2 also produced pT_reg cell numbers resembling the combined TGM1 and IL-2 treatment, although TGM1–IL-2(N88D) generated fewer pT_reg cells than the other two variants (Fig. 1d). Truncation of any TGM1 domain reduced pT_reg cell induction (Extended Data Fig. 1f). Consistently, TGM1–IL-2 more efficiently converted mouse CD44⁺FOXP3⁻ CD4⁺ conventional T cells (T_conv cells) and human peripheral blood mononuclear cell (PBMC) CD4⁺ T cells into pT_reg cells (Extended Data Fig. 1g,h). In vitro suppression assays showed that mouse pT_reg cells generated with TGM1–IL-2 potently suppressed naive T cell proliferation and activation, as evidenced by reduced Cell-Trace Violet (CTV) dilution and decreased CD25 expression (Fig. 1e and Extended Data Fig. 1i). These findings demonstrate that TGM1–IL-2 efficiently drives highly functional pT_reg cell induction in vitro.

TGM1–IL-2 also potently suppressed mouse CD4⁺ T cell activation and effector differentiation in vitro, as shown by left-shifted inhibitory dose–response curves for T-bet, CD25 and CD69 expression and for IFNγ and TNF production compared with either TGM1 alone or the combination of TGM1 and IL-2 (Fig. 1f and Extended Data Fig. 2a,b). TGM1–IL-2 did not induce IL-17A production and, particularly for the wild-type and H9 variants, dampened RORγt induction at high concentrations; it also exerted concentration-dependent effects on CD62L expression (Fig. 1f and Extended Data Fig. 2b,c). Notably, TGM1–IL-2 maintained CD4⁺ T cell numbers resembling the TGM1 plus IL-2 combination (Extended Data Fig. 2d).

## In vivo effects of surrogate agonists

Next, we assessed the in vivo tolerability of TGM1–IL-2 and its effects on IL-2R-expressing cells in mice under steady-state conditions (Extended Data Fig. 3a). A 50 pmol dose of TGM1–IL-2 variants or IL-2(N88D) did not notably affect body weight, whereas TGM1, IL-2(WT) and IL-2(H9) caused mild weight loss (Extended Data Fig. 3b,c). TGM1–IL-2 also did not induce appreciable splenomegaly, in contrast to IL-2, which drove marked splenomegaly with spleen volumes positively correlated with IL-2Rβ-binding affinity (Extended Data Fig. 3d,e). Quantification of splenic immune populations showed that IL-2(N88D) efficiently expanded FOXP3⁺ T_reg cells in vivo, reflected by a marked increase in the ratios of T_reg cells to CD4⁺ T cells, CD8⁺ T cells and natural killer (NK) cells, as well as increased numbers, accompanied by an increased fraction of Ki-67⁺ proliferating T_reg cells among total CD4⁺ T cells and within the T_reg cell compartment, whereas TGM1–IL-2(N88D) did not (Extended Data Fig. 3f–h). By contrast, IL-2(WT), and particularly IL-2(H9), increased splenic CD8⁺ T cell and NK cell numbers, as well as the proportions of Ki-67⁺ proliferating and IFNγ-producing cells, whereas the corresponding TGM1–IL-2 showed markedly attenuated effects (Extended Data Fig. 3i–k). Moreover, all three IL-2 variants robustly promoted T_reg cell activation and function, as indicated by elevated expression of activation markers (CD25, ICOS and GITR) and functional molecules (CTLA4 and IL-10) relative to PBS controls, whereas TGM1–IL-2 did not (Extended Data Fig. 4a,b). Under steady-state conditions, most splenic T_reg cells were Helios⁺RORγt⁻, consistent with a predominance of tT_reg cells, and IL-2 further increased Helios⁺ cell proportions whereas TGM1–IL-2 did not (Extended Data Fig. 4c). Both IL-2 and TGM1–IL-2 had minimal effects on CD62L expression (Extended Data Fig. 4d). These findings demonstrate that TGM1 fusion potently antagonizes IL-2–driven expansion and activation of T_reg cells and other IL-2R⁺ T and NK cells under steady-state conditions (Extended Data Fig. 4e).

## Surrogate agonists induce pT_reg cells in OVA-immunized mice

Next, we evaluated the capacity of TGM1–IL-2 to induce antigen-specific pT_reg cells in an OVA activation model. First, in vitro OT-II pT_reg cell differentiation assays stimulated with OVA peptide showed that TGM1–IL-2, particularly the wild-type variant, strongly promoted FOXP3⁺ pT_reg cell differentiation at low concentrations, increasing both percentage and numbers of cells (Extended Data Fig. 5a). Furthermore, in in vivo pT_reg cell induction assays in which mice received donor naive OT-II cells and repeated intraperitoneal OVA protein (Fig. 2a), analysis on day 11 showed that all three TGM1–IL-2 variants, particularly the IL-2(WT) fusion, drove robust FOXP3⁺ OT-II pT_reg cell induction, reaching up to about 80% in the mesenteric lymph node (mLN), inguinal lymph node

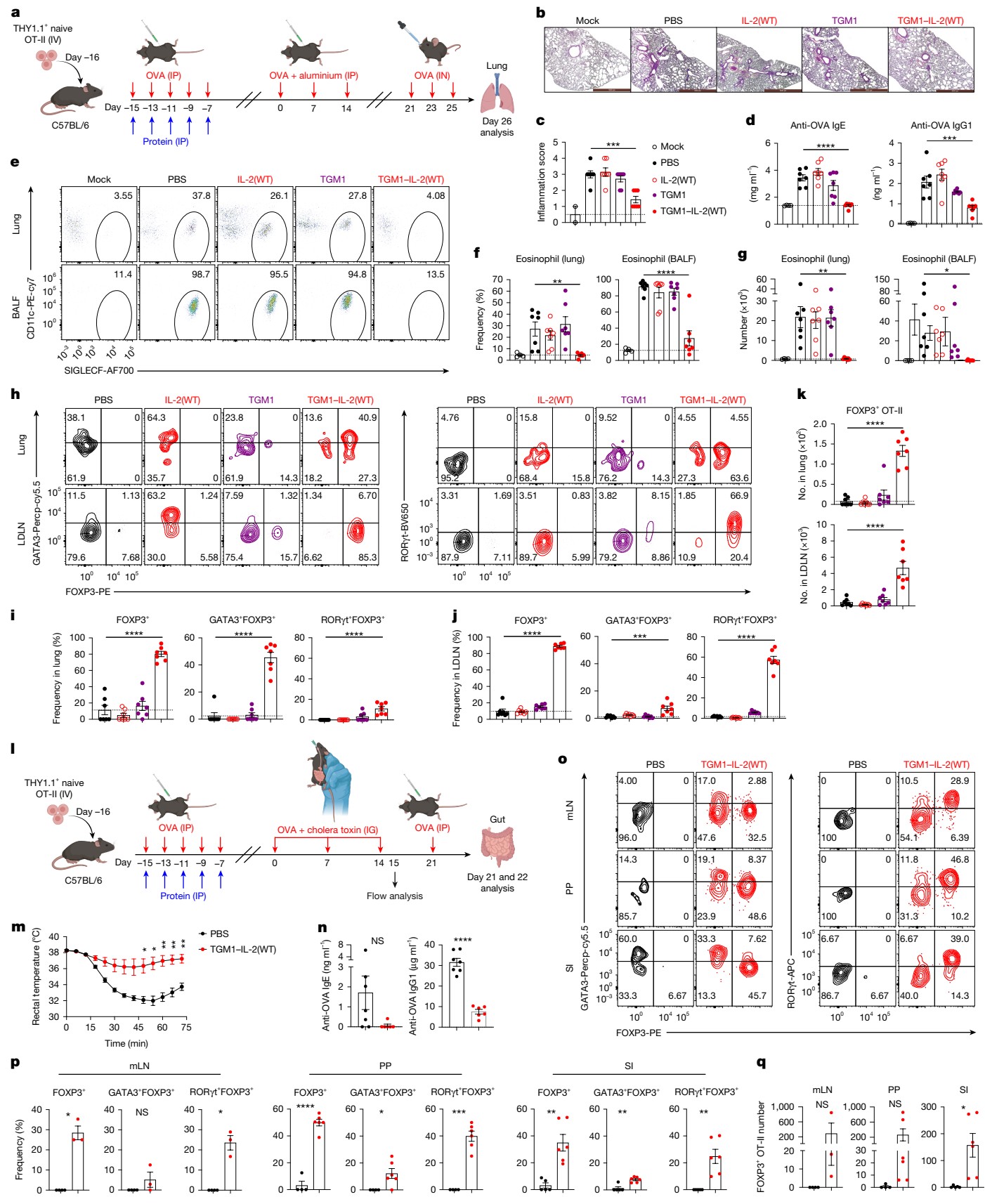

**Fig. 3 | See next page for caption.**

(ILN), and spleen, notably yielding a high proportion of RORγt⁺ pT_reg cells (Fig. 2b,c). By contrast, low-affinity TGM1 induced only about 10% FOXP3⁺ cells, and PBS, IL-2(WT) and IL-2(H9) had minimal effects

(Fig. 2b,c). Although IL-2(N88D) generated a substantial FOXP3⁺ population, these cells were predominantly Helios⁺RORγt⁻ (Fig. 2b,c and Extended Data Fig. 5b), probably reflecting expansion of contaminating

**Fig. 3 | IL-2–TGFβ co-agonists establish immune tolerance to suppress OVA-induced allergic inflammation. a–k**, Therapeutic effect of TGM1–IL-2 in establishing tolerance to suppress OVA-induced airway inflammation (mock, $n = 4$ mice; treatment groups, $n = 7$ mice per group). **a**, Schematic illustration. IN, intranasal. Created in BioRender; Sun, Q. https://BioRender.com/kk38324 (2026). **b,c**, Representative lung histology (**b**) and quantification of inflammation scores (**c**). Scale bars, 1 mm. **d**, Quantification of serum OVA-specific IgE and IgG1 levels. **e–g**, Representative flow cytometry plots (**e**) and quantification of CD11c⁻SIGLECF⁺ eosinophil frequencies (**f**) and absolute numbers (**g**) among CD45.2⁺CD11b⁺ cells. **h–j**, Representative flow cytometry plots (**h**) and quantification of FOXP3⁺, GATA3⁺FOXP3⁺ and RORγt⁺FOXP3⁺ OT-II cell frequencies in lung (**i**) and LDLN (**j**). **k**, Quantification of FOXP3⁺ OT-II cell absolute numbers. **l–n**, Therapeutic effect of TGM1–IL-2 in establishing tolerance to suppress OVA-induced food allergy (PBS, $n = 7$ mice; TGM1–IL-2, $n = 6$ mice). **l**, Schematic illustration. IG, intragastric. Created in BioRender; Sun, Q. https://BioRender.com/5n5ru32 (2026). **m**, Quantification of mean rectal temperature. **n**, Quantification of serum OVA-specific IgE and IgG1 levels. **o–q**, OT-II cell phenotypes (PBS, $n = 5$ mice; TGM1–IL-2, $n = 6$ mice). **o,p**, Representative flow cytometry plots (**o**) and quantification (**p**) of FOXP3⁺, GATA3⁺FOXP3⁺ and RORγt⁺FOXP3⁺ OT-II cell frequencies. PP, Peyer's patches; SI, small intestine. **q**, Quantification of FOXP3⁺ OT-II cell absolute numbers. Data are presented as mean ± s.e.m. Data in **a–q** are representative of two independent experiments. Statistics were obtained by one-way ANOVA coupled with Dunnett's multiple-comparisons test (**c,d,f,g,i–k**), two-way ANOVA coupled with Šídák's multiple-comparisons test (**m**) or unpaired Welch's $t$-test (two-tailed) (**n,p,q**).

OT-II tT$_{reg}$ cells present in the naive donors. Similarly, TGM1–IL-2 substantially increased the FOXP3⁺ T$_{reg}$ cell fraction among endogenous OVA-specific CD4⁺ T cells in lymph nodes (Extended Data Fig. 5c). Moreover, TGM1–IL-2, especially the wild-type and H9 variants, notably increased OT-II cell numbers and Ki-67⁺ cell fractions relative to PBS controls, whereas IL-2 did not, possibly owing to suppression by expanded tT$_{reg}$ cells or nutrient competition with other expanded IL-2R⁺ cells (Extended Data Fig. 5d,e). Consequently, TGM1–IL-2, particularly TGM1–IL-2(WT), yielded substantial numbers of total and RORγt⁺ OT-II pT$_{reg}$ cells (Fig. 2d). These results demonstrate that TGM1–IL-2, especially the wild-type fusion, potently promotes antigen-specific pT$_{reg}$ cell development in peripheral lymphoid organs of OVA-immunized mice (Extended Data Fig. 4e).

We used FTY720 to test whether RORγt⁺ OT-II pT$_{reg}$ cells induced by TGM1–IL-2(WT) can differentiate locally in distinct lymphoid organs; efficacy was confirmed by a marked reduction in blood CD4⁺ T cells (Extended Data Fig. 5f,g). In FTY720-treated mice, substantial FOXP3⁺ OT-II pT$_{reg}$ cells co-expressing CD25 and RORγt were present on day 11 in the mLN, ILN and spleen, with frequencies similar to those in PBS controls (Fig. 2e and Extended Data Fig. 5h,i). By contrast, only a small FOXP3⁺ fraction was detected in the mLN and ILN on day 5, whereas the spleen contained a high FOXP3⁺ frequency with low RORγt expression, indicating that RORγt⁺ pT$_{reg}$ cells were primarily induced between days 5 and 11 (Fig. 2e). FTY720 also markedly reduced both total and pT$_{reg}$ OT-II cells in the spleen, but not in the mLN and ILN, on day 11 (Extended Data Fig. 5j,k), consistent with blocking egress from lymph nodes but not spleen. These data indicate that RORγt⁺ OT-II pT$_{reg}$ cells can differentiate independently in the ILN, mLN and spleen. After OT-II pT$_{reg}$ cell induction by TGM1–IL-2(WT) followed by OVA-supplemented drinking water to promote gut homing, we detected substantial populations of RORγt⁺ OT-II pT$_{reg}$ cells with low GATA3 expression in Peyer's patches, small intestine and colon, demonstrated by both frequencies and numbers (Extended Data Fig. 6a–d).

Low-affinity TGM1 increased proportions of CD4⁺ follicular helper T cells (T$_{FH}$ cells; FOXP3⁻PD-1⁺BCL6⁺) relative to PBS, whereas TGM1–IL-2 slightly reduced this population (Fig. 2f and Extended Data Fig. 6e). IL-2(N88D) and TGM1–IL-2(H9) notably increased type 2 helper CD4⁺ T cell (T$_H$2 cell; FOXP3⁻GATA3⁺) frequencies, whereas TGM1–IL-2(N88D) and TGM1–IL-2(WT) did not (Fig. 2f). Low-affinity TGM1 also substantially increased IL-17A-producing and FOXP3⁻RORγt⁺ CD4⁺ type 17 helper T cell (T$_H$17 cell) percentages, whereas TGM1–IL-2 only modestly increased IL-17A⁺ cell proportions (Fig. 2f,g and Extended Data Fig. 6f). By contrast, IL-2 increased IFNγ- and TNF-producing cell frequencies, whereas TGM1–IL-2 strongly reduced them (Fig. 2g and Extended Data Fig. 6f,g). These results indicate that TGM1–IL-2 suppresses CD4⁺ type 1 helper T cell (T$_H$1 cell) and T$_{FH}$ cell development while modestly enhancing T$_H$17 differentiation in OVA-immunized mice.

OT-II pT$_{reg}$ cells induced by TGM1–IL-2, particularly the wild-type and H9 variants, contained a large fraction of CD103⁺ and IL-10-producing cells and exhibited increased expression of activation markers (CD25, ICOS, CD69 and GITR), suppressive molecules (CTLA4, CD39, TIGIT

and CD73) and the chemokine receptor CXCR3, exceeding levels in endogenous T$_{reg}$ cells (Fig. 2h,i and Extended Data Fig. 6h,i). By contrast, they expressed modest to low levels of NRP1 and CD62L (Extended Data Fig. 6h,i). These findings demonstrate that TGM1–IL-2 induces highly activated, functionally potent pT$_{reg}$ cells in vivo.

## Surrogate co-agonist establishes allergic tolerance

We next tested whether TGM1–IL-2–induced pT$_{reg}$ cells confer in vivo tolerance in an OVA-induced type 2 airway inflammation model, in which mice were sensitized with aluminium-adjuvanted OVA and then re-challenged intranasally with OVA after OT-II pT$_{reg}$ cell induction (Fig. 3a). TGM1–IL-2(WT) pretreatment markedly protected against allergic airway inflammation compared with PBS, as evidenced by lower lung inflammation scores, reduced serum total IgE and OVA-specific IgE/IgG1, and decreased infiltration of CD45.2⁺ immune cells (including CD11b⁺ myeloid cells and CD3⁺ T cells) in lung and bronchoalveolar lavage fluid (BALF), with a pronounced reduction in eosinophil frequency and numbers, whereas IL-2(WT) or low-affinity TGM1 provided no meaningful benefit (Fig. 3b–g and Extended Data Fig. 7a,b). Moreover, in TGM1–IL-2(WT)–treated mice, up to 80% of OT-II cells in the lung, lung-draining lymph nodes (LDLNs) and spleen persisted as FOXP3⁺ pT$_{reg}$ cells, with high absolute numbers (Fig. 3h–k and Extended Data Fig. 7c). Lung-infiltrating OT-II pT$_{reg}$ cells were predominantly GATA3⁺RORγt⁻, whereas those in LDLN and spleen were mainly GATA3⁻RORγt⁺ (Fig. 3h–j and Extended Data Fig. 7c). LDLNs also contained abundant CD25⁺ and IL-10⁺ OT-II pT$_{reg}$ cells with high expression of ICOS, CD103 and CD39 (Extended Data Fig. 7d,e). By contrast, mice treated with PBS, IL-2(WT) or low-affinity TGM1 showed few FOXP3⁺ OT-II pT$_{reg}$ cells, and IL-2(WT) instead increased GATA3⁺FOXP3⁻ T$_H$2 OT-II cell frequency (Fig. 3h–k and Extended Data Fig. 7c,f).

We further evaluated whether TGM1–IL-2 induced tolerance to ameliorate OVA-induced food allergy by sensitizing mice with OVA and cholera toxin after OT-II pT$_{reg}$ cell induction (Fig. 3l). Compared with PBS controls, TGM1–IL-2(WT) pretreatment led to smaller decreases in rectal temperature and reduced serum OVA-specific IgE and IgG1, indicating attenuated allergic responses (Fig. 3m,n). Notably, in mice pre-treated with TGM1–IL-2(WT), a large proportion of OT-II cells in the mLN, Peyer's patches and small intestine persisted as FOXP3⁺ pT$_{reg}$ cells, predominantly RORγt⁺ in both frequency and absolute number, whereas few OT-II pT$_{reg}$ cells were detected in PBS controls (Fig. 3o–q). These findings demonstrate that TGM1–IL-2 establishes tolerance that suppresses OVA-induced airway inflammation and food allergy, probably by generating stable, functional pT$_{reg}$ cells.

## Surrogate co-agonist suppresses neuroinflammation

We next evaluated TGM1–IL-2 in the MOG$_{35–55}$-induced experimental autoimmune encephalomyelitis (EAE) model of neuroinflammation (Fig. 4a). TGM1–IL-2(WT) robustly drove donor MOG$_{35–55}$-reactive

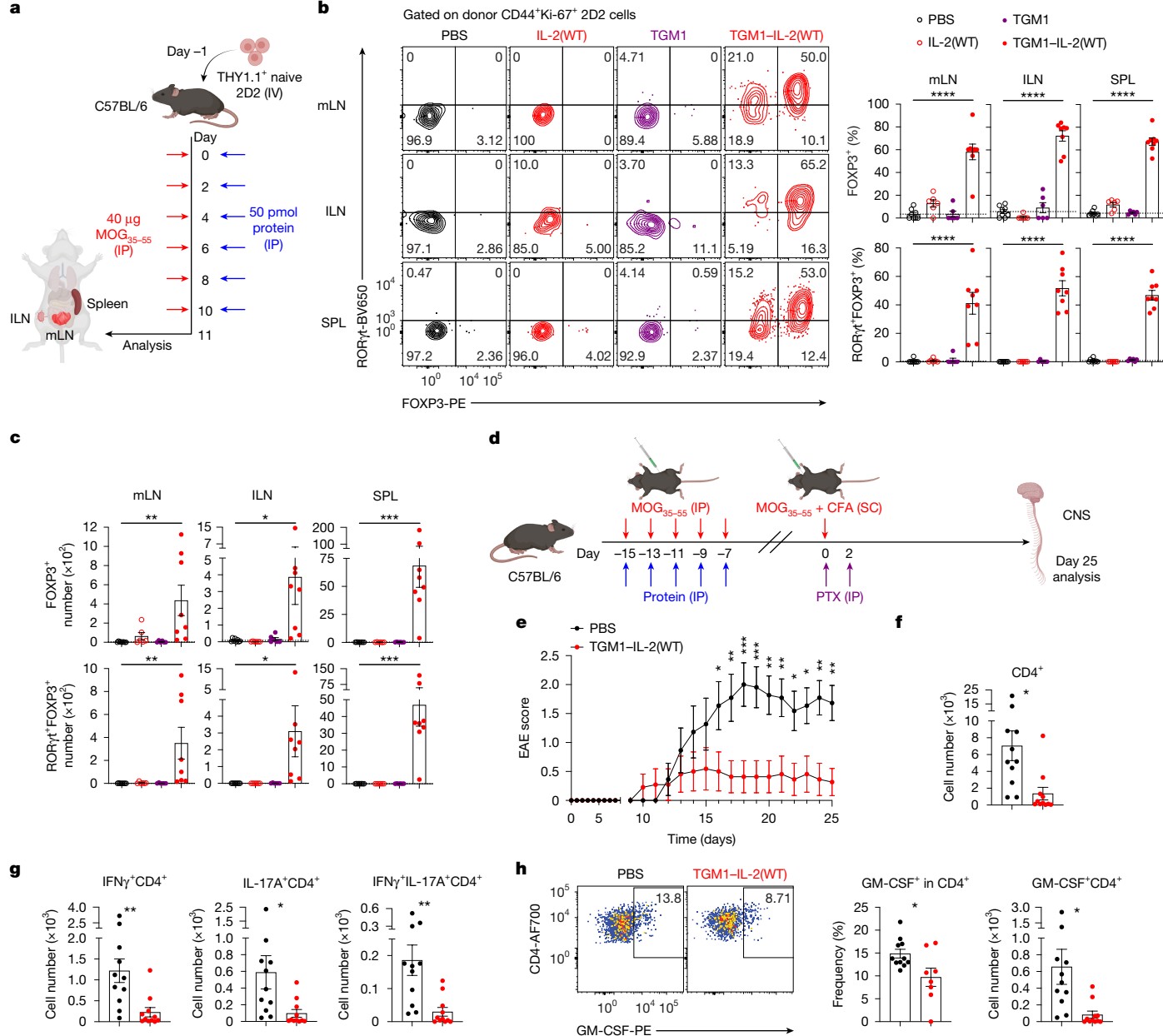

**Fig. 4 | IL-2–TGFβ co-agonists establish immune tolerance to suppress MOG-induced EAE. a**, Schematic illustration of MOG$_{35-55}$ and protein administration. Created in BioRender; Sun, Q. https://BioRender.com/j3crwdr (2026). **b,c**, Representative flow cytometry plots (**b**) and quantification of FOXP3$^+$ and RORγt$^+$FOXP3$^+$ cell frequencies (**b**) and absolute numbers (**c**) among donor CD44$^+$Ki-67$^+$ 2D2 cells in the indicated lymphoid organs ($n$ = 8 samples per group). **d–h**, Therapeutic effect of TGM1–IL-2 in establishing tolerance to suppress MOG$_{35-55}$-induced EAE ($n$ = 11 mice per group). **d**, Schematic illustration. CNS, central nervous system; PTX, pertussis toxin; SC, spinal cord; CFA, complete Freund's adjuvant. Created in BioRender; Sun, Q. https://BioRender.com/t3trkm2

(2026). **e**, Quantification of mean EAE scores. **f**, Quantification of CD4$^+$ T cell absolute numbers in the spinal cord. **g**, Quantification of IFNγ- and IL-17A-producing CD4$^+$ T cell numbers in the spinal cord. **h**, Representative flow cytometry plots and quantification of GM-CSF$^+$CD4$^+$ T cell frequencies and absolute numbers in the spinal cord. Data are presented as mean ± s.e.m. Data in **b,c** are pooled from two independent experiments. Data in **d–h** are representative of two independent experiments. Statistics were obtained by one-way ANOVA coupled with Dunnett's multiple-comparisons test (**b,c**), two-way ANOVA coupled with Šídák's multiple-comparisons test (**e**) or unpaired Welch's $t$-test (two-tailed) (**f–h**).

CD44$^+$Ki-67$^+$ 2D2 cells to differentiate to FOXP3$^+$ pT$_{reg}$ cells in mLNs, ILNs and spleens, with a substantial fraction expressing RORγt$^+$, as indicated by both frequencies and numbers, whereas PBS, IL-2(WT) or TGM1 induced few pT$_{reg}$ cells (Fig. 4b,c and Extended Data Fig. 8a). These 2D2 pT$_{reg}$ cells also upregulated activation and functional markers (CD25, ICOS, CTLA4, CD39, CD103 and IL-10) (Extended Data Fig. 8b), indicating that TGM1–IL-2 robustly induces functional 2D2 pT$_{reg}$ cells in peripheral lymphoid organs of MOG$_{35-55}$-immunized mice.

Mice pretreated with TGM1–IL-2(WT) showed markedly reduced EAE clinical scores after rechallenging with MOG$_{35-55}$ emulsified in complete Freund's adjuvant (9 out of 11 remained EAE-free), whereas nearly all control mice developed disease (Fig. 4d,e, and Extended Data Fig. 8c). TGM1–IL-2(WT) markedly reduced CD45.2$^+$ immune infiltration in the spinal cord, including CD11b$^+$CD3$^-$ myeloid cells, CD11b$^-$CD3$^+$ T cells and CD4$^+$ T cells, and also decreased IFNγ- and IL-17A-producing CD4$^+$ T cell numbers (despite similar frequencies) (Fig. 4f,g, and Extended Data Fig. 8d,e). Moreover, TGM1–IL-2(WT) reduced GM-CSF-producing CD4$^+$ T cell percentage and numbers, indicating a decrease in pathogenic T$_H$17 cells (Fig. 4h). These results demonstrate that TGM1–IL-2 establishes tolerance that protects mice from EAE.

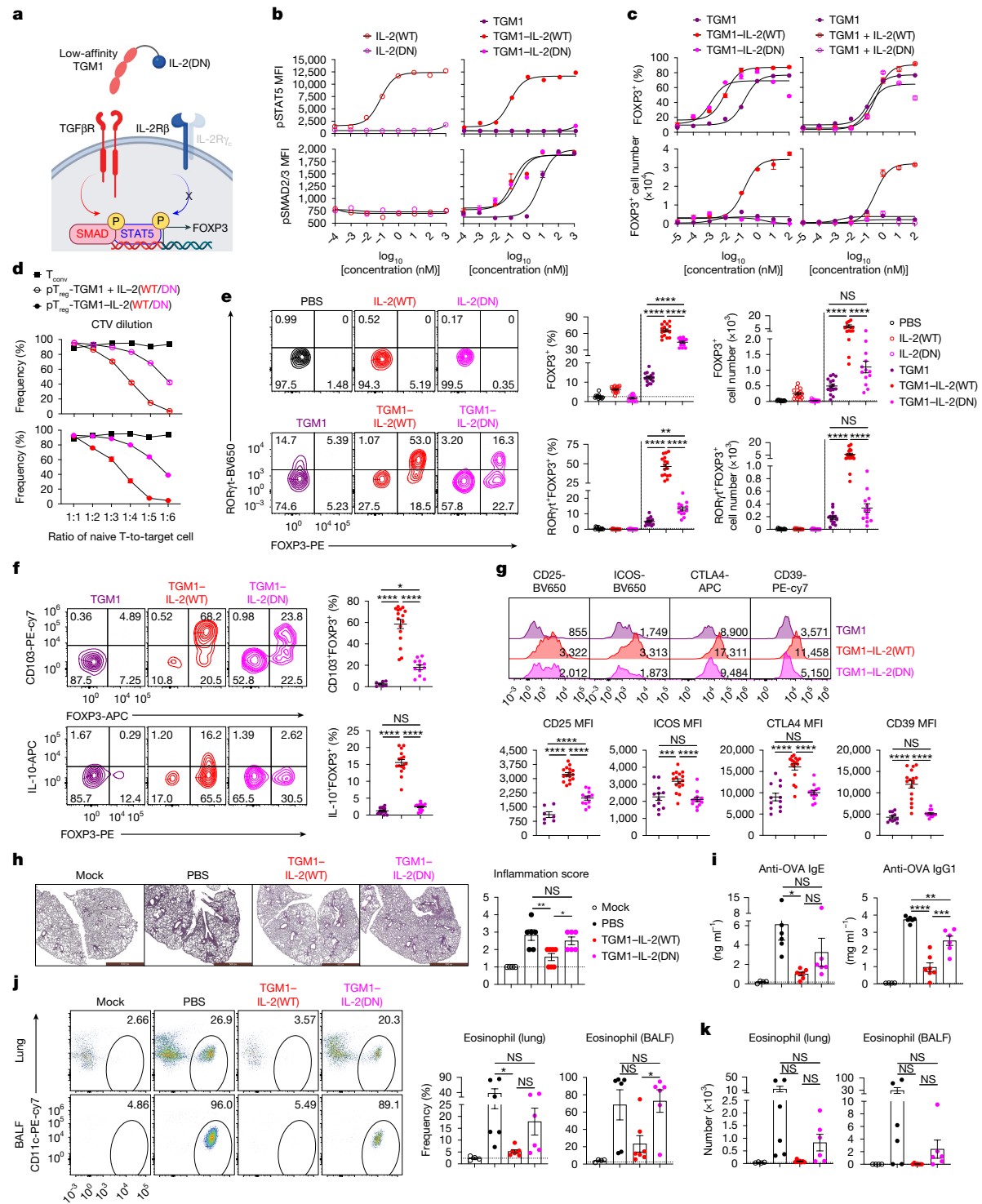

**Fig. 5 | IL-2–TGFβ co-agonists activate IL-2 signalling to enable optimal pT_reg cell development and immune tolerance. a**, Schematic illustration of TGM1–IL-2(DN). Created in BioRender; Sun, Q. https://BioRender.com/enpd8hs (2026). **b**, Ex vivo dose–response curves of pSTAT5 and pSMAD2/3 in primary CD25⁺CD4⁺ T cells (n = 2 biological replicates). **c**, In vitro dose–response curves for FOXP3⁺ cell frequency and number in cultured mouse naive CD4⁺ T cells (n = 2 biological replicates). **d**, CTV dilution of naive T cells co-cultured in vitro with the indicated cells (n = 2 biological replicates). **e–g**, Quantification and phenotypic analysis of OT-II pT_reg cells in the lymph nodes (n = 16 samples per group). **e**, Representative flow cytometry plots and quantification of FOXP3⁺ and RORγt⁺FOXP3⁺ OT-II cell frequencies and absolute numbers. **f**, Representative flow cytometry plots and quantification of CD103⁺FOXP3⁺ and IL-10⁺FOXP3⁺

OT-II cell frequencies. **g**, Representative flow cytometry plots and quantification showing expression of the indicated molecules on OT-II pT_reg cells. **h–k**, Therapeutic effect of TGM1–IL-2 in OVA-induced airway inflammation model (mock, n = 4 mice; PBS, n = 6 mice; TGM1–IL-2(WT), n = 7 mice; TGM1–IL-2(DN), n = 6 mice). **h**, Representative lung histology and quantification of inflammation scores. Scale bars, 1 mm. **i**, Quantification of serum levels of OVA-specific IgE and IgG1. **j**,**k**, Representative flow cytometry plots (**j**) and quantification of CD11c⁻SIGLECF⁺ eosinophil frequencies (**j**) and absolute numbers (**k**) among CD45.2⁺CD11b⁺ cells in the indicated tissues. Data are presented as mean ± s.e.m. Data in **b–k** are representative of two independent experiments. Statistics were obtained by one-way ANOVA coupled with Tukey's multiple-comparisons test (**e–k**).

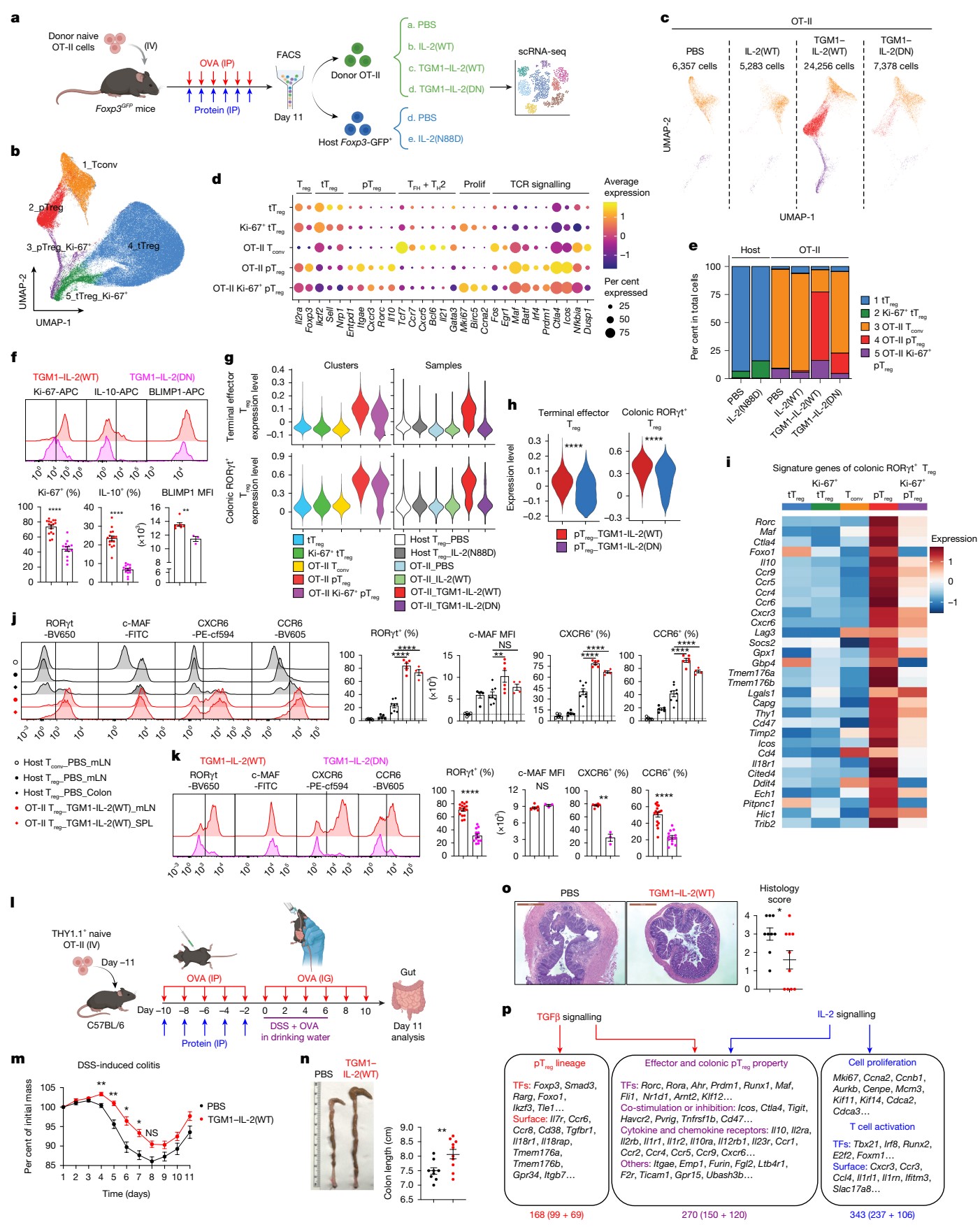

**Fig. 6 |** See next page for caption.

**Fig. 6 | Transcriptomic profiling of pT_reg cells induced by IL-2–TGFβ co-agonists in vivo. a**, Schematic illustration of the scRNA-seq sample collection workflow. FACS, fluorescence-activated cell sorting. Created in BioRender; Sun, Q. https://BioRender.com/dgxkovg (2026). **b**, Uniform manifold approximation and projection (UMAP) plot showing distinct cell clusters. **c**, UMAP plots showing the distribution of OT-II cells across clusters. **d**, Bubble plot illustrating the expression of selected genes across clusters. Prolif, proliferation. **e**, Bar plots showing the distribution of cells within distinct clusters. **f**, Representative flow cytometry plots and quantification of Ki-67 and IL-10 expression ($n$ = 16 samples per group) and BLIMP1 expression ($n$ = 7 samples per group) in OT-II pT_reg cells from lymph nodes. **g**, Violin plots showing the expression of terminal effector (Gene Expression Omnibus (GEO) GSE207969) and colonic RORγt⁺ T_reg cell (GEO GSE160053) signature gene sets across indicated clusters and samples. **h**, Violin plots showing expression of the indicated gene sets in cells from clusters 2 and 3. **i**, Heat map of selected gene expression across clusters. **j**, Representative flow cytometry plots and quantification of expression of the indicated molecules in the specified CD4⁺ T cell subsets ($n$ = 8 samples per group). **k**, Representative flow cytometry plots and quantification of RORγt and CCR6 expression ($n$ = 16 samples per group) and c-MAF and CXCR6 expression ($n$ = 7 samples per group) in OT-II pT_reg cells from lymph nodes. **l–o**, Therapeutic effect of TGM1–IL-2 in the dextran sulfate sodium (DSS)-induced colitis model (PBS, $n$ = 9 mice; TGM1–IL-2, $n$ = 10 mice). **l**, Schematic illustration. Created in BioRender; Sun, Q. https://BioRender.com/0fn86uv (2026). **m**, Quantification of mean changes in body mass. **n**, Representative colon images and quantification of colon lengths. **o**, Representative colon histology (left) and inflammation scores (right). Scale bars, 1 mm. **p**, Gene regulatory networks and signature genes identified by scRNA-seq analysis, enumerated below. Numbers in parentheses below boxes (bottom) represent the specific values from the Venn diagram in Extended Data Fig. 12f (left). TFs, transcription factors. Data are presented as mean ± s.e.m. Data in **f** and **j–o** are representative of two independent experiments. Statistics were obtained by unpaired Welch's $t$-test (two-tailed) (**f,h,k,n,o**), one-way ANOVA coupled with Tukey's multiple-comparisons test (**j**) or two-way ANOVA coupled with Šídák's multiple-comparisons test (**m**).

## Role of IL-2 in pT_reg cell induction and tolerance

To investigate the contribution of IL-2 signalling in TGM1–IL-2, we generated TGM1–IL-2(DN) by fusing TGM1 to the IL-2 mutant H9-RETR, which increases IL-2Rβ affinity but abolishes common γ-chain binding and thus IL-2R signalling[30] (Fig. 5a, Extended Data Fig. 9a and Supplementary Fig. 1). As expected, similar to IL-2(DN), TGM1–IL-2(DN) did not induce pSTAT5, but it triggered pSMAD2/3 with a dose–response curve resembling that of TGM1–IL-2(WT) (Fig. 5b). In vitro, TGM1–IL-2(DN) induced FOXP3⁺ pT_reg cells in a dose-dependent manner similar to TGM1–IL-2(WT), but markedly reduced total CD4⁺ T cell and pT_reg cell numbers and suppressed CD25 and T-bet expression and IFNγ production in CD4⁺ T cells (Fig. 5c and Extended Data Fig. 9b–d). In suppression assays, pT_reg cells generated with TGM1–IL-2(DN) (or TGM1 + IL-2(DN)) were less suppressive than those induced by TGM1–IL-2(WT) (or TGM1 + IL-2(WT)), as shown by reduced inhibition of T_conv cell proliferation (CTV dilution) and higher CD25 expression (Fig. 5d and Extended Data Fig. 9e). These results suggest that IL-2 signalling activated by TGM1–IL-2 is critical for optimal expansion and suppressive function of induced pT_reg cells in vitro.

In in vivo OT-II pT_reg cell induction assays, IL-2(DN) and TGM1–IL-2(DN) caused no notable weight loss or splenomegaly (Extended Data Fig. 9f,g). Compared with TGM1–IL-2(WT), TGM1–IL-2(DN) induced a lower fraction of OT-II pT_reg cells in the lymph nodes, particularly RORγt⁺ pT_reg cells; however, relative to low-affinity TGM1, TGM1–IL-2(DN) increased pT_reg cell frequency while reducing T_FH and T_H17 cells (Fig. 5e and Extended Data Fig. 9h). TGM1–IL-2(DN) also markedly decreased OT-II cell numbers and Ki-67⁺ cell proliferation, resulting in fewer total and RORγt⁺ OT-II pT_reg cells, and reduced T-bet expression and IFNγ production in OT-II cells (Fig. 5e and Extended Data Fig. 9i-k). Phenotypically, OT-II pT_reg cells induced by TGM1–IL-2(DN) showed reduced proportions of CD103⁺ and IL-10⁺ cells, lower expression of activation markers (CD25, ICOS and GITR), suppressive molecules (CTLA4, CD39 and CD73) and CXCR3, but increased NRP1 compared with TGM1–IL-2(WT) (Fig. 5f,g and Extended Data Fig. 9l). Notably, TGM1–IL-2(DN) conferred markedly less protection than TGM1–IL-2(WT) in the OVA-induced airway inflammation model, evidenced by higher lung inflammation scores, increased total serum IgE and OVA-specific IgE and IgG1, and greater infiltration of CD45.2⁺ immune cells (including CD11b⁺ myeloid cells and CD3⁺ T cells) in the lung and BALF, with a pronounced increase in eosinophil frequency and numbers (Fig. 5h–k and Extended Data Fig. 9m,n). Consistently, TGM1–IL-2(DN) markedly reduced OT-II pT_reg cell frequency and numbers in the LDLNs, particularly the CD25-expressing and IL-10-producing subsets (Extended Data Fig. 9o). These data show that IL-2 signalling in TGM1–IL-2 is required for optimal pT_reg cell differentiation, expansion and functional maturation in vivo, thereby establishing tolerance.

## Transcriptomic profiling of induced pT_reg cells

Next, we performed single-cell RNA sequencing (scRNA-seq) on donor OT-II cells from the OVA-immunized mice treated with PBS, IL-2(WT), TGM1–IL-2(WT) or TGM1–IL-2(DN), together with endogenous *Foxp3*-GFP⁺ CD4⁺ T cells from PBS or IL-2(N88D) groups as controls (Fig. 6a). Unsupervised dimensionality reduction identified five clusters, in which donor OT-II cells were distributed mainly across clusters 1, 2 and 3 and showed increased expression of TCR signalling genes (*Batf*, *Irf4*, *Egr1*, *Ctla4*, *Icos* and *Nfkbia*), consistent with OVA-driven activation (Fig. 6b–d). Cluster 1 was annotated as a T_conv population, showing minimal *Foxp3* expression but high levels of T_conv genes, including markers of T_FH (*Bcl6*, *Tox2*, *Cxcr5*, *Il21*, *Pdcd1* and *Slamf6*), T_H2 (*Gata3* and *Il4*), T_H1 (*Ifng*) and naive (*Tcf7*, *Slamf6* and *Ccr7*) cells; clusters 2 and 3 were annotated as pT_reg cell populations on the basis of enriched expression of pT_reg cell transcription factors (*Foxp3* and *Rorc*), surface molecules (*Cd25*, *Itgae*, *Nt5e*, *Entpd1* and *Tigit*), suppressive cytokines (*Il10*) and chemokine receptors (*Cxcr3*, *Ccr6*, *Ccr9* and *Cxcr6*) (Fig. 6b,d and Extended Data Fig. 10a–d). Cluster 3 was further defined as a proliferative Ki-67⁺ pT_reg cell subset on the basis of high expression of proliferation and cell cycle-associated genes and pathways, along with markedly increased S phase and G2/M phase cell cycle scores (Fig. 6b,d and Extended Data Fig. 10a,e,f). By contrast, endogenous *Foxp3*-GFP⁺ cells were mainly found in clusters 4 and 5, representing bystander tT_reg cell populations with high expression of *Foxp3*, *Ikzf2*, *Sell* and *Nrp1* and transcriptional profiles that were distinct from OT-II pT_reg cell clusters 2 and 3 (Fig. 6b,d and Extended Data Fig. 10g). Cluster 5 was further identified as a proliferative Ki-67⁺ tT_reg cell subset owing to increased expression of proliferation-associated genes (Fig. 6b,d and Extended Data Fig. 10a).

OT-II cells from the PBS and IL-2(WT) groups mapped almost exclusively to the T_conv cluster 1, whereas a substantial fraction from the TGM1–IL-2(WT) and TGM1–IL-2(DN) groups localized to the pT_reg clusters 2 and 3 (Fig. 6b,c,e). Consistently, OT-II cells from both the TGM1–IL-2(WT) and TGM1–IL-2(DN) groups, as well as cells in pT_reg clusters 2 and 3, showed higher expression of a TGFβ signalling gene set (Extended Data Fig. 11a and Supplementary Table 1). Notably, relative to the TGM1–IL-2(WT) group, a substantially smaller fraction of OT-II cells from the TGM1–IL-2(DN) group mapped to cluster 2, with minimal representation in cluster 3 (Fig. 6b,c,e). Consistently, TGM1–IL-2(WT) OT-II cells showed stronger enrichment of IL-2/STAT5-promoted gene sets and a higher proportion of Ki-67⁺ cells than TGM1–IL-2(DN) cells (Fig. 6f, Extended Data Fig. 11a and Supplementary Table 1). Similarly, endogenous *Foxp3*-GFP⁺ cells from IL-2(N88D)-treated mice were more enriched in the proliferating tT_reg cluster 5 than those from PBS-treated mice (Fig. 6b,e). These results further demonstrate that IL-2 is required for optimal pT_reg differentiation and to drive T_reg proliferation.

Moreover, OT-II cells in the $pT_{reg}$ and proliferating $pT_{reg}$ clusters 2 and 3, particularly from the TGM1–IL-2(WT) group, were strongly enriched for activated and terminal effector $T_{reg}$ gene signatures[31,32], whereas OT-II cells from the TGM1–IL-2(DN) group showed reduced enrichment (Fig. 6g,h, Extended Data Fig. 11b–d and Supplementary Table 1). Consistently, TGM1–IL-2(DN)-induced OT-II $pT_{reg}$ cells exhibited reduced IL-10 production and BLIMP1 expression compared with those induced by TGM1–IL-2(WT) (Fig. 6f). Moreover, a colonic $T_{reg}$ cell signature and a RORγt[+] colonic $T_{reg}$ cell programme from two independent studies were strongly enriched in OT-II cells from the TGM1–IL-2(WT) group, moderately enriched in the TGM1–IL-2(DN) group, and minimally expressed in endogenous $T_{reg}$ cells, and were also enriched in $pT_{reg}$ clusters 2 and 3 (refs. 33,34) (Fig. 6g,h, Extended Data Fig. 11e,f and Supplementary Table 1). Specifically, these cells expressed higher levels of genes encoding colonic RORγt[+] $T_{reg}$ transcription factors (*Rorc* and *Maf*), migration molecules (*Ccr4*, *Ccr5*, *Ccr6* and *Cxcr6*) and immunoregulatory molecules (*Il10*, *Ctla4*, *Tmem176a* and *Lgals1*) (Fig. 6i). Consistently, TGM1–IL-2(WT) induced higher protein expression of RORγt, c-MAF, CXCR6 and CCR6 in OT-II $pT_{reg}$ cells, with levels exceeding those in endogenous colonic $T_{reg}$ cells (Fig. 6j). However, OT-II cells from the TGM1–IL-2(DN) group, particularly the $pT_{reg}$ cell subpopulation, exhibited reduced expression of the RORγt[+] colonic $T_{reg}$ cell signature and decreased frequencies of RORγt[+], CXCR6[+] and CCR6[+] OT-II $pT_{reg}$ cells, whereas c-MAF levels remained similar (Fig. 6h,k). We then assessed whether these OT-II $pT_{reg}$ cells could suppress intestinal inflammation by administering DSS and OVA orally after cell induction (Fig. 6l). Mice pretreated with TGM1–IL-2(WT) exhibited delayed and attenuated weight loss, preserved colon length and reduced histological inflammation scores compared with PBS controls (Fig. 6m–o and Extended Data Fig. 12a). The proportion of IFNγ–producing colonic CD4[+] T cells decreased, whereas IL-17A–producing cells were unchanged (Extended Data Fig. 12b). Moreover, FOXP3[+] OT-II cells represented a substantially higher fraction of total CD4[+] T cells in the small intestine and colon compared with mLNs and Peyer's patches, and were predominantly RORγt[+]GATA3[-] (Extended Data Fig. 12c,d). These findings indicate that TGM1–IL-2 programs $pT_{reg}$ cells with an RORγt[+] effector, colonic-like phenotype in peripheral lymphoid organs, enhancing migration and suppression of intestinal inflammation.

Transcription factor activity analysis revealed increased FOXP3, SMAD3, RORγt and BLIMP1 activity in $pT_{reg}$ clusters 2 and 3, with reduced Helios and BCL6 activity (Extended Data Fig. 12e). Finally, an overlap analysis of differentially expressed genes identified 168 TGFβ-induced genes, 343 IL-2-induced genes, and 270 genes that were co-induced by both pathways (Fig. 6p, Extended Data Fig. 12f and Supplementary Table 2). TGFβ uniquely upregulated $T_{reg}$ cell identity transcription factors (*Foxp3*, *Smad3* and *Rarg*); IL-2 specifically induced proliferation (*Mki67*, *Ccna2*, *Ccnb1* and *Mcm3*) and activation (*Tbx21*, *Irf8*, *Runx2* and *Cxcr3*) genes; and coordinated activation of both pathways induced an effector and RORγt[+] $pT_{reg}$ programme that included transcription factors (*Rorc*, *Rora*, *Maf*, *Prdm1* and *Fli1*), immunoregulatory molecules (*Il10*, *Ctla4* and *Tigit*), activation markers (*Il2ra*, *Icos*, *Havcr2* and *Pvrig*), migration-related genes (*Ccr5*, *Ccr9*, *Cxcr6* and *Itgae*) and other functional mediators (*Emp1*, *Furin*, *Ltb4r1* and *F2r*) (Fig. 6p). Similarly, we identified 132 TGFβ-suppressed genes, 548 IL-2-suppressed genes, and 483 genes that were co-suppressed by activation of both pathways (Extended Data Fig. 12f,g and Supplementary Table 3). TGFβ alone suppressed activation (*Tbx21*, *Nr4a2*, *Il2*, *Il12rb2*, *Gzmk* and *Nkg7*) and proliferation genes (*Cdca3*, *Cdca5*, *Ccnb1*, *Ccna2* and *Cdc25c*), opposing IL-2 induced signatures; IL-2 alone suppressed naive and stemness genes (*Tcf7*, *Id3*, *Ikzf2*, *Sox4* and *Tgfbr3*); and combined TGFβ and IL-2 suppressed alternative CD4[+] T cell lineage programmes, including $T_{FH}$ (*Bcl6*, *Ascl2* and *Il21*), $T_{H}1$ (*Ifng*) and $T_{H}2$ (*Il4*) programmes (Extended Data Fig. 12g). Thus, although TGFβ and IL-2 antagonistically regulate proliferation and activation, their coordination together with TCR stimulation robustly drives $pT_{reg}$ cell differentiation, expansion and

functional maturation while suppressing alternative CD4[+] T cell fates in vivo (Extended Data Fig. 12h).

## Discussion

TGFβ is a potent but dangerous and challenging cytokine to harness for therapeutic applications[23], yet modulating $T_{reg}$ cells is a desirable therapeutic area for TGFβ agonism in autoimmune and inflammatory diseases. Current $T_{reg}$ cell therapies mainly expand polyclonal $tT_{reg}$ cells using IL-2 agents, but the expansion is short-lived, tissue-specific precursors remain low, and bystander $tT_{reg}$ cells lack antigen specificity, limiting efficacy and increasing systemic immunosuppression risk[35,36]. As an alternative, antigen-specific $pT_{reg}$ cells offer targeted control of pathological immune responses while preserving overall immune function[37], and our findings advance this paradigm by using a designed IL-2–TGFβ co-agonist to robustly induce these cells in vivo. This agonist also suppresses $T_{H}1$ and $T_{FH}$ cell differentiation, providing dual immunosuppression by inducing pTreg cells and inhibiting pro-inflammatory subsets. Although the helminth-derived TGM1 component in this co-agonist (D1–D3) has a foreign origin, we observed no loss of exposure in vivo, but anti-drug antibodies could still limit its use in humans. TGM1 is composed of tandem immunoglobulin domains resembling variable heavy chain ($V_{H}H$) fragments, so a more human-tolerable drug-like TGFβ mimic could be built from tandem $V_{H}H$ domains, which have been shown to act as surrogate cytokine agonists[38].

RORγt[+] APCs have been implicated in inducing food antigen and microbiota-driven $pT_{reg}$ cells in the gut[15–19,39–42]. Unlike oral antigen administration, intraperitoneal antigens plus TGM1–IL-2 efficiently induces $pT_{reg}$ cells systemically, including in lymphoid organs that are distant from the gut, suggesting that it is likely to provide sufficient in vivo inductive signals without requiring RORγt[+] APCs. Additionally, the instability of $pT_{reg}$ cells limits their therapeutic use, but in vivo TGM1–IL-2-induced $pT_{reg}$ cells persist and remain functionally stable after multiple antigen re-challenges in disease, suggesting that coordinated IL-2 and TGFβ signalling may also be sufficient to generate durable $pT_{reg}$ cells without co-factors (such as retinoic acid or vitamin C)[43,44]. $T_{reg}$ cell development is governed by a transcriptional network involving FOXP3, RORγt and BLIMP1 that integrates TGFβ and IL-2 signals to ensure appropriate $T_{reg}$ cell differentiation and function after antigen stimulation[45]. This $T_{reg}$ programming framework may provide a platform to reprogram other CD4[+] T cell lineage commitments using bi-specific agonists that pair distinct cytokine-STAT programmes with TGFβ–SMAD2/3 signalling to induce lineage-specific transcription factors.

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

## Methods

### Mice

Six-to-eight-week-old female and male C57BL/6 J mice (IMSR_JAX:000664), as well as other strains, were purchased from The Jackson Laboratory. OT-II (IMSR_JAX:004194) and Thy1.1 (IMSR_JAX:000406) mice were crossed to generate OT-II Thy1.1 mice. *Foxp3*-GFP mice (IMSR_JAX:006772) were crossed with OT-II Thy1.1 mice. 2D2 mice (IMSR_JAX:006912) were crossed with Thy1.1 mice. All animals were housed in AAALAC-accredited facilities. Sample sizes were not pre-determined but are reported with each result, and randomization was performed across littermates.

### Flow cytometry

The following antibodies were purchased from BioLegend: mouse CD45.2 (109839), CD3 (100206), CD4 (100453, 100430, 100428, 100451), CD8 (100706), NK1.1 (156506), TCRα2 (127806, 127822), TCRα3.2 (135404), Thy1.1 (202528, 202522), CTLA4 (106310), CD62L (104453), CD25 (102012, 102022, 102047, 102038), CD44 (103026, 103032), SIGLECF (155534), CD73 (127215), ICOS (313550), CD69 (104530), CD11b (101259), CXCR3 (126514), CD39 (143806), NRP1 (145218), CD11c (117318), CD103 (110910), GITR (126316), CXCR6 (151117), CCR6 (129819), IL-17A (506928), GM-CSF (505406), IFNγ (505832, 505826), IL-10 (505034, 505026, 505034), BLIMP1 (150008), Helios (137214), Ki-67 (151212, 652406), TNFα (506346); human CD3 (317324), CD4 (980806) and FOXP3 (320126). The following antibodies were purchased from BD Biosciences: mouse BCL6 (562401), RORγt (564722, 562682, 562683), SMAD2 (pS465/pS467)/SMAD3 (pS423/pS425) (562696) and STAT5 (pY694) (612599). The following antibodies and reagents were purchased from Invitrogen: mouse PD-1 (48-9985-82), FOXP3 (12-5773-82, 17-5773-82, 404-5773-82), T-bet (25-5825-82), c-MAF (53-9855-82), GATA3 (46-9966-42) and Fixable Viability Dye (65-0865-18). PE- and Brilliant Violet 421-labelled I-A$^b$ OVA$_{328-337}$ tetramers (HAAHAEINEA) were provided by the NIH Tetramer Core Facility.

For surface marker staining, live cells were incubated with antibodies and viability dye in PBS at 4 °C for 1 h. Dead cells were excluded on the basis of viability dye staining. For I-A$^b$ OVA$_{328-337}$ tetramer staining, live cells were first incubated with tetramers in PBS at 37 °C for 1 h, followed by surface antibody and viability dye staining. For transcription factor and cytokine staining, cells were fixed and permeabilized using the FOXP3/Transcription Factor Staining Buffer Set (00-5521-00, Invitrogen), followed by intracellular staining at room temperature for 2 h. Prior to cytokine staining, cells were stimulated with Cell Stimulation Cocktail (00-4970-03, Invitrogen) and Protein Transport Inhibitor Cocktail (00-4980-03, Invitrogen) at 37 °C for 5 h. For pSMAD2/3 and pSTAT5 staining, freshly isolated T cells were stimulated ex vivo with the indicated proteins for 25 min, then were immediately fixed with Cytofix Fixation Buffer (554655, BD) and permeabilized using Phosflow Perm Buffer III (558050, BD). Cells were resuspended in PBS and analysed on CytoFLEX flow cytometer (Beckman Coulter). Data analysis was performed using FlowJo software v.10.10.0.

### Protein production

Recombinant proteins were cloned into the pD649 mammalian expression vector (ATUM), which includes a haemagglutinin (HA) secretion signal peptide, an N-terminal MSA fusion, and a C-terminal 6×His tag. Expression constructs were transfected into Expi293F cells using the Expi293 Expression System (Gibco). After 3–4 days of culture, proteins were purified from supernatants by Ni-NTA Agarose (Qiagen), followed by size-exclusion chromatography using a Superdex 200 column in ÄKTA chromatography system (Cytiva). Endotoxin was removed using the Proteus NoEndo HC Spin Column Kit (VivaProducts), and levels were confirmed to be acceptable using the Pierce Chromogenic Endotoxin Quant Kit (Thermo Fisher Scientific). Final protein preparations were formulated and concentrated in sterile PBS, flash-frozen in liquid nitrogen, and stored at −80 °C until use.

### CD4$^+$ T cell isolation and in vitro differentiation

Mouse lymph nodes and spleens were collected and mechanically dissociated to obtain single-cell suspensions. Red blood cells were lysed using ACK lysis buffer (A10492-01, Gibco), followed by magnetic isolation of CD4$^+$ T cells using the EasySep Mouse CD4$^+$ T Cell Isolation Kit (19852, STEMCELL). Naive CD4$^+$ T cells (CD4$^+$CD44$^-$CD25$^-$*Foxp3*-GFP$^-$) and activated CD4$^+$ T$_{conv}$ cells (CD4$^+$CD44$^+$CD25$^-$*Foxp3*-GFP$^-$) were subsequently sorted using a Sony SH800S Cell Sorter. The purity of the sorted populations was consistently greater than 99%. Human CD4$^+$ T cells were isolated from frozen PBMCs using the EasySep Human CD4$^+$ T Cell Isolation Kit (17952, STEMCELL). Complete medium was prepared using RPMI 1640 with GlutaMAX supplement (Gibco, 61870036) and supplemented with 10% fetal bovine serum (Gibco, A5256701), 10 mM HEPES (Gibco, 15630080), 1% sodium pyruvate (Gibco, 11360070), 1% penicillin–streptomycin (Gibco, 15140122) and 0.1% 2-mercaptoethanol (Gibco, 21985023).

For the in vitro mouse CD4$^+$ T cell differentiation assay, naive CD4$^+$ T cells were plated at $0.75 \times 10^6$ cells per ml in flat-bottom 96-well plates pre-coated overnight with 5 μg ml$^{-1}$ InVivoMAb anti-mouse CD3 (Bio X Cell, BE0002) and 5 μg ml$^{-1}$ InVivoMAb anti-mouse CD28 (Bio X Cell, BE0015-1). For OT-II cell differentiation, mouse splenocytes were plated at $1.5 \times 10^6$ cells per ml in flat-bottom 96-well plates and stimulated with 0.05 μg ml$^{-1}$ OVA$_{323-339}$ peptide (GenScript, RP10610). For human CD4$^+$ T cell differentiation, CD4$^+$ T cells were plated at $0.75 \times 10^6$ cells per ml in flat-bottom 96-well plates pre-coated overnight with InVivoMAb anti-human CD3 (Bio X Cell, BE0001-2) and 5 μg ml$^{-1}$ InVivoMAb anti-human CD28 (Bio X Cell, BE0248). Recombinant proteins were added at the indicated concentrations at the start of the culture, and cells were incubated at 37 °C for ~4 days.

### In vitro suppression assay

Thy1.1$^+$ mouse naive CD4$^+$ T cells were labelled with CellTrace Violet Cell Proliferation Dye (Invitrogen, C34571) and plated at $0.1 \times 10^6$ cells per well in flat-bottom 96-well plates pre-coated with 5 μg ml$^{-1}$ anti-CD3 and 5 μg ml$^{-1}$ anti-CD28 antibodies. Thy1.1$^-$ mouse T$_{conv}$ cells or in vitro–differentiated pT$_{reg}$ cells generated with 1 nM proteins were added at the indicated ratios. Thy1.1$^+$ cells were analysed after 48 h of co-culture.

### Naive OT-II cell transfer and OVA administration

One million sorted Thy1.1$^+$ naive OT-II cells were adoptively transferred into Thy1.1$^-$ C57BL/6 recipient mice via retro-orbital intravenous injection. Starting one day post-transfer, mice were administered 100 μg OVA protein (A5503, Sigma-Aldrich) along with 50 pmol of the indicated proteins via intraperitoneal injection every other day, for a total of six injections. On day 11, mLN, ILN, and spleens were collected for analysis. For FTY720 treatment, 20 μg FTY720·HCl (ENZO Life Sciences) was dissolved in 100 μl of 5% DMSO in PBS and administered via intraperitoneal injection daily from day 0 to day 10, for a total of 11 doses.

### OVA-induced airway inflammation model

One million sorted Thy1.1$^+$ naive OT-II cells were adoptively transferred into Thy1.1$^-$ C57BL/6 recipient mice, followed by five intraperitoneal injections of 100 μg OVA protein combined with 50 pmol of the indicated proteins. One week after the final injection, mice received three weekly intraperitoneal injections of 100 μg OVA protein formulated in 150 μl Alhydrogel adjuvant (2%; InvivoGen, vac-alu-50). One week after the last adjuvant injection, mice were administered three intranasal doses of 100 μg OVA protein every other day. Tissues were collected and analysed one day after the final intranasal dose.

## OVA-induced food allergy model

One million sorted Thy1.1+ naive OT-II cells were adoptively transferred into Thy1.1– C57BL/6 recipient mice, followed by five intraperitoneal injections of 100 µg OVA protein combined with 50 pmol of the indicated proteins. One week after the final injection, mice received three weekly oral gavages of 5 mg OVA protein together with 10 µg cholera toxin (C8052, Sigma-Aldrich). Donor OT-II cells in the gut were analysed one day after the last gavage. One week later, mice were challenged with 200 µg OVA protein via intraperitoneal injection. Rectal temperature was recorded immediately thereafter every 5–10 min for 75 min using a Type J/K/T thermocouple thermometer (Kent Scientific), and serum was collected one day later for analysis.

## Naive 2D2 cell transfer and MOG$_{35-55}$ administration

Three million sorted Thy1.1+ naive CD44–CD25– 2D2 cells were adoptively transferred into female Thy1.1– C57BL/6 recipient mice via retro-orbital intravenous injection. Beginning one day after transfer, mice received 40 µg MOG$_{35-55}$ peptide (Genemed Synthesis) together with 50 pmol of the indicated proteins by intraperitoneal injection every other day, for a total of six doses. On day 11, mLN, ILN and spleens were collected for analysis.

## MOG$_{35-55}$-induced EAE model

Female mice were administered 40 µg MOG$_{35-55}$ peptide together with 50 pmol of the indicated proteins every other day for a total of five doses. One week later, mice were immunized subcutaneously with 100 µg MOG$_{35-55}$ peptide per mouse in incomplete Freund's adjuvant (BD Biosciences) containing 200 µg *Mycobacterium tuberculosis* per mouse (BD Biosciences), injected at the axilla of both sides. Concurrently, 400 ng pertussis toxin per mouse (PTX, List Labs) was administered intraperitoneally, with a second dose given 48 h later. Mouse body weight and clinical signs of disease were recorded daily and scored according to the following scale: 1, tail paralysed; 1.5, mild hind limb weakness; 2, moderate/typical hind limb weakness; 2.5, severe hind limb weakness without paralysis; 3, one or both hind limbs paralysed, front limbs fully functional; 3.5, both hind limbs paralysed, front limbs/paws weak but not paralysed; 4, front limb paralysis; 5, moribund or deceased. Tissues were collected and analysed on day 25.

## DSS-induced colitis model

Two million sorted Thy1.1+ naive OT-II cells were adoptively transferred into Thy1.1– C57BL/6 recipient mice via retro-orbital intravenous injection. Beginning one day after transfer, mice received 100 µg OVA protein together with 50 pmol of the indicated proteins by intraperitoneal injection every other day, for a total of five doses. Two days later, the mice were given drinking water containing 2.5% DSS (colitis grade, 36,000–50,000; MP Biomedicals) supplemented with 2.5 mg ml$^{-1}$ OVA protein. Simultaneously, mice were administered 5 mg OVA protein via oral gavage every other day for a total of six doses. After six days, the DSS- and OVA-supplemented water was replaced with regular drinking water. Mouse body weight was recorded daily throughout the experiment, and tissues were collected and analysed at the endpoint.

## Isolation of immune cells from tissues

Mice were euthanized after completion of the respective treatments. BALF was collected by flushing the lungs three times with 0.75 ml of PBS via a catheter inserted into the trachea. For lymphocyte isolation from the lung, tissues were mechanically dissociated using the plunger of a 1 ml syringe and filtered through 70-µm cell strainers to obtain single-cell suspensions. For lymphocyte isolation from the lamina propria, Peyer's patches in the small and large intestines were first removed. The intestines were then opened longitudinally, cut into ~2-cm pieces, and incubated in 5 mM EDTA (15575020, Invitrogen) with 1 mM DTT (R0861, Thermo Fisher Scientific) at 37 °C for 30 min

to remove epithelial cells. Tissues were then minced and digested in DNase I (40 µg ml$^{-1}$; Roche) and collagenase D (0.5 mg ml$^{-1}$; Roche) at 37 °C for 30 min with shaking to generate single-cell suspensions, which were filtered through 70-µm cell strainers. For lymphocyte isolation from the spinal cord, mice were first perfused, and the collected spinal cords were mechanically dissociated using the plunger of a 1 ml syringe and passed through 70-µm cell strainers to obtain single-cell suspensions. The resulting cells were subjected to density gradient centrifugation using a 40%/70% Percoll (Cytiva) gradient. Immune cells located at the interface between the two Percoll layers were collected and processed for flow cytometry analysis.

## ELISA

Blood samples were centrifuged at 3,000*g*, and serum was collected from the supernatant. Immunoglobulin levels were measured using ELISA kits: total IgE (ELISA MAX Standard Set Mouse IgE; BioLegend, 432401), OVA-specific IgE (LEGEND MAX Mouse OVA-Specific IgE ELISA Kit; BioLegend, 439807) and OVA-specific IgG1 (Mouse Anti-OVA IgG1 Antibody Assay Kit; Chondrex, 3013).

## Histology

Lung tissues from perfused mice and colon tissues were fixed in 10% neutral buffered formalin (Sigma-Aldrich, HT501128) and submitted to S. Avolicino for paraffin embedding, sectioning, and haematoxylin and eosin staining. Slides were imaged on a Leica DM2000 microscope, and histopathology was evaluated in a blinded manner. Lung inflammation was scored on the basis of the extent of peribronchial and perivascular cellular infiltration: 0, no infiltrates; 1, a few inflammatory cells; 2, a one-cell-thick ring of inflammatory cells; 3, a 2–3-cell-thick ring; 4, a 4–5-cell-thick ring; 5, a ring >5 cells thick. Colon inflammation was scored as follows: 0, no evidence of inflammation; 1, low-level inflammation with scattered infiltrating mononuclear cells (1–2 foci); 2, moderate inflammation with multiple foci; 3, high-level inflammation with increased vascular density and marked wall thickening; 4, severe inflammation with transmural leukocyte infiltration and loss of goblet cells.

## scRNA-seq

Donor OT-II cells and endogenous *Foxp3*-GFP+ CD4+ T cells were sorted from the mLNs of the respective treatment groups on day 11 and submitted to MedGenome for library preparation and RNA sequencing. The FASTQ files were processed using Cell Ranger v.9.0.0. The gene expression matrix was processed and analysed using Seurat (v.5.1.0)[46]. For quality control, we excluded cells that contained fewer than 500 read counts for genes or fewer than 200 genes detected (minimal cutoff), or more than 50,000 read counts for genes or more than 6,500 genes detected (maximum cutoff). We also excluded cells in which more than 20% of transcripts were derived from mitochondrial RNA. These QC filters left 190,728 cells. Graph-based unsupervised clustering was employed to identify clusters representing minor contaminant cells other than T cells, such as neurons (expressing *Cntn1*, *Dscam* and *Pde7b*) and B cells (expressing *Igkc*, *Ms4a1* and *Cd79a*). These minor clusters were excluded from subsequent analyses, leaving 180,038 cells. UMAP embedding was computed with 10 principal components, with n.neighbors being 20 and min.dist being 0.1. Differential expression analysis between groups was performed using Wilcoxon's rank sum test implemented in Seurat's FindMarkers function. Gene set enrichment analysis[47] was performed using the log$_2$FC ranking of differentially expressed genes using fgsea. Human hallmark gene sets were retrieved from MSigDB. The signature scores were calculated using Seurat's AddModuleScore function. To calculate cell cycle score, we used the CellCycleScoring function in Seurat. Built-in human gene sets for the S and G2M phases in Seurat were converted into mouse homologues and used to calculate S and G2M scores. Transcription factor activity inference was conducted using pySCENIC with default parameter settings[48]. To identify differentially expressed genes from public bulk RNA-seq datasets for computing

gene set signature scores, bulk RNA-seq FASTQ files were aligned to the GENCODE VM25 (mm10) reference genome using Rsubread[49], and gene expression was quantified with featureCounts[50]. Differential expression analysis was performed using DESeq2 (ref. 51). Pathway analysis was performed using Metascape[52].

## Statistical analysis

Statistical analyses were performed using GraphPad Prism 10. Differences were considered statistically significant at $P < 0.05$, with significance denoted as follows: $*P < 0.05$, $**P < 0.01$, $***P < 0.001$ and $****P < 0.0001$. Detailed statistical information, including sample sizes, numbers of independent experiments, and the statistical tests used, is provided in the corresponding figure legends.

## Ethics statement

All experimental mouse procedures were approved by the Stanford University Institutional Animal Care and Use Committee (IACUC; protocol IDs 32279 and 34708) and conducted in accordance with institutional guidelines. Blood for PBMC isolation from healthy donors was provided by the Stanford Blood Center, which also obtained ethical approval for the donors.

## Reporting summary

Further information on research design is available in the Nature Portfolio Reporting Summary linked to this article.

## Data availability

The raw and processed scRNA-seq data have been deposited in the Gene Expression Omnibus (GEO) under accession GSE315102 (aligned to the mm10 mouse reference genome). Source data are provided with this paper.

## Code availability

The R code used to analyse the scRNA-seq data has been deposited on Zenodo (https://doi.org/10.5281/zenodo.18166788)[53].

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

**Acknowledgements** We thank the Stanford Animal Care and Facilities and the Stanford Cell Sciences Imaging Facility for technical assistance. We thank the US National Institutes of Health (NIH) Tetramer Core Facility (NIH contract 75N93020D00005 and RRID:SCR_026557) for providing OVA$_{328-337}$ tetramers. We acknowledge support for this research from Howard Hughes Medical Institute (K.C.G.), NIH-R01-AI-51321 (K.C.G.), NIH-R01-AI-173189 (T.V.L.), Department of Defense HT9425-23-1-059 (T.V.L.), Yosemite Innovation Fund (K.C.G.) and Ludwig Institute (K.C.G.). M. Ogishi is supported by the NCI Predoctoral to Postdoctoral Fellow Transition (K00) award (4K00CA274708).

**Author contributions** K.C.G. conceived the study, supervised the experiments and collaborated with the other authors on the manuscript. Q.S. designed and conducted all experiments, analysed the data and prepared the figures, manuscript and related materials. A.K.B. established the EAE model, recorded clinical scores and performed mouse perfusions. M. Ogishi processed the scRNA-seq data and wrote the corresponding methods section. H.J., Y.Z., G.E.R. and K.D.H. assisted with mouse tissue processing. H. Liu assisted with BALF collection. G.E.R. provided the vector backbone and IL-2 plasmid construction and protein purification. P.T. provided the IL-2 plasmid. H.J. and M. Obenaus contributed to the ELISA experiments. H. Lyu and Q.T. contributed to experimental design and manuscript preparation. T.V.L. supervised the EAE experiments and contributed to manuscript revision.

**Competing interests** K.C.G. and Q.S. have filed a provisional patent application on IL-2–TGFβ co-agonists (serial no. 63/924,993). The other authors declare no competing interests.

**Additional information**
**Correspondence and requests for materials** should be addressed to K. Christopher Garcia.

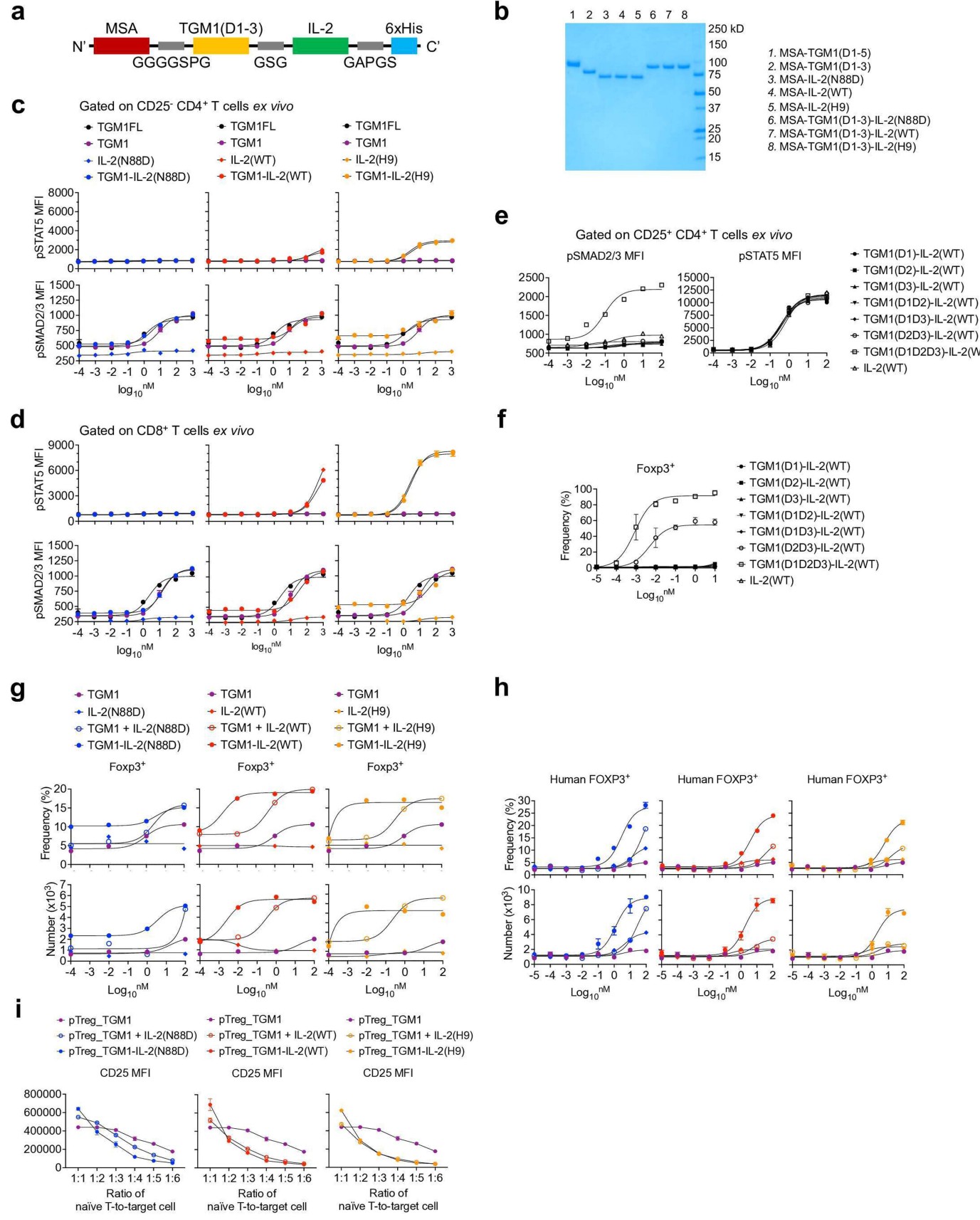

**Extended Data Fig. 1** | See next page for caption.

**Extended Data Fig. 1 | Design and functional characterization of MSA-TGM1-IL-2 fusion proteins. a**, Plasmid construct encoding the MSA-TGM1-IL-2 fusion protein. **b**, SDS-PAGE gel images of the indicated proteins. For gel source data, see Supplementary Fig. 1. **c**, Ex vivo dose-response curves of pSTAT5 and pSMAD2/3 in primary CD25⁻CD4⁺ T cells (n = 2 biological replicates). **d**, Ex vivo dose-response curves of pSTAT5 and pSMAD2/3 in primary CD8⁺ T cells (n = 2 biological replicates). **e**, Ex vivo dose-response curves of pSTAT5 and pSMAD2/3 in primary CD25⁺CD4⁺ T cells (n = 1). **f**, In vitro dose–response curves for Foxp3⁺ cell frequency in cultured mouse naïve CD4⁺ T cells (n = 2 biological replicates). **g**, In vitro dose–response curves for Foxp3⁺ cell frequency and number in cultured mouse CD44⁺Foxp3-GFP⁻ CD4⁺ T cells (n = 1). **h**, In vitro dose–response curves for FOXP3⁺ cell frequency and number in cultured human PBMC CD4⁺ T cells (n = 2 biological replicates). **i**, CD25 expression on naïve T cells co-cultured in vitro with the indicated cells (n = 2 biological replicates). Data are presented as mean ± s.e.m. The data in **b**–**i** are representative of two or three independent experiments.

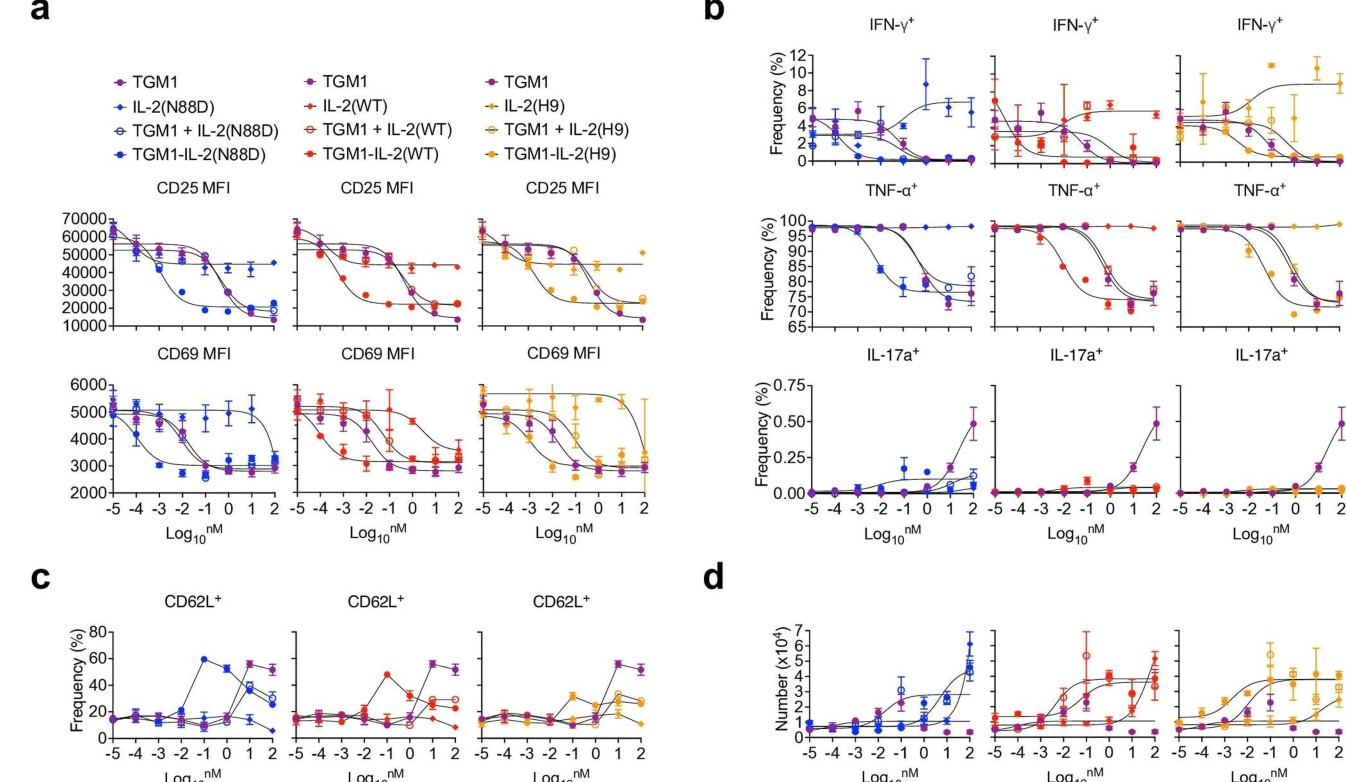

**a**

**b**

**c**

**d**

**Extended Data Fig. 2 | Regulation of CD4⁺ T cell activation, differentiation, and expansion by IL-2-TGF-β co-agonists in vitro. a-c**, In vitro dose–response curves for CD25 and CD69 expression (**a**) and the frequencies of IFN-γ⁺, TNF-α⁺, IL-17a⁺ (**b**), and CD62L⁺ cells (**c**) on cultured mouse naïve CD4⁺ T cells (n = 2 biological replicates). **d**, In vitro dose–response curves for cell number in cultured mouse naïve CD4⁺ T cells (n = 2 biological replicates). Data are presented as mean ± s.e.m. The data in **a**–**d** are representative of two or three independent experiments.

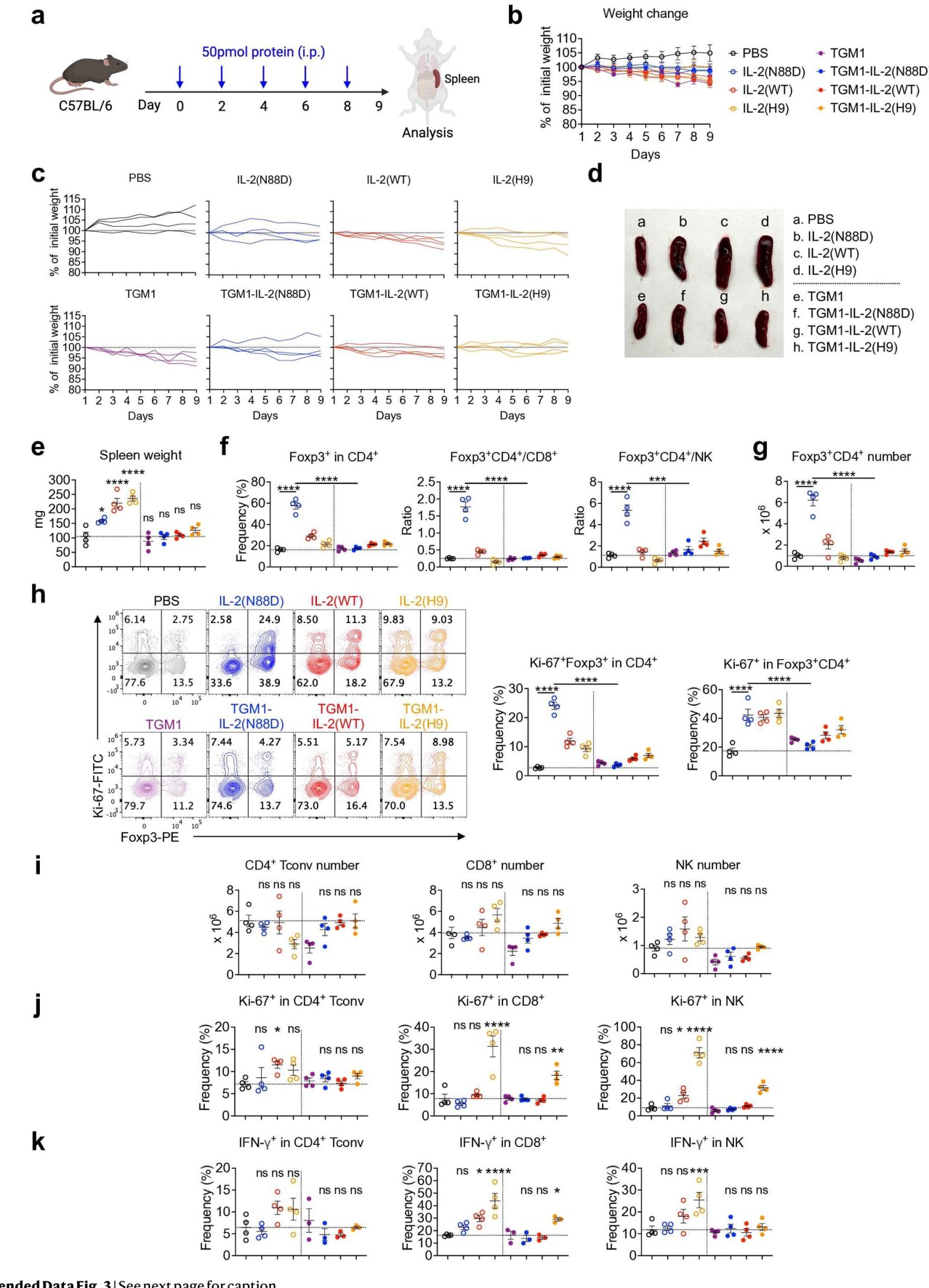

**Extended Data Fig. 3** | See next page for caption.

**Extended Data Fig. 3 | Effect of IL-2-TGF-β co-agonists on IL-2R⁺ immune cells in vivo under steady-state conditions. a-k**, Effect of TGM1–IL-2 on splenic immune cells at steady state (n = 4 mice/group). **a**, Schematic illustration of in vivo protein administration. Created in BioRender; Sun, Q. https://BioRender. com/su1gufm (2026). **b, c**, Mean (**b**) and individual (**c**) body weight change curves of indicated mice. **d, e**, Representative spleen (SPL) images (**d**) and quantification of spleen weight (**e**) in the indicated mice. **f**, Quantification of Foxp3⁺ cell frequencies among splenic CD4⁺ T cells and their ratios to CD8⁺ T and NK cells. **g**, Quantification of Foxp3⁺ CD4⁺ T cell numbers in the spleens.

**h**, Representative flow cytometry plots and quantification of Ki-67⁺ Foxp3⁺ cell frequencies among the indicated cell populations. **i**, Quantification of the numbers of the indicated cell populations in the spleens. **j, k**, Quantification of Ki-67⁺ (**j**) and IFN-γ⁺ cells (**k**) among the indicated splenic cell populations. Data are presented as mean ± s.e.m. The data in **a**–**k** are representative of two independent experiments. The statistics were obtained by one-way ANOVA coupled with Tukey's multiple-comparisons test (**e-h**) or Dunnett's multiple-comparisons test (**i-k**).

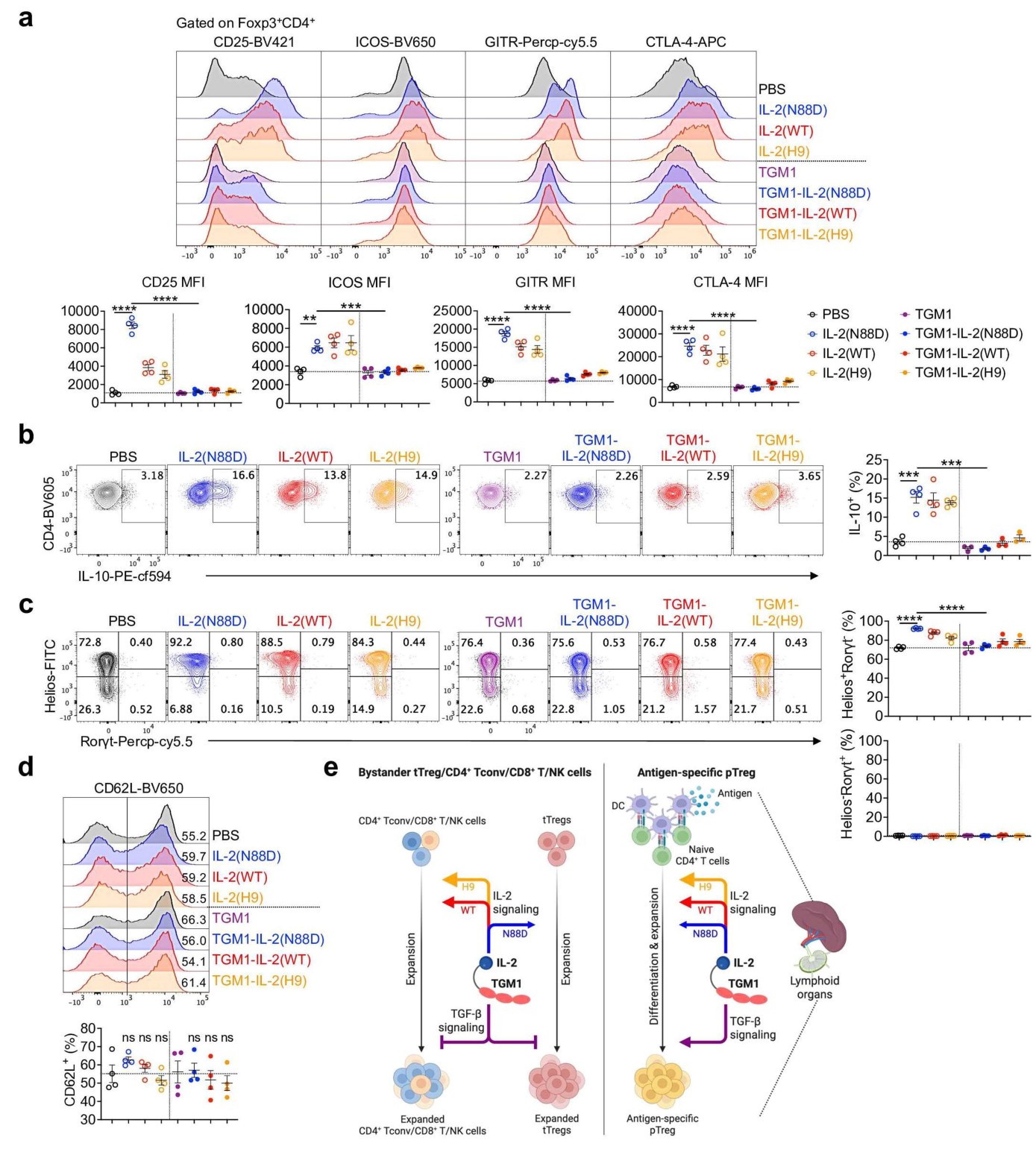

**Extended Data Fig. 4 | Effect of IL-2-TGF-β co-agonists on Tregs in vivo under steady-state conditions. a–d**, Representative flow cytometry plots and quantification of the indicated molecule expression in splenic Foxp3+ CD4+ T cells (n = 4 mice/group). **e**, Schematic illustration of the distinct effects of TGM1–IL-2 variants on IL-2R+ immune-cell homeostatic expansion and antigen-specific pTreg induction. Created in BioRender; Sun, Q. https://BioRender.com/3lqp71g (2026). Data are presented as mean ± s.e.m. The data in **a**–**d** are representative of two independent experiments. The statistics were obtained by one-way ANOVA coupled with Tukey's multiple-comparisons test (**a**–**c**) or one-way ANOVA coupled with Dunnett's multiple-comparisons test (**d**).

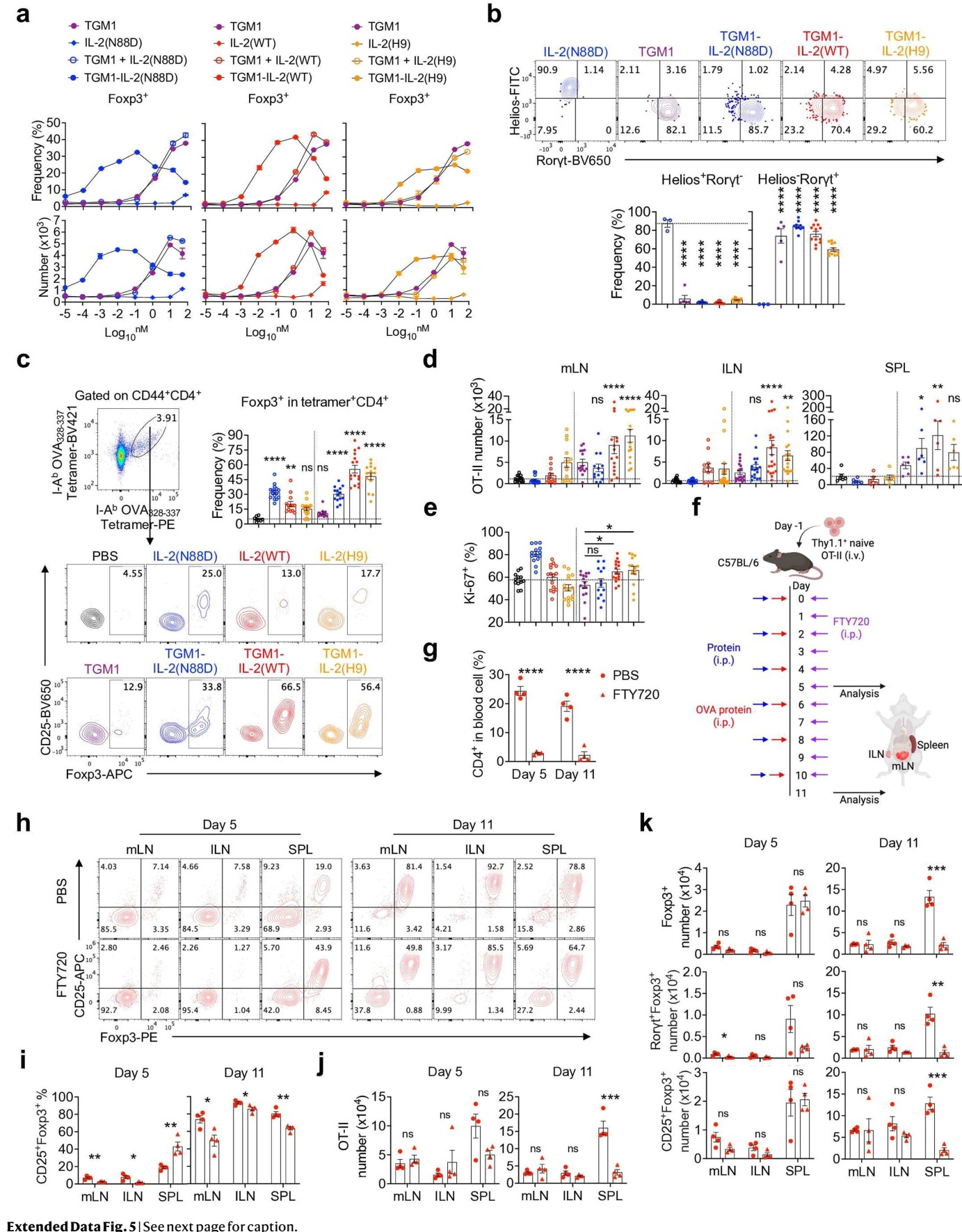

**Extended Data Fig. 5** | See next page for caption.

**Extended Data Fig. 5 | Effect of IL-2-TGF-β co-agonists on OT-II pTreg induction in OVA-immunized mice. a**, In vitro dose–response curves for Foxp3+ cell frequency and number in cultured mouse naïve OT-II cells (n = 2 biological replicates). **b**–**e**, Phenotypic analysis of OT-II cells, pTreg OT-II cells, and endogenous CD4+ T cells (n = 16 samples/group). **b**, Representative flow cytometry plots and quantification of Helios+ and Rorγt+ cell frequencies among Foxp3+ OT-II cells in LNs. **c**, Representative flow cytometry plots and quantification of Foxp3+ cell frequencies among OVA-tetramer+ CD4+ T cells in LNs. **d**, Quantification of the OT-II cell numbers in the indicated lymphoid organs. **e**, Representative flow cytometry plots and quantification of Ki-67+ cell frequencies among OT-II cells in LNs. **f**–**k**, OT-II pTreg cell induction in mice treated with FTY720 (n = 4 mice/group). **f**, Schematic illustration created in BioRender; Sun, Q. https://BioRender.com/essbx6h (2026). **g**, Quantification of CD4+ T cell percentages among total live cells in blood. **h, i**, Representative flow cytometry plots (**h**) and quantification (**i**) of CD25+Foxp3+ OT-II cells in the indicated lymphoid organs. **j, k**, Quantification of total (**j**) and indicated (**k**) OT-II cell numbers in the indicated lymphoid organs. Data are presented as mean ± s.e.m. The data in **a**–**k** are representative of two independent experiments. The statistics were obtained by one-way ANOVA coupled with Dunnett's multiple-comparisons test (**b**–**d**), one-way ANOVA coupled with Tukey's multiple comparisons test (**e**), or unpaired Welch's t-test (two-tailed) (**g** and **i**–**k**).

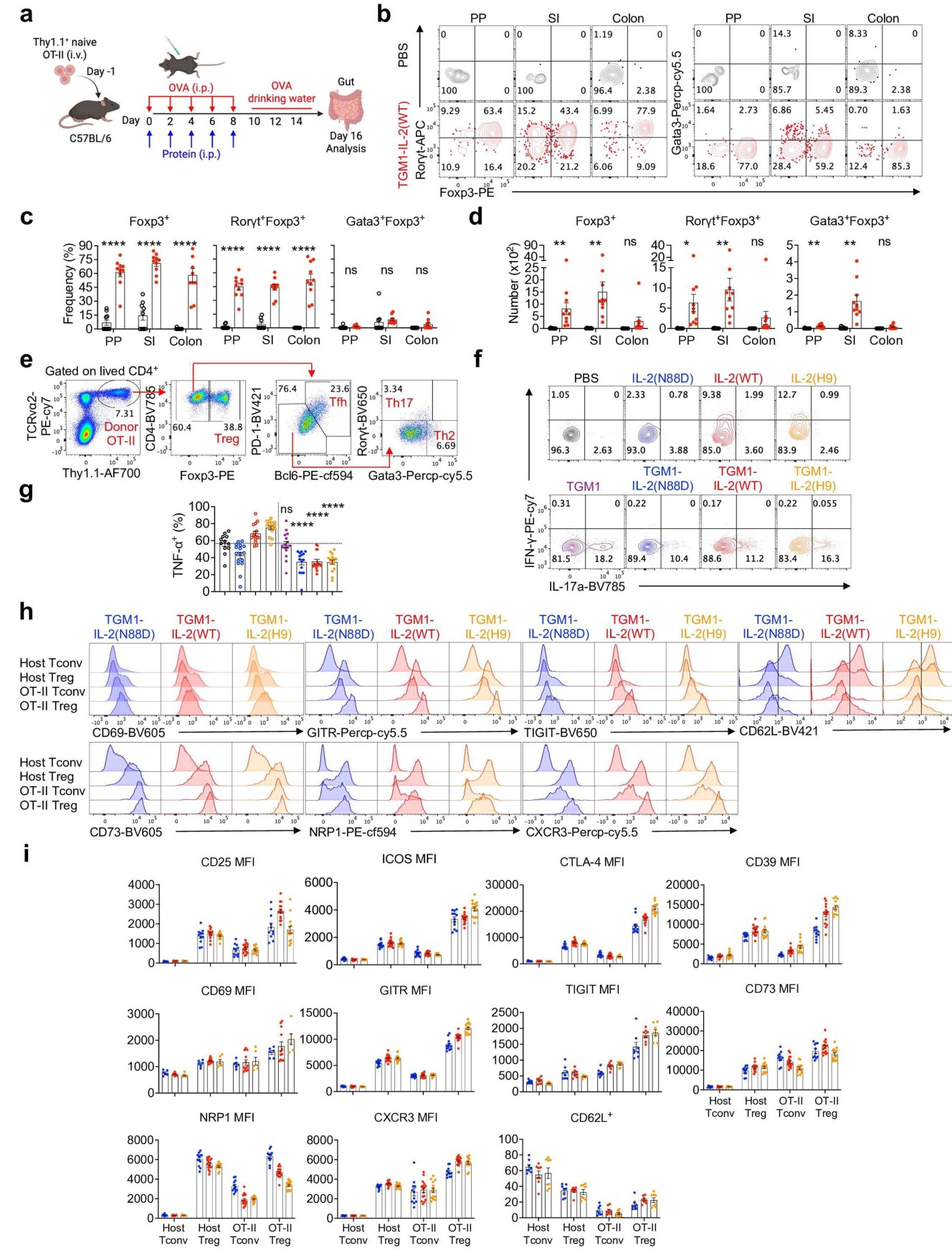

**Extended Data Fig. 6** | See next page for caption.

**Extended Data Fig. 6 | Phenotypic characterization of OT-II pTregs in peripheral lymphoid organs and the gut. a**–**d**, OT-II cell phenotypes after gut migration (n = 10 mice/group). **a**, Schematic illustration of OVA administration in the drinking water. Created in BioRender; Sun, Q. https://BioRender.com/0njaeg9 (2026). **b**–**d**, Representative flow cytometry plots (**b**) and quantification of the frequencies (**c**) and numbers (**d**) of the indicated OT-II cell subsets in the indicated tissues. **e**–**i**, Phenotypic analysis of total and pTreg OT-II cells (n = 16 samples/group). **e**, Gating strategy for identifying distinct subsets among donor OT-II cells. **f**, Representative flow cytometry plots showing IFN-γ and IL-17a production by donor OT-II cells in LNs. **g**, Quantification of TNF-α⁺ OT-II cell frequencies in LNs. **h, i**, Representative flow cytometry plots and quantification showing expression of the indicated molecules in the indicated CD4⁺ T cell subsets from LNs. Data are presented as mean ± s.e.m. The data in **a**–**i** are representative of two independent experiments. The statistics were obtained by unpaired Welch's t-test (two-tailed) (**c** and **d**) or one-way ANOVA coupled with Tukey's multiple-comparisons test (**g**).

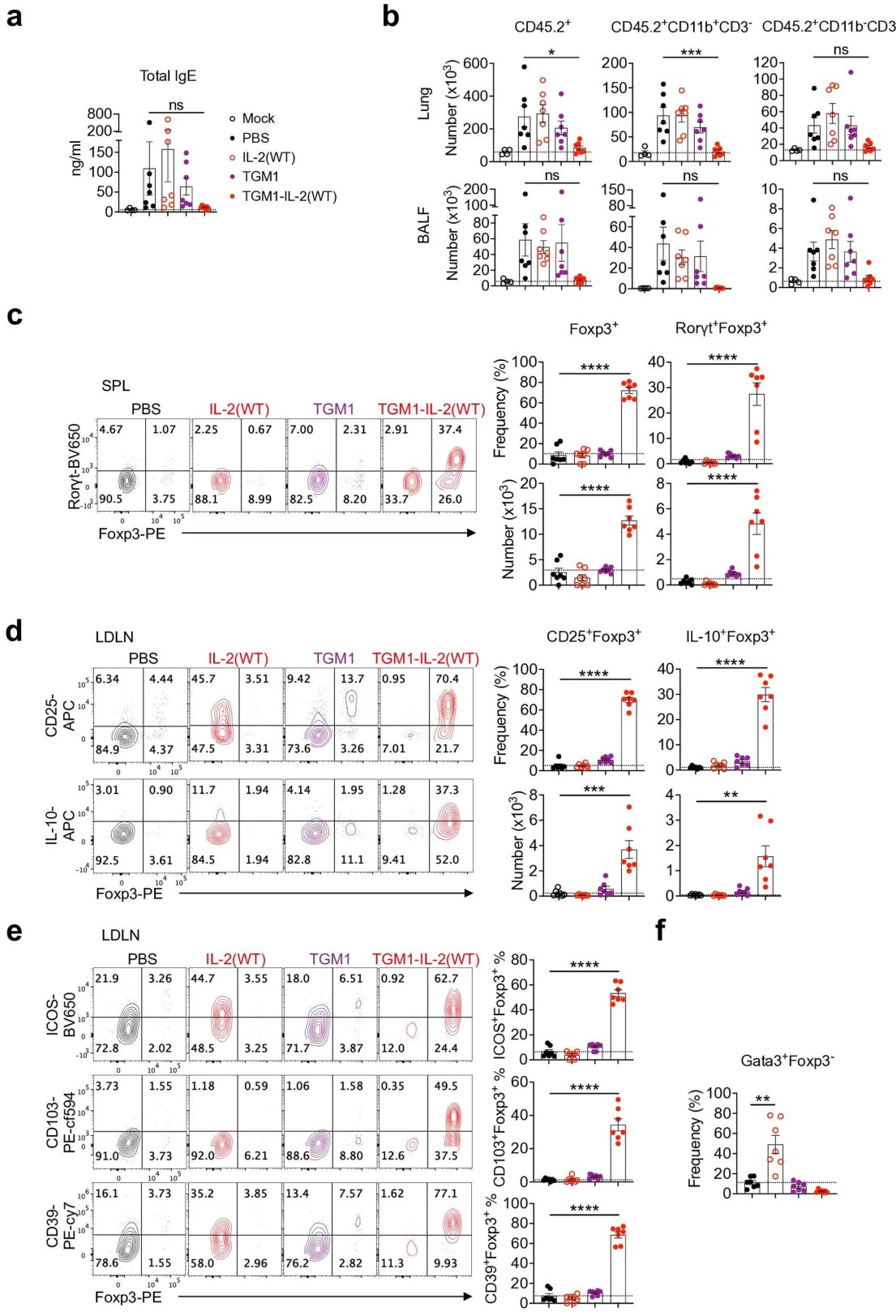

**Extended Data Fig. 7 | IL-2-TGF-β co-agonists establish immune tolerance to suppress OVA-induced airway inflammation. a–f,** Therapeutic effect of TGM1–IL-2 in establishing tolerance to suppress OVA-induced airway inflammation (Mock, n = 4 mice; treatment groups, n = 7 mice/group). **a,** Quantification of serum toal IgE levels. **b,** Quantification of the numbers of the indicated cell populations in the lung and BALF. **c,** Representative flow cytometry plots and quantification of Foxp3+ and Rorγt+Foxp3+ OT-II cell frequencies and numbers in spleen. **d,** Representative flow cytometry plots and quantification of

CD25+Foxp3+ and IL-10+Foxp3+ OT-II cell frequencies and numbers in LDLNs. **e,** Representative flow cytometry plots and quantification of ICOS+Foxp3+, CD103+Foxp3+, and CD39+Foxp3+ OT-II cell frequencies in LDLNs. **f,** Quantification of Gata3+Foxp3- OT-II cell frequencies in LDLNs. Data are presented as mean ± s.e.m. The data in **a–f** are representative of two independent experiments. The statistics were obtained by one-way ANOVA coupled with Dunnett's multiple-comparisons test (**a–f**).

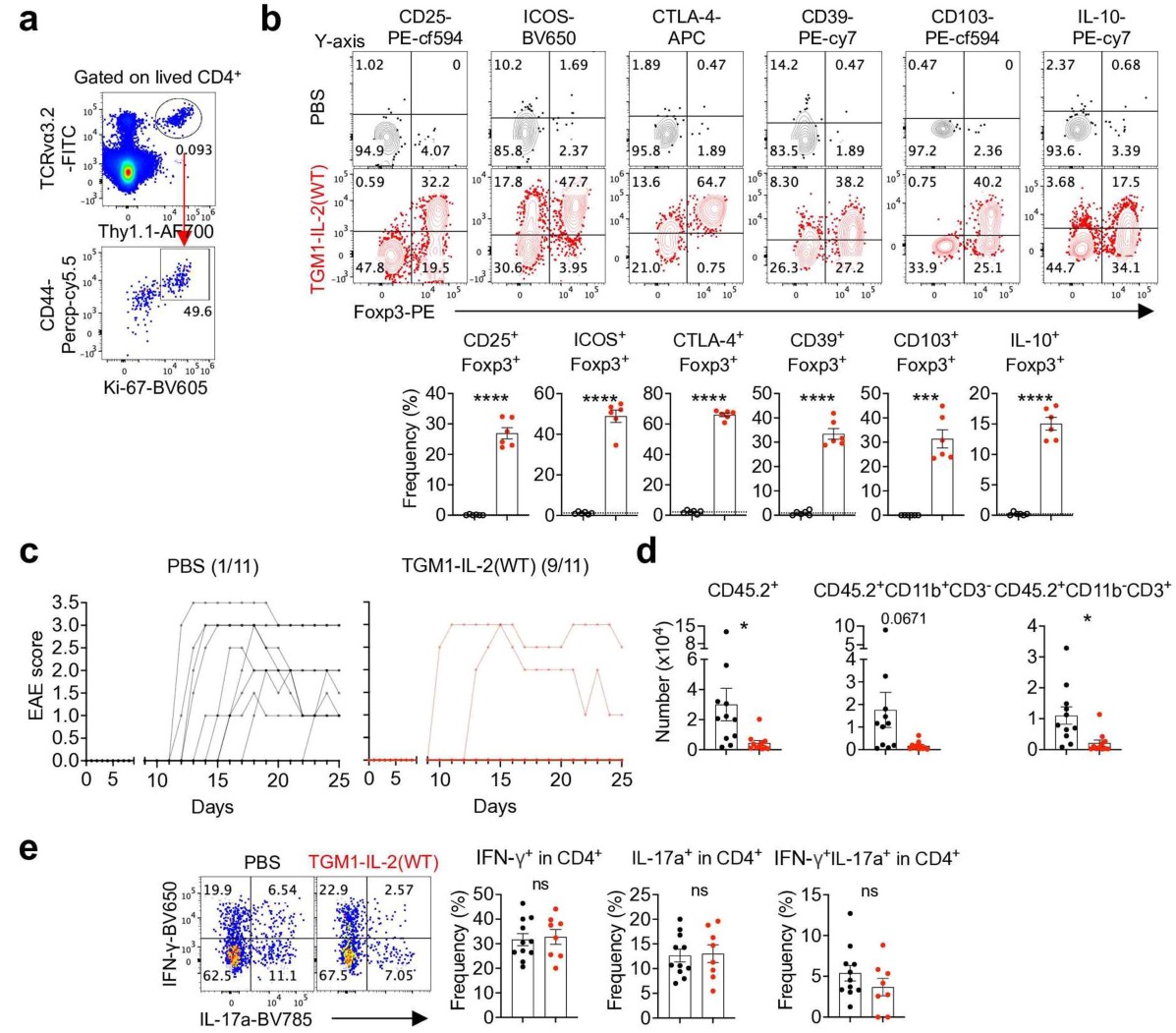

**Extended Data Fig. 8 | IL-2–TGF-β co-agonists establish immune tolerance to suppress MOG$_{35-55}$-induced EAE. a**, Gating strategy for identifying MOG$_{35-55}$-reactive 2D2 cells. **b**, Representative flow cytometry plots and quantification of expression of the indicated markers in splenic 2D2 cells (n = 6 samples/group). **c**–**e**, Therapeutic effect of TGM1–IL-2 in establishing tolerance to suppress MOG$_{35-55}$-induced EAE (n = 11 mice/group). **c**, Individual EAE scores for mice in the indicated groups. **d**, Quantification of the numbers of the indicated cell populations in the spinal cord. **e**, Representative flow cytometry plots and quantification of IFN-γ$^+$ and IL-17a$^+$ CD4$^+$ T-cell frequencies in the spinal cord. Data are presented as mean ± s.e.m. The data in **b**–**e** are representative of two independent experiments. The statistics were obtained by unpaired Welch's t-test (two-tailed) (**b,d** and **e**).

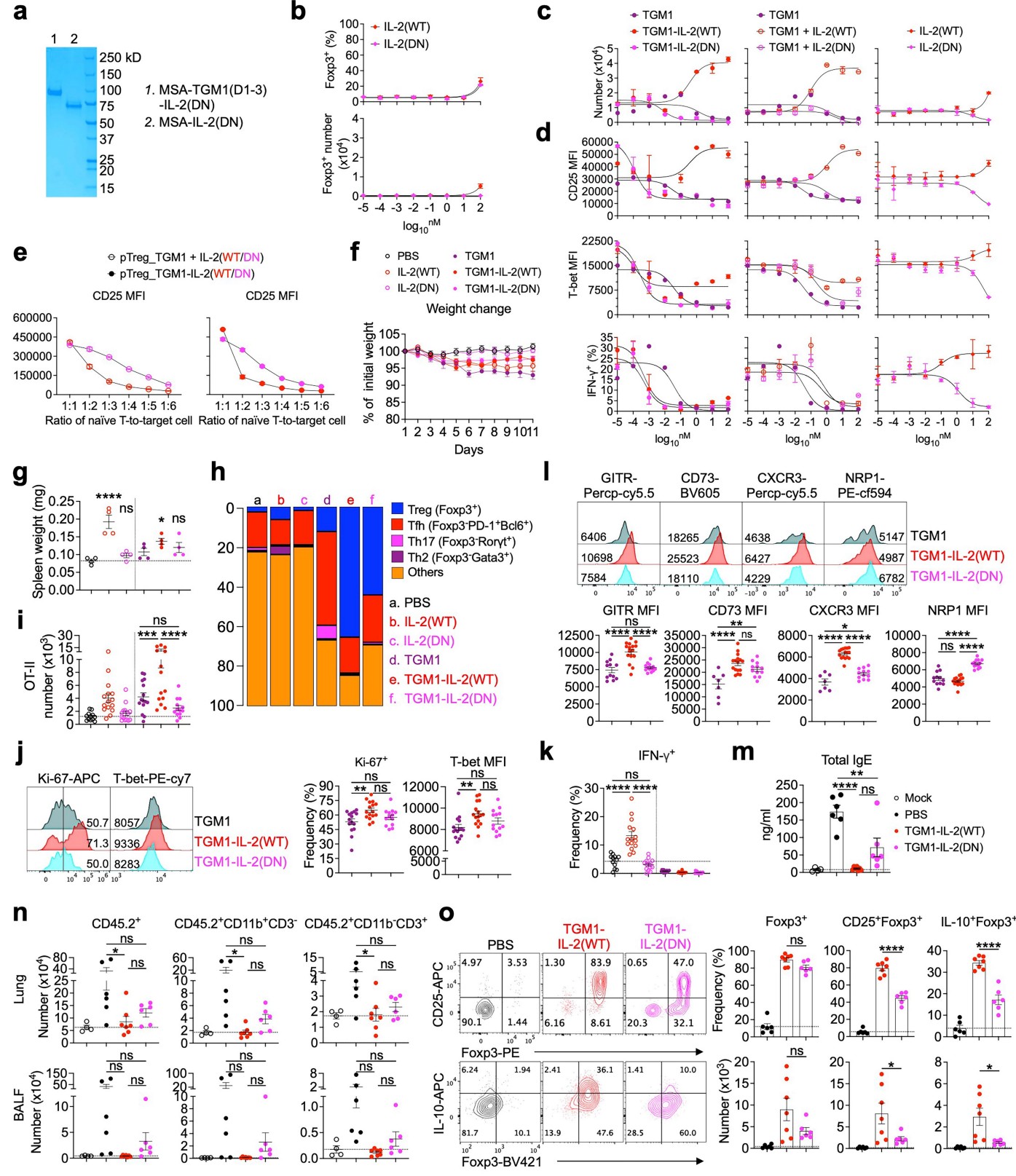

**Extended Data Fig. 9** | See next page for caption.

**Extended Data Fig. 9 | IL-2-TGF-β co-agonists activate IL-2 signalling to enable optimal pTreg development and immune tolerance. a**, SDS-PAGE gel images of the indicated proteins. For gel source data, see Supplementary Fig. 1. **b**, In vitro dose–response curves for Foxp3$^+$ cell frequency and number in cultured mouse naïve CD4$^+$ T cells (n = 2 biological replicates). **c**, In vitro dose–response curves for cell number in cultured mouse naïve CD4$^+$ T cells (n = 2 biological replicates). **d**, in vitro dose–response curves for expression of the indicated molecules in cultured mouse naïve CD4$^+$ T cells (n = 2 biological replicates). **e**, CD25 expression on naïve T cells co-cultured in vitro with the indicated cells (n = 3 biological replicates). **f**, Quantification of mean body weight changes (n = 4 mice/group). **g**, Quantification of spleen weights (n = 4 mice/group). **h–l**, Phenotypic analysis of total and pTreg OT-II cells in the LNs (n = 16 samples/group). **h**, Histogram showing the distribution of distinct subsets among donor OT-II cells. **i**, Quantification of the OT-II cell numbers.

**j**, Representative flow cytometry plots and quantification of Ki-67 and T-bet expression in OT-II cells. **k**, Quantification of IFN-γ$^+$ OT-II cell frequencies. **l**, Representative flow cytometry plots and quantification showing expression of the indicated molecules on OT-II pTregs. **m–o**, Therapeutic effect of TGM1–IL-2 in OVA-induced airway inflammation (Mock, n = 4 mice; PBS, n = 6 mice; TGM1–IL-2WT, n = 7 mice; TGM1–IL-2DN, n = 6 mice). **m**, Quantification of serum toal IgE levels. **n**, Quantification of the numbers of the indicated cell populations in the lung and BALF. **o**, Representative flow cytometry plots and quantification of CD25$^+$Foxp3$^+$ and IL-10$^+$Foxp3$^+$ OT-II cell frequencies and numbers in LDLNs. Data are presented as mean ± s.e.m. The data in **a–o** are representative of two independent experiments. The statistics were obtained by one-way ANOVA coupled with Dunnett's multiple-comparisons test (**g**) or Tukey's multiple-comparisons test (**i–o**).

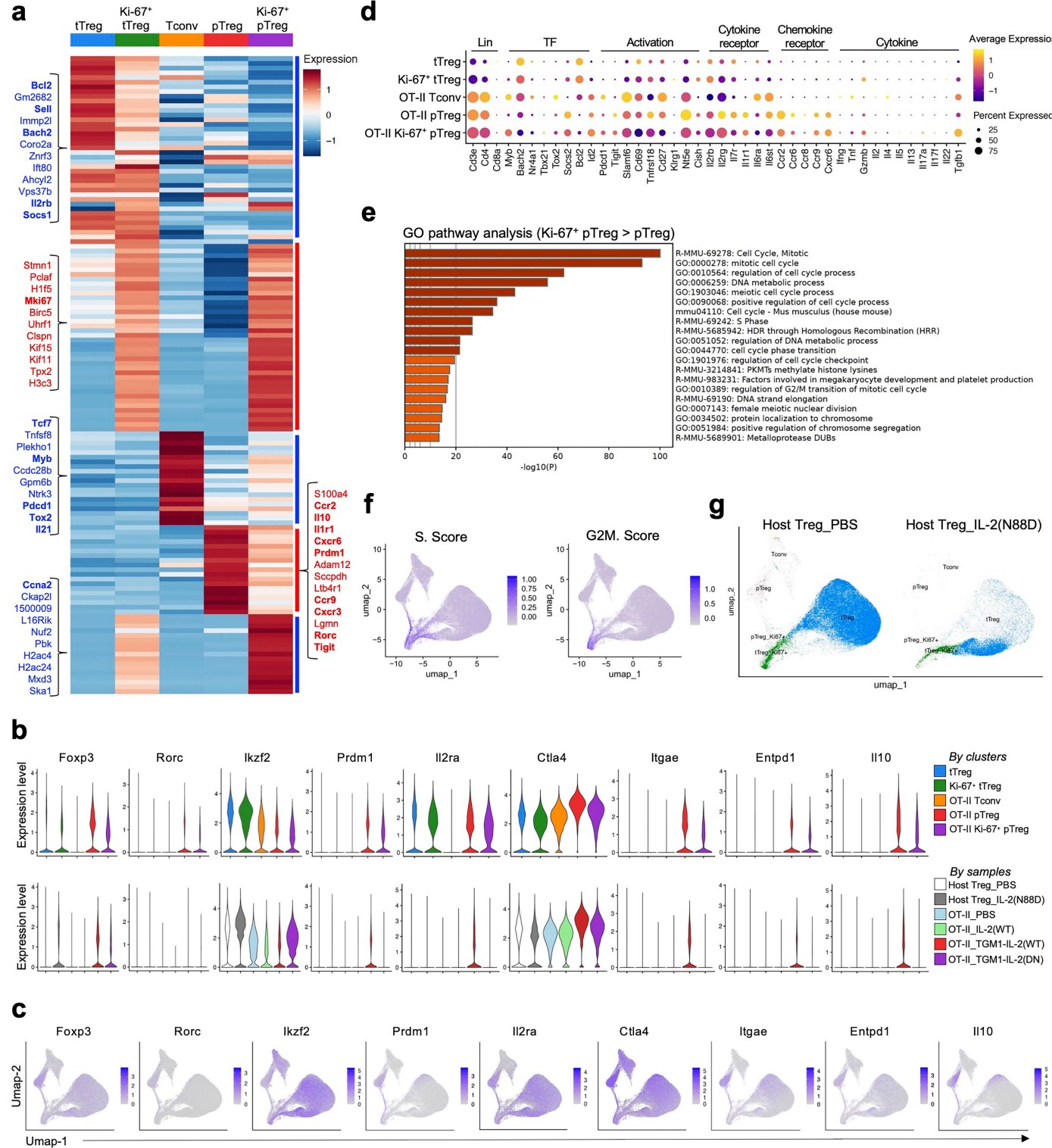

**Extended Data Fig. 10 | Gene expression signatures defining distinct cell clusters revealed by scRNA-seq analysis. a**, Heatmap showing top 20 DEGs across distinct cell clusters. **b–d**, Violin (**b**), UMAP (**c**) and bubble plots (**d**) showing expression of selected genes across the indicated samples and clusters. **e**, Histogram showing the top 20 Gene Ontology (GO) pathways enriched among upregulated DEGs between the Ki-67+ pTreg cell cluster and the pTreg cell cluster, based on Metascape analysis (https://metascape.org). **f**, UMAP plots showing the indicated signature scores across clusters. **g**, UMAP plots showing the distribution of endogenous Foxp3-GFP+ CD4+ T cells from the indicated groups across distinct cell clusters.

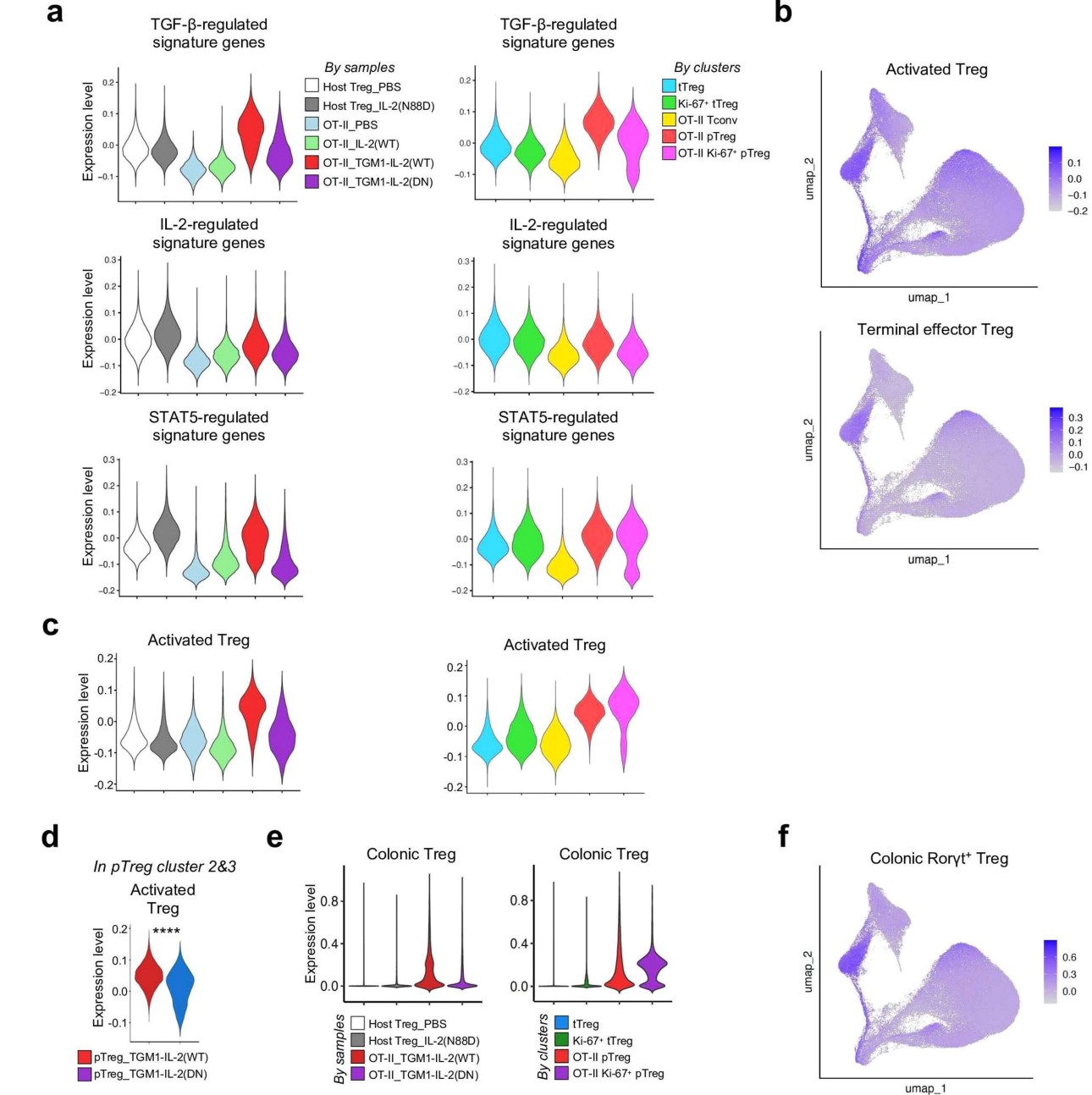

**Extended Data Fig. 11 | pTregs induced by IL-2-TGF-β co-agonists in vivo exhibit effector and colonic Treg features. a**, Violin plots showing expression levels of TGF-β– (GEO, GSE264451), IL-2– (GEO, GSE162928), and STAT5-upregulated (GEO, GSE84553) gene sets across the indicated samples and clusters. **b**–**d**, UMAP (**b**) and violin plots (**c** and **d**) showing expression levels of activated Treg (GEO, GSE149674) and terminal effector Treg signature gene sets across the indicated samples and clusters. **e**, Violin plots showing expression levels of the colonic Treg signature gene set (ArrayExpress, E-MTAB-7311) across the indicated samples and clusters. **f**, UMAP plot showing expression levels of the colonic Roryt⁺ Treg signature gene set across clusters. The statistics were obtained by unpaired Welch's t-test (two-tailed) (**d**).

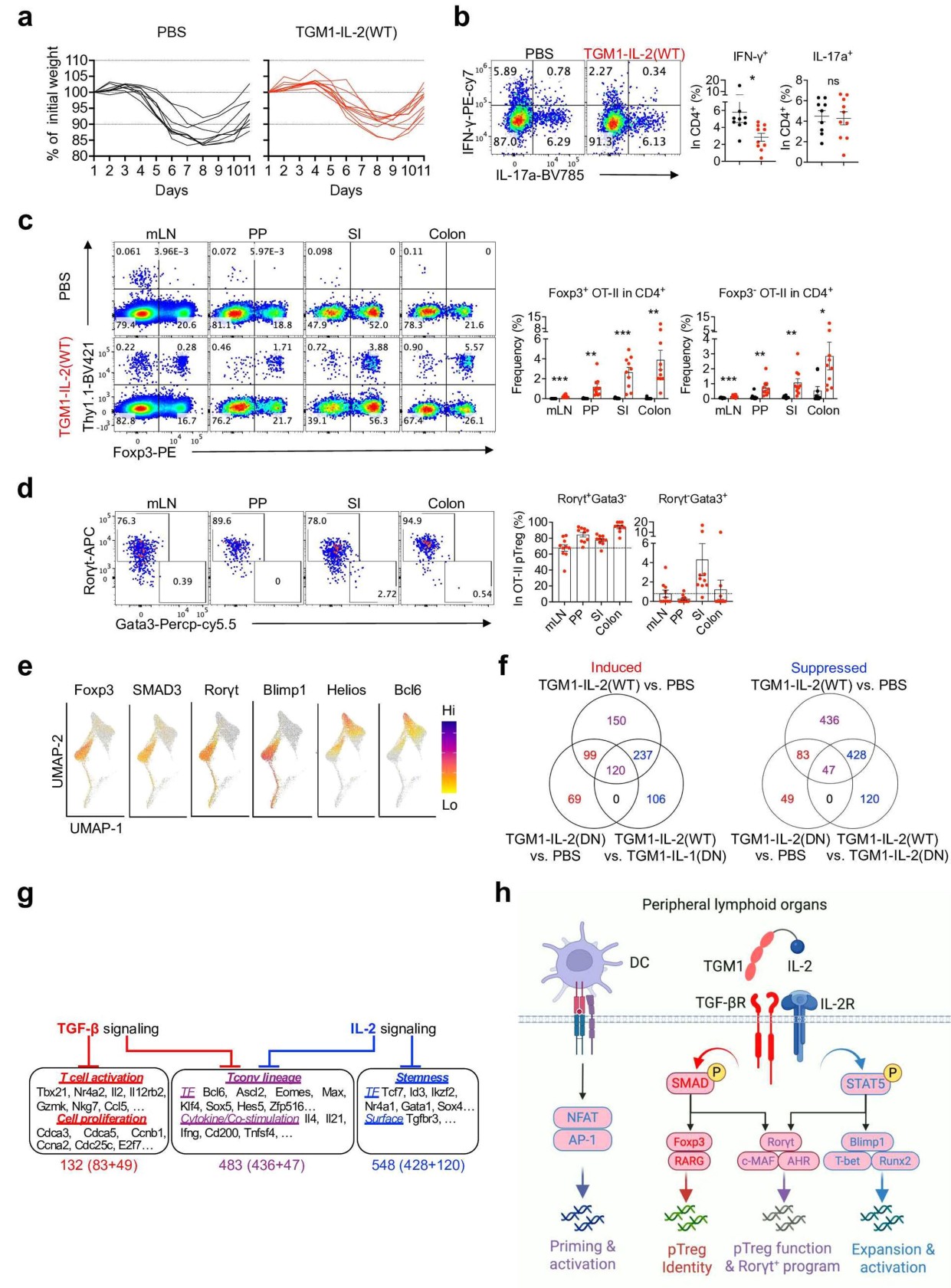

**Extended Data Fig. 12 | Gene regulation and functional characterization of pTregs induced by IL-2–TGF-β co-agonists. a–d**, Therapeutic effect of TGM1–IL-2 in DSS-induced colitis (PBS, n = 9 mice; TGM1–IL-2, n = 10 mice). **a**, Individual weight changes of mice in the indicated groups. **b**, Representative flow cytometry plots and quantification of IFN-γ⁺ and IL-17a⁺ CD4⁺ T-cell frequencies in the colon. **c**, Representative flow cytometry plots and quantification of Foxp3⁺ and Foxp3⁻ Thy1.1⁺ OT-II cell frequencies among total CD4⁺ T cells in the indicated tissues. **d**, Representative flow cytometry plots and quantification of Rorγt⁺ and Gata3⁺ cell frequencies among Foxp3⁺ OT-II cells in the indicated tissues. **e**, UMAP plots showing the indicated transcription factor activity scores across clusters. **f**, Venn diagram illustrating the overlap of up- and down-regulated DEGs between the indicated groups. DEGs were filtered using an adjusted p value < 0.0001 and log₂ fold change > 1. **g**, Gene regulatory networks and signature genes identified by scRNA-seq analysis. The two numbers in parentheses represent the specific values from the Venn diagram shown in Extended Data Fig. 12f (right). **h**, Schematic illustration of antigen-specific pTreg differentiation in vivo driven by TCR stimulation, TGM1–IL-2, and key transcription factors. Created in BioRender; Sun, Q. https://BioRender.com/2rbbpii (2026). Data are presented as mean ± s.e.m. The data in **a–d** are representative of two independent experiments. The statistics were obtained by unpaired Welch's t-test (two-tailed) (**b** and **c**).

# Reporting Summary

## Statistics

For all statistical analyses, confirm that the following items are present in the figure legend, table legend, main text, or Methods section.

| n/a | Confirmed | |
|---|---|---|
| ☐ | ☒ | The exact sample size (*n*) for each experimental group/condition, given as a discrete number and unit of measurement |
| ☐ | ☒ | A statement on whether measurements were taken from distinct samples or whether the same sample was measured repeatedly |
| ☐ | ☒ | The statistical test(s) used AND whether they are one- or two-sided *Only common tests should be described solely by name; describe more complex techniques in the Methods section.* |
| ☒ | ☐ | A description of all covariates tested |
| ☐ | ☒ | A description of any assumptions or corrections, such as tests of normality and adjustment for multiple comparisons |
| ☐ | ☒ | A full description of the statistical parameters including central tendency (e.g. means) or other basic estimates (e.g. regression coefficient) AND variation (e.g. standard deviation) or associated estimates of uncertainty (e.g. confidence intervals) |
| ☐ | ☒ | For null hypothesis testing, the test statistic (e.g. *F*, *t*, *r*) with confidence intervals, effect sizes, degrees of freedom and *P* value noted *Give P values as exact values whenever suitable.* |
| ☒ | ☐ | For Bayesian analysis, information on the choice of priors and Markov chain Monte Carlo settings |
| ☒ | ☐ | For hierarchical and complex designs, identification of the appropriate level for tests and full reporting of outcomes |
| ☒ | ☐ | Estimates of effect sizes (e.g. Cohen's *d*, Pearson's *r*), indicating how they were calculated |

*Our web collection on statistics for biologists contains articles on many of the points above.*

## Software and code

Policy information about availability of computer code

| Data collection | Flow cytometry data were acquired using a CytoFLEX (Beckman Coulter). scRNA-seq libraries were sequenced on an Illumina NovaSeq X. Tissue sections were imaged using a Leica DM2000 microscope. |
|---|---|
| Data analysis | The FASTQ files were processed using Cell Ranger v9.0.0. The gene expression matrix was processed and analyzed using Seurat (version 5.1.0). For quality control, we excluded cells that contained fewer than 500 read counts for genes or fewer than 200 genes detected (minimal cutoff), or more than 50,000 read counts for genes or more than 6500 genes detected (maximum cutoff). We also excluded cells in which more than 20% of transcripts were derived from mitochondrial RNA. These QC filters left 190,728 cells. Graph-based unsupervised clustering was employed to identify clusters representing minor contaminant cells other than T cells, such as neurons (expressing Cntn1, Dscam, and Pde7b) and B cells (expressing Igkc, Ms4a1, and Cd79a). These minor clusters were excluded from subsequent analyses, leaving 180,038 cells. Uniform Manifold Approximation and Projection (UMAP) embedding was computed with 10 principal components, with n.neighbors being 20 and min.dist being 0.1. Differential expression (DE) analysis between groups was performed using Wilcoxon's rank sum test implemented in Seurat's FindMarkers function. Gene set enrichment analysis (GSEA) was performed using the log2FC ranking of DE genes using fgsea. Human Hallmark gene sets were retrieved from MSigDB. The signature scores were calculated using Seurat's AddModuleScore function. To calculate Cell Cycle score, Seurat's CellCycleScoring function was used. Built-in human genesets for the S and G2M phases in Seurat were converted into mouse homologues and used to calculate S and G2M scores. Transcription factor activity inference was conducted using pySCENIC with default parameter settings. To identify DEGs from public bulk RNA-seq datasets for computing gene-set signature scores, bulk RNA-seq FASTQ files were aligned to the GENCODE VM25 (mm10) reference genome using Rsubread, and gene expression was quantified with featureCounts. DE analysis was performed using DESeq2. Pathway analysis was performed using Metascape. The R code used to analyze the scRNA-seq data has been deposited on Zenodo (10.5281/zenodo.18166788). Analysis details are provided in the Methods. |

For manuscripts utilizing custom algorithms or software that are central to the research but not yet described in published literature, software must be made available to editors and reviewers. We strongly encourage code deposition in a community repository (e.g. GitHub). See the Nature Portfolio guidelines for submitting code & software for further information.

## Data

Policy information about availability of data

All manuscripts must include a data availability statement. This statement should provide the following information, where applicable:
- Accession codes, unique identifiers, or web links for publicly available datasets
- A description of any restrictions on data availability
- For clinical datasets or third party data, please ensure that the statement adheres to our policy

> The raw and processed scRNA-seq data have been deposited in the Gene Expression Omnibus (GEO) under accession GSE315102 (aligned to the mm10 mouse reference genome). The R code used to analyze the scRNA-seq data has been deposited on Zenodo (10.5281/zenodo.18166788). Analysis details are provided in the Methods.

## Research involving human participants, their data, or biological material

Policy information about studies with human participants or human data. See also policy information about sex, gender (identity/presentation), and sexual orientation and race, ethnicity and racism.

| | |
|---|---|
| Reporting on sex and gender | Gender information was not collected. |
| Reporting on race, ethnicity, or other socially relevant groupings | N/A |
| Population characteristics | Human T cells were isolated from buffy coats obtained from anonymous healthy donors (male and female; sex was not recorded) purchased from the Stanford Blood Center. |
| Recruitment | Anonymous healthy donors were recruited by the Stanford Blood Center. |
| Ethics oversight | Ethical approval pertaining to donors was obtained by the Stanford Blood Center. |

Note that full information on the approval of the study protocol must also be provided in the manuscript.

# Field-specific reporting

Please select the one below that is the best fit for your research. If you are not sure, read the appropriate sections before making your selection.

☒ Life sciences ☐ Behavioural & social sciences ☐ Ecological, evolutionary & environmental sciences

For a reference copy of the document with all sections, see nature.com/documents/nr-reporting-summary-flat.pdf

# Life sciences study design

All studies must disclose on these points even when the disclosure is negative.

| | |
|---|---|
| Sample size | Group sizes for in vivo and in vitro validation experiments were determined based on prior knowledge of expected variability, with a minimum of three mice per group. |
| Data exclusions | Rout outlier tests were run with default parameters (Q = 1%) in Prism on all mouse experimental data due to inherent variability within the model system. |
| Replication | All presented results were repeatable. Replicates were used in all experiments as noted in figure captions or methods. |
| Randomization | Age and sex-matched animals were used for each experiment. Mice were randomized prior to treatment. In the in vitro experiments, samples with same pretreatment conditions were randomly assigned to a treatment group. |
| Blinding | Histological analyses were performed in a fully blinded manner. Blinding was not performed for the remaining experiments due to requirements for cage labeling and staffing constraints. |

# Reporting for specific materials, systems and methods

We require information from authors about some types of materials, experimental systems and methods used in many studies. Here, indicate whether each material, system or method listed is relevant to your study. If you are not sure if a list item applies to your research, read the appropriate section before selecting a response.

## Materials & experimental systems

| n/a | Involved in the study |
|---|---|
| ☐ | ☒ Antibodies |
| ☐ | ☒ Eukaryotic cell lines |
| ☒ | ☐ Palaeontology and archaeology |
| ☐ | ☒ Animals and other organisms |
| ☒ | ☐ Clinical data |
| ☒ | ☐ Dual use research of concern |
| ☒ | ☐ Plants |

## Methods

| n/a | Involved in the study |
|---|---|
| ☒ | ☐ ChIP-seq |
| ☐ | ☒ Flow cytometry |
| ☒ | ☐ MRI-based neuroimaging |

# Antibodies

| | |
|---|---|
| Antibodies used | The following antibodies were purchased from BioLegend: mouse CD45.2 (109839), CD3 (100206), CD4 (100453, 100430, 100428, 100451), CD8 (100706), NK1.1 (156506), TCRvα2 (127806, 127822), TCRvα3.2 (135404), Thy1.1 (202528, 202522), CTLA4 (106310), CD62L (104453), CD25 (102012, 102022, 102047, 102038), CD44 (103026, 103032), SIGLECF (155534), CD73 (127215), ICOS (313550), CD69 (104530), CD11b (101259), CXCR3 (126514), CD39 (143806), NRP1 (145218), CD11c (117318), CD103 (110910), GITR (126316), CXCR6 (151117), CCR6 (129819), IL-17A (506928), GMCSF (505406), IFNγ (505832, 505826), IL-10 (505034, 505026, 505034), BLIMP1 (150008), Helios (137214), Ki-67 (151212, 652406), TNFα (506346); human CD3 (317324), CD4 (980806) and FOXP3 (320126). The following antibodies were purchased from BD Biosciences: mouse BCL6 (562401), RORγt (564722, 562682, 562683), SMAD2 (pS465/pS467)/SMAD3 (pS423/pS425) (562696) and STAT5 (pY694) (612599). The following antibodies and reagents were purchased from Invitrogen: mouse PD-1 (48-9985-82), FOXP3 (12-5773-82, 17-5773-82, 404-5773-82), T-bet (25-5825-82), c-MAF (53-9855-82), GATA3 (46-9966-42) and Fixable Viability Dye (65-0865-18). PE- and Brilliant Violet 421-labeled I-Ab OVA328-337 tetramers (HAAHAEINEA) were provided by the NIH Tetramer Core Facility. |
| Validation | All commercially available antibodies were validated by the manufacturers. |

# Eukaryotic cell lines

Policy information about cell lines and Sex and Gender in Research

| | |
|---|---|
| Cell line source(s) | The Expi293F™ cells were purchased from Thermo Fisher Scientific. |
| Authentication | None of the cell lines were authenticated in these studies. In all studies, cell lines with low passage number were used. |
| Mycoplasma contamination | All cell lines were confirmed mycoplasma negative. |
| Commonly misidentified lines (See ICLAC register) | No commonly misidentified cell lines were used. |

# Animals and other research organisms

Policy information about studies involving animals; ARRIVE guidelines recommended for reporting animal research, and Sex and Gender in Research

| | |
|---|---|
| Laboratory animals | Six- to eight-week-old female and male C57BL/6J mice (IMSR_JAX:000664), as well as other strains, were purchased from The Jackson Laboratory. OT-II (IMSR_JAX:004194) and Thy1.1 (IMSR_JAX:000406) mice were crossed to generate OT-II Thy1.1 mice. Foxp3-GFP mice (IMSR_JAX:006772) were crossed with OT-II Thy1.1 mice. 2D2 mice (IMSR_JAX:006912) were crossed with Thy1.1 mice. All animals were housed in AAALAC-accredited facilities. |
| Wild animals | No wild animals were involved. |
| Reporting on sex | Female mice were used for EAE experiments. In all other experiments, both male and female mice were used, and no sex-specific phenotypes were observed. |
| Field-collected samples | There were no field-collected samples. |
| Ethics oversight | All experimental mouse procedures were approved by the Stanford University Institutional Animal Care and Use Committee (IACUC; protocol IDs 32279 and 34708) and conducted in accordance with institutional guidelines. |

Note that full information on the approval of the study protocol must also be provided in the manuscript.

## Plants

| Seed stocks | N/A |
|---|---|

| Novel plant genotypes | N/A |
|---|---|

| Authentication | N/A |
|---|---|

# Flow Cytometry

## Plots

Confirm that:

☒ The axis labels state the marker and fluorochrome used (e.g. CD4-FITC).

☒ The axis scales are clearly visible. Include numbers along axes only for bottom left plot of group (a 'group' is an analysis of identical markers).

☒ All plots are contour plots with outliers or pseudocolor plots.

☒ A numerical value for number of cells or percentage (with statistics) is provided.

## Methodology

| Sample preparation | Mouse lymph nodes and spleens were harvested and mechanically dissociated to obtain single-cell suspensions. Red blood cells were lysed using ACK lysis buffer (A10492-01, Gibco), followed by magnetic isolation of CD4$^+$ T cells using the EasySep™ Mouse CD4$^+$ T Cell Isolation Kit (19852, STEMCELL). Naïve CD4$^+$ T cells (CD4$^+$CD44$^-$CD25$^-$Foxp3-GFP$^-$) and activated CD4$^+$ Tconv cells (CD4$^+$CD44$^+$CD25$^-$Foxp3-GFP$^-$) were subsequently sorted using a Sony SH800S Cell Sorter. The purity of the sorted populations was consistently greater than 99%. Human CD4$^+$ T cells were isolated from frozen PBMCs using the EasySep™ Human CD4$^+$ T Cell Isolation Kit (17952, STEMCELL). BALF was collected by flushing the lungs three times with 0.75mL of PBS via a catheter inserted into the trachea. For lymphocyte isolation from the lung, tissues were mechanically dissociated using the plunger of a 1mL syringe and filtered through 70µm cell strainers to obtain single-cell suspensions. For lymphocyte isolation from the lamina propria, Peyer's patches in the small and large intestines were first removed. The intestines were then opened longitudinally, cut into ~2-cm pieces, and incubated in 5 mM EDTA (15575020, Invitrogen) with 1 mM DTT (R0861, Thermo Fisher Scientific) at 37 °C for 30 minutes to remove epithelial cells. Tissues were then minced and digested in DNase I (40µg/mL; Roche) and collagenase D (0.5mg/mL; Roche) at 37°C for 30min with shaking to generate single-cell suspensions, which were filtered through 70µm cell strainers. For lymphocyte isolation from the spinal cord, mice were first perfused, and the collected spinal cords were mechanically dissociated using the plunger of a 1mL syringe and passed through 70µm cell strainers to obtain single-cell suspensions. The resulting cells were subjected to density gradient centrifugation using a 40%/70% Percoll (Cytiva) gradient. Immune cells located at the interface between the two Percoll layers were harvested and processed for flow cytometry analysis. |
|---|---|
| Instrument | CytoFlex (Beckman Coulter) |
| Software | FlowJo (v 10.10.0) |
| Cell population abundance | Sort purity was determined to be 95% by analyzing a post-sort sample. |
| Gating strategy | Donor OT-II cells were identified as TCR Vα2$^+$ Thy1.1$^+$, and donor 2D2 cells as TCR Vα3.2$^+$ Thy1.1$^+$. Eosinophils were gated as CD45$^+$ CD11b$^+$ CD11c$^-$ Siglec-F$^+$. Naive CD4$^+$ T cells were gated from CD4$^+$ CD44$^-$ CD25$^-$ Foxp3-GFP$^-$ cells, and activated CD4$^+$ conventional T cells (Tconv) were gated from CD4$^+$ CD44$^+$ CD25$^-$ Foxp3-GFP$^-$ cells. |

☒ Tick this box to confirm that a figure exemplifying the gating strategy is provided in the Supplementary Information.

