## [Peer Review File · Nature]

Facile Induction of Immune Tolerance by an Interleukin-2-TGF- β Surrogate Agonist

Corresponding Author: Professor K. Christopher Garcia

Version 1:

Reviewer comments:

Referee #1

(Remarks to the Author)

In this study, Sun et al. designed and evaluated an IL-2-TGF- β agonist. Using a series of in vitro and in vivo experiments, the authors demonstrate that this antibody effectively induces iTreg cells in vitro and pTreg cells in vivo. In vivo generated pTreg cells expressed Ror γ t and transcriptionally resembled colonic pTreg cells. In vivo administration ameliorated disease severity in a model of allergic airway inflammation.

Overall, the experiments are performed to a high standard and provide important proof-of-concept in vivo therapeutic studies. However, there are some aspects of the study that should be addressed to strengthen the manuscript.

While the authors nicely demonstrate the efficacy of the antibody to generate pTreg cells in vivo which are associated with protection from allergen-induced airway inflammation, further studies are required to establish the broader therapeutic relevance of this antibody.

The authors show that the pTreg cells resemble colonic pTreg cells therefore one exciting avenue that could extend the scope of the present work would be to determine whether the bispecific antibody can promote colonic pTreg differentiation and restore homeostasis in settings of intestinal inflammation for example following pre-sensitization to oral antigens within the gut (mimicking food allergy) or following onset of colitis.

Does the bispecific antibody bypass the need for other tolerogenic/pTreg-inducing signals or is IL-2 and TGF- β signaling sufficient? Given that pTreg inducing Ror γ t+ APCs have been identified, the authors could determine whether their presence is required for antibody induced pTreg differentiation.

Related to this, the authors state that the role of TGF- β and IL-2 in promoting pTreg differentiation in vivo have not been fully defined but there are a number of recent studies that have established a role for avb8+ Ror γ t+ APCs in microbiota and food-specific pTreg differentiation (Kedmi et al. Nature 2022, Rudnitsky et al. Nature 2025, Sun et al. JEM 2025, Akagbosu et al. Nature 2022, Cabric et al. 2025). These studies are referenced but should be discussed in relation to avb8/TGF- β activation.

What is the impact of the antibody on thymic Treg cells following in vivo administration? Self-reactive T cells (tTreg precursors) express CD25 and there is an opportunity to evaluate the role of IL2/TGF- β signaling during tTreg differentiation.

In Fig. 6 the authors perform a considerable amount of analysis to show that the pTreg cells resemble native colonic pTreg cells. It is unclear why there are no native pTreg cells in the scRNA-seq data given that ~10-15% of mLN Treg cells are Ror γ t+ pTreg cells. A scRNA-seq analysis that includes these cells, allowing direct comparison of the antibody induced pTreg cells with native pTreg cells from mLN and large intestine, would be more informative and strengthen the conclusion that pTreg cells resemble colonic pTreg cells.

Several of the plots are missing statistical analysis/p values.

Minor comments

The authors state the pTreg induction by the IL-2-TGF- β antibody does not require gut homing or mucosal conditioning but this has not been demonstrated. It is possible that the T cells were primed within gut lymph nodes and migrated to systemic sites.

Extended data Fig. 4b – the authors suggest that tTreg cells arise from minor contamination in the transferred OT-II cells. What proportion of the transferred cells express OT-II va and vb chains at the time of analysis?

Referee #2

(Remarks to the Author)

Building on the suite of designer IL-2R ligands that the Garcia group have advanced over recent years, Sun et al. describe a new set of dual-specific, AND ligands targeting the TGF β and IL-2 receptor complexes with the aim of driving pTreg cell development and function for application to immune-mediated disease. The authors appear to have succeeded, and in a well-written, logical exposition of a very nice data set show that using the bi-specific ligand produced by fusion of a three-domain helminth-derived TGF β mimic (TGM1) to wild-type IL-2, they can drive development of colonic-like pTreg cells in vivo that can prevent pulmonary inflammation in an OVA airway sensitization model. The protein engineering and in vitro characterization of the candidate molecules are a tour-de-force, and the immunology data are generally potent and highly interesting. There is a lot to unwrap here that is beyond the scope of a focused critique. Despite the remarkable findings, however, there are curious omissions that could strengthen the study substantially, particularly regarding the in vivo findings. Nevertheless, this is an important advance and should be of sufficient impact to warrant publication in Nature in revision.

Major Concerns:

1. One of the potential assets but also unexpected features of the TGM1-IL-2 protein is its potent induction of ROR γ t+ pTreg cells in vivo. This is both a curiosity and potentially interesting finding. These are striking data and are well enforced by the scRNA-seq data provided. However, this is an unusual "colonic" phenotype to find in abundance in the secondary lymphoid tissues and it is odd that the authors made no effort to assess this cell population in the gut where these cells dominate—especially in view of the contention that TGM1-IL-2 does not induce much amplification of endogenous pTreg cell populations. These would be straightforward experiments and should be included in the analysis. Moreover, the authors use published scRNA-seq data to draw parallels between this "colonic" pTreg cell phenotype and the TGM1-IL-2-induced cells (and might have included Sarah Teichman's published data here), but why not just extend the scRNA-seq to colonic pTregs isolated from their own mice? This would be the ideal control and comparator for their study.
2. Given the potent pTreg (and eTreg) induction by delivery of TGM1-IL-2, it is surprising that the authors chose to limit study of the treatment efficacy of their protein using a tolerance-inducing pre-treatment protocol in the OT-II airway sensitization model. While the results are compelling, this model does not offer a rigorous test of the therapeutic use of the protein but more of a test of its tolerance-inducing capacity. Why not test the protein in another airway model (house dust mite Ag) or in gut inflammation models to determine if induction of pTregs occurs in the face of active inflammation or even as a differentiation deviation schema (e.g. T cell transfer colitis, DSS colitis, EAE)? Along the same lines, why not test the protein in a bystander repression setting, using the Ag-specific targeting to examine effects on polyclonally-driven disease? Would isolation of the induced pTregs treat disease in adoptive transfers where Ag is provided to support the pTregs? Especially in view of the dominant colonic phenotype of the pTregs induced, seems it would make sense to explore the function of these cells in the gut. Limiting the study to a Th2 sensitization model is tantalizing but ultimately only weakens the story.

Minor Concerns:

1. It was difficult to determine exactly how the in vitro effector T cell polarization studies in Fig. 1 and Ext Data Fig. 2 were performed. This should be made more clear in the Methods and Fig. legends.
2. Inclusion of the TGM1-IL-2(DN) data using the REH variant are interesting but seem a digression from the main thrust of the study and Fig. 5 might be moved to the Extended Data set.
3. Similarly, the text describing the scRNA-seq findings is long and saps a bit of the energy of the main thrust of the paper. It could be toned down and some of the detail moved to Extended Data set.

Version 2:

Reviewer comments:

Referee #1

(Remarks to the Author)

The authors have undertaken considerable work to address the points raised with new experiments and analyses that reinforce the manuscript's conclusions and support earlier results. Overall, the revised study appears robust and the findings are compelling. One interesting interpretation emerging from the data is that the "colonic" pTreg signature may not reflect a colon or gut-specific program per se, but rather a general transcriptional signature of TGF- β /IL2 induced peripheral Treg cell differentiation, independent of the site of priming. The colonic association may simply arise from the fact that pTreg cells are generated almost exclusively within gut-draining lymph nodes under homeostatic conditions.

Referee #2

(Remarks to the Author)

In response to the prior critique, the authors have contributed new experimental data that strengthen their study. Although the design of authors' additional in vivo studies remains focused on testing the dual agonist in a preventive rather than therapeutic mode, the additional data reinforce and extend their prior claims.

We appreciate the valuable feedback provided by both the editor and reviewers. We have revised the manuscript to address each comment point by point. A summary of our major responses and the newly added experiments is provided below:

In response to the common question raised by both reviewers regarding the inclusion of an additional therapeutic model, we incorporated a MOG₃₅₋₅₅-induced EAE mouse model to strengthen our study. Specifically, we:

- Evaluated the efficacy of TGM1–IL-2 in inducing MOG₃₅₋₅₅-specific 2D2 pTregs in MOG₃₅₋₅₅-immunized mice (**Rebuttal Fig. 1**).
- Assessed the therapeutic effect of TGM1–IL-2-induced immune tolerance in protecting against MOG₃₅₋₅₅-induced EAE (**Rebuttal Fig. 2**).

In response to reviewer #1's specific comments, we additionally:

- Assessed the therapeutic effect of TGM1–IL-2-induced immune tolerance in protecting against OVA-induced food allergy (**Rebuttal Fig. 3**).
- Tested the efficacy of TGM1–IL-2 in inducing OT-II pTregs across distinct peripheral lymphoid organs in OVA-immunized mice under FTY720 treatment (**Rebuttal Fig. 5**).

In response to reviewer #2's specific comments, we additionally:

- Analyzed the phenotypes of OT-II pTregs induced by TGM1–IL-2 following their migration into the gut in OVA-immunized mice (**Rebuttal Fig. 7**).
- Compared the expression of key colonic Treg signature molecules between OT-II pTregs and colonic Tregs (**Rebuttal Fig. 8**).
- Evaluated the therapeutic effect of TGM1–IL-2-induced OT-II pTregs in suppressing DSS-induced gut inflammation (**Rebuttal Fig. 9**).

Referees' comments:

Referee #1 (Remarks to the Author):

In this study, Sun et al. designed and evaluated an IL-2-TGF- β agonist. Using a series of *in vitro* and *in vivo* experiments, the authors demonstrate that this antibody effectively induces iTreg cells *in vitro* and pTreg cells *in vivo*. *In vivo* generated pTreg cells expressed Ror γ t and transcriptionally resembled colonic pTreg cells. *In vivo* administration ameliorated disease severity in a model of allergic airway inflammation.

Overall, the experiments are performed to a high standard and provide important proof-of-concept *in vivo* therapeutic studies. However, there are some aspects of the study that should be addressed to strengthen the manuscript.

Response: We thank the reviewer for the constructive comments. We have addressed questions point by point as follows.

#1 While the authors nicely demonstrate the efficacy of the antibody to generate pTreg cells *in vivo* which are associated with protection from allergen-induced airway inflammation, further studies are required to establish the broader therapeutic relevance of this antibody.

Response: To address this question, we included a MOG₃₅₋₅₅-induced EAE model to extend the therapeutic relevance of TGM1-IL-2, complementing our findings from the *in vivo* OVA models. First, we evaluated the ability of TGM1-IL-2 to induce MOG₃₅₋₅₅-specific pTreg cells *in vivo*. We adoptively transferred thy1.1⁺ naïve MOG₃₅₋₅₅-specific 2D2 cells into thy1.1⁻ C57BL/6 recipient mice, followed by intraperitoneal administration of 40 μ g MOG₃₅₋₅₅ peptide together with 50 pmol protein every other day for a total of six doses. Donor 2D2 cells were analyzed in peripheral lymphoid organs on day 11 (**Rebuttal Fig. 1a**).

The data showed that, similar to our observations in the OVA model, TGM1-IL-2(WT) robustly promoted the differentiation of donor MOG₃₅₋₅₅-reactive CD44⁺Ki-67⁺ 2D2 cells into Foxp3⁺ pTregs in mLNs, ILNs, and spleens, with a substantial fraction of Ror γ t⁺Foxp3⁺ pTregs, as indicated by both their frequencies and absolute numbers (**Rebuttal Fig. 1b-e**). In contrast, few Foxp3⁺ pTregs were observed in the PBS, TGM1, or IL-2(WT) groups (**Rebuttal Fig. 1c-e**). Moreover, the 2D2 pTregs induced by TGM1-IL-2(WT) exhibited robust expression of activation and functional markers, including CD25, ICOS, CTLA-4, CD39, CD103, and IL-10 (**Rebuttal Fig. 1f**). These data suggest that TGM1-IL-2 effectively promotes the induction of functional pTregs from MOG₃₅₋₅₅-stimulated 2D2 cells in peripheral lymphoid organs. These data can be found in the revised **Figure 5** and **Extended Data Figure 9**.

Rebuttal figure 1. (a) Experimental scheme showing the adoptive transfer of naïve 2D2 cells and subsequent administration of MOG₃₅₋₅₅ peptide and protein. (b) Gating strategy for identifying MOG₃₅₋₅₅-reactive 2D2 cells. (c) Representative flow cytometry plots showing Foxp3 and Roryt expression in CD44⁺Ki-67⁺ 2D2 cells from the indicated tissues of mice treated with the specified proteins. (d and e) Quantification of the frequencies and absolute numbers of Foxp3⁺ and Roryt⁺Foxp3⁺ 2D2 cells. (f) Representative flow plots and quantification of Foxp3⁺ 2D2 cells expressing activation and functional markers CD25, ICOS, CTLA-4, CD39, CD103, and IL-10.

To further evaluate the therapeutic effects of TGM1–IL-2–induced immune tolerance in protecting against EAE, we first administered MOG₃₅₋₅₅ together with TGM1–IL-2 (WT) every other day for a total of five doses. One week later, the mice were re-challenged with MOG₃₅₋₅₅ emulsified in CFA (**Rebuttal Fig. 2a**). Compared with the PBS group, mice treated with TGM1–IL-2 (WT) exhibited markedly reduced EAE clinical scores (**Rebuttal Fig. 2b**). Notably, 9 of 11 mice in the TGM1–IL-2 (WT) group remained EAE-free, whereas nearly all PBS-treated mice developed the disease (**Rebuttal Fig. 2c**). Flow cytometry analysis further revealed a substantial reduction in

infiltrating CD45.2⁺ immune cells, including CD11b⁺CD3⁻ myeloid cells, CD11b⁻CD3⁺ T cells, and CD4⁺ T cells in the spinal cord (**Rebuttal Fig. 2d and 2e**). Although the frequencies of IFN- γ - and IL-17a-producing CD4⁺ T cells were not altered by TGM1-IL-2 (WT) treatment, their absolute numbers were significantly decreased in the spinal cord (**Rebuttal Fig. 2f and 2g**). Moreover, TGM1-IL-2 (WT) reduced both the percentage and total number of GM-CSF⁺ CD4⁺ T cells, indicating a decrease in pathogenic Th17 cells (**Rebuttal Fig. 2h**). Together, these data demonstrate that TGM1-IL-2 (WT) effectively induces immune tolerance and provides robust protection against EAE development in mice. These data can be found in the revised **Figure 5** and **Extended Data Figure 9**.

Rebuttal Figure 2. (a) Experimental scheme illustrating the administration of MOG₃₅₋₅₅ peptide and TGM1-IL-2 (WT), followed by MOG₃₅₋₅₅ emulsion in CFA one week later. (b) Mean EAE scores of mice in the indicated groups. (c) Individual EAE scores for mice in the indicated groups. (d) Quantification of infiltrating CD45.2⁺, CD45.2⁺CD11b⁺CD3⁻, and CD45.2⁺CD11b⁻CD3⁺ cells in the spinal cord. (e) Quantification of infiltrating CD4⁺ T cells in the spinal cord. (f) Representative flow cytometry plots showing the frequency of IFN- γ - and IL-17a-producing CD4⁺ T cells in the spinal cord. (g) Quantification of IFN- γ - and IL-17a-producing CD4⁺ T cells in the spinal cord. (h) Representative flow cytometry plots and quantification of GM-CSF⁺ CD4⁺ T cells in the spinal cord.

#2 The authors show that the pTreg cells resemble colonic pTreg cells therefore one exciting

avenue that could extend the scope of the present work would be to determine whether the bispecific antibody can promote colonic pTreg differentiation and restore homeostasis in settings of intestinal inflammation for example following pre-sensitization to oral antigens within the gut (mimicking food allergy) or following onset of colitis.

Response: We agree that exploring the potential of TGM1–IL-2 to promote pTreg differentiation in the gut could be promising. However, in this study, we administered antigens via intraperitoneal injection, which allows systemic distribution and elicits a systemic immune response rather than restricting the antigen to the gut. Under these conditions, analyzing pTreg differentiation in the gut is challenging due to limited antigen presence and consequently low infiltration of antigen-specific CD4⁺ T cells. Nevertheless, given the strong colonic pTreg features of TGM1–IL-2–induced pTregs, we hypothesize that these pTregs can efficiently migrate to the gut and suppress antigen-induced food allergy upon subsequent oral antigen administration.

To test this, we further evaluated whether TGM1–IL-2–induced immune tolerance could ameliorate food allergy using an OVA-induced food allergy model¹. Naïve OT-II cells were adoptively transferred into recipient mice, followed by intraperitoneal injection of OVA protein and TGM1–IL-2(WT) every other day for a total of five doses. The mice were then sensitized with OVA protein and cholera toxin (CT) three times, and subsequently challenged by intraperitoneal injection of OVA protein one week later (**Rebuttal Fig. 3a**).

The data showed that, compared with PBS-treated mice, those pretreated with TGM1–IL-2 (WT) exhibited markedly smaller drops in rectal temperature, with half of the mice showing no temperature decline at all (**Rebuttal Fig. 3b and 3c**). Furthermore, TGM1–IL-2(WT) treatment markedly decreased serum levels of OVA-specific IgE and IgG1 compared to PBS, indicating substantial attenuation of OVA-induced allergic responses (**Rebuttal Fig. 3d**). In addition, flow cytometry analysis showed that a large proportion of OT-II cells in the mesenteric lymph nodes (mLN), Peyer's patches (PP), and small intestine (SI) remained Foxp3⁺ pTregs, the majority of which were Rorγt⁺, both in frequency and absolute number, whereas few OT-II pTregs were detected in the PBS group (**Rebuttal Fig. 3e-g**). Collectively, these results suggest that TGM1–IL-2–induced pTregs are capable of efficiently migrating to the gut and protecting against allergic responses to oral antigens. These data can be found in the revised **Figure 5l-q**.

Rebuttal figure 3. (a) Experimental scheme of naïve OT-II cell transfer and administration of OVA and TGM1-IL-2, followed by OVA and cholera toxin (CT) sensitization and challenge. (b) Mean rectal temperature of mice in the indicated groups following OVA protein i.p. challenge. (c) Individual rectal temperature of mice in the indicated groups. (d) Serum levels of OVA-specific IgE and IgG1 in mice from the indicated groups. (e and f) Representative flow cytometry plots and quantification of Gata3, Ror γ t, and Foxp3 expression in OT-II cells from indicated tissues. (g) Quantification of Foxp3⁺ OT-II cell numbers in the indicated tissues.

#3 Does the bispecific antibody bypass the need for other tolerogenic/pTreg-inducing signals or is IL-2 and TGF- β signaling sufficient? Given that pTreg inducing Ror γ t⁺ APCs have been identified, the authors could determine whether their presence is required for antibody induced pTreg

differentiation.

Response: Our phenotypic, functional, and transcriptional analyses provide strong evidence that IL-2 and TGF- β signaling delivered by TGM1–IL-2 alone are sufficient to induce effective pTreg differentiation *in vivo*, without additional signals. In this study, antigens were administered via intraperitoneal injection, resulting in systemic distribution rather than enrichment in tissues naturally enriched with tolerogenic signals, such as the gut following oral administration. Despite this, TGM1–IL-2 robustly induced pTreg differentiation across multiple lymphoid organs, an effect not observed in PBS-treated controls. Collectively, these data strongly suggest that TGM1–IL-2 can likely bypass the need for additional tolerogenic or pTreg-inducing signals and are sufficient to drive pTreg induction *in vivo*.

In the design of the TGM1–IL-2 molecule, the high-affinity IL-2 moiety functions as a targeting arm that specifically engages T cells with high IL-2 receptor expression, while exerting minimal effects on APCs, which express low or negligible levels of IL-2 receptors. Furthermore, data from subsequent FTY720 treatment indicate that OT-II pTreg induction can occur independently and to a similar extent in the spleen and ILNs, distant from the mLN and gut where Ror γ ⁺ APCs are enriched and active. These findings suggest that pTreg induction by TGM1–IL-2 *in vivo* may not require Ror γ ⁺ APCs, but this will require further exploration in the next phase of our studies.

Hence, we have added the following discussion to **lines 759–764** of the revised manuscript:

“Unlike oral gavage antigen administration, which induces robust pTreg generation in the gut in a Ror γ ⁺ APC–dependent manner, intraperitoneal administration of antigens together with TGM1–IL-2(WT) efficiently promotes pTreg induction systemically, including in lymphoid organs distant from the gut. These findings suggest that TGM1–IL-2 likely provides sufficient signals for pTreg induction *in vivo* without relying on Ror γ ⁺ APCs, but this question warrants further investigation.”

#4 Related to this, the authors state that the role of TGF- β and IL-2 in promoting pTreg differentiation *in vivo* have not been fully defined but there are a number of recent studies that have established a role for avb8⁺ Ror γ ⁺ APCs in microbiota and food-specific pTreg differentiation (Kedmi et al. Nature 2022 (PMID: 36071167), Rudnitsky et al. Nature 2025 (PMID: 40425043), Sun et al. JEM 2025 (PMID: 40298935), Akagbosu et al. Nature 2022 (PMID: 36070798), Cabric et al. 2025 (PMID: 40373113)). These studies are referenced but should be discussed in relation to avb8/TGF- β activation.

Response: We agree that we cited these publications, but have now incorporated a more detailed description of these studies into **lines 66–70** of the Introduction section in the revised manuscript as follows:

“Recent studies have also shown that disruption of *Itgav* or *Itgb8* expression in Ror γ ⁺ antigen-presenting cells (APCs) impairs microbiota- and food antigen–induced pTreg differentiation in the gut²⁻⁶, highlighting a critical role for active TGF- β signaling in driving pTreg induction *in vivo*. However, the synergistic harnessing of TGF- β and IL-2 signaling for pTreg generation *in vivo* remains largely unexplored.”

#5 What is the impact of the antibody on thymic Treg cells following *in vivo* administration? Self-reactive T cells (tTreg precursors) express CD25 and there is an opportunity to evaluate the role of IL2/TGF- β signaling during tTreg differentiation.

Response: We agree that TGM1–IL-2 would be well-suited to investigate the roles of IL-2 and TGF- β signaling in thymic Treg (tTreg) development. However, in the thymus, in addition to tTreg precursors, subsets of double-negative pre-T cells and mature tTregs also express CD25^{7,8}, which would make it challenging to draw clear conclusions from a small-scale experiment. As our current study primarily focuses on *in vivo* pTreg induction, we feel that such studies are out of scope of this manuscript.

#6 In Fig. 6 the authors perform a considerable amount of analysis to show that the pTreg cells resemble native colonic pTreg cells. It is unclear why there are no native pTreg cells in the scRNA-seq data given that ~10-15% of mLN Treg cells are Ror γ t⁺ pTreg cells. A scRNA-seq analysis that includes these cells, allowing direct comparison of the antibody induced pTreg cells with native pTreg cells from mLN and large intestine, would be more informative and strengthen the conclusion that pTreg cells resemble colonic pTreg cells.

Response: We appreciate the opportunity to clarify this point. As shown by flow cytometry plots, approximately 8.47% of endogenous Foxp3⁺ Tregs in the mLNs were Ror γ t⁺ (**Rebuttal Fig. 4a**). However, only about 1.32% of these cells showed detectable *Rorc* mRNA in the scRNA-seq data (**Rebuttal Fig. 4b**), likely due to the low abundance of *Rorc* transcripts, which is a common technical limitation in RNA sequencing and difficult to overcome.

Additionally, OT-II pTregs are strongly activated by OVA antigen stimulation, as reflected by their elevated expression of TCR signaling–related genes, whereas these genes are expressed at low levels in endogenous Tregs (**Rebuttal Fig. 4c, and revised Figure 7d**). This difference in activation status may make OT-II pTregs transcriptionally distinct from endogenous pTregs, which is likely the main reason why we were unable to identify a comparable pTreg population using the current dimensionality reduction cutoff.

Importantly, we believe that analyzing the enrichment of colonic Treg–associated genes from high-quality published RNA-seq datasets in the OT-II pTreg population provides strong evidence that OT-II pTregs exhibit colonic Treg signatures. To further support this, we calculated the percentage of OT-II pTregs with a colonic Ror γ t⁺ Treg signature score ≥ 0.25 . The data showed that 8.8% of endogenous Tregs from the mLN and 74.7% of OT-II cells in the TGM1–IL-2(WT) group reached this threshold, which aligns well with the percentages of Ror γ t⁺ cells observed at the protein level, reflecting the high accuracy of this colonic Treg gene signature set (**Rebuttal Fig. 4d**). For greater clarity, we have included the colonic Treg signature gene lists in the **Extended Data Table 1** of the revised manuscript (**Rebuttal Fig. 4e**).

Rebuttal figure 4. (a) Representative flow plots showing the percentage of $Roryt^+$ cells in endogenous and OT-II Tregs from indicated tissues. (b) Quantification of the percentage of cells with detectable *Rorc* expression in the indicated samples. (c) Bubble plots illustrating the expression of TCR signaling-related genes in the indicated cell populations. (d) Quantification of the percentage of cells with a colonic $Roryt^+$ Treg signature score ≥ 0.25 . (e) Gene lists for colonic $Roryt^+$ Tregs used in the scRNA-seq analyses.

#7 Several of the plots are missing statistical analysis/p values.

Response: We have added the missing statistical analyses and p-values throughout all figures in the revised manuscript.

#8 Minor comments

The authors state the pTreg induction by the IL-2-TGF- β antibody does not require gut homing or mucosal conditioning, but this has not been demonstrated. It is possible that the T cells were primed within gut lymph nodes and migrated to systemic sites.

Response: To address this point, we treated OVA-immunized mice with FTY720 daily from day 0 for 11 consecutive days during *in vivo* OT-II pTreg differentiation (**Rebuttal Fig. 5a**). FTY720 is a sphingosine-1-phosphate receptor (S1PR) modulator that blocks T-cell egress from lymph nodes, thereby allowing us to examine OT-II differentiation independently within the corresponding lymphoid organs where the cells are retained.

OVA-immunized mice were analyzed on days 5 and 11 (**Rebuttal Fig. 5a**). Blood analysis confirmed effective inhibition of T-cell egress by showing that few $CD4^+$ T cells were detectable in the blood following FTY720 treatment (**Rebuttal Fig. 5b**). In the FTY720-treated mice, we

observed substantial Foxp3⁺ OT-II pTregs on day 11, with more than 60% present in the mLN, ILN and spleen, which was similar to the percentages observed in PBS-treated controls (**Rebuttal Fig. 5c and 5d**). Most of these OT-II pTregs co-expressed CD25 and Ror γ t (**Rebuttal Fig. 5c and 5d**). In contrast, on day 5, only a small fraction of Foxp3⁺ OT-II pTregs with low Ror γ t expression were detected in the mLN and ILN, whereas higher frequencies of Foxp3⁺ cells with low Ror γ t expression were observed in the spleen, indicating that a large fraction of Ror γ t⁺Foxp3⁺ OT-II pTregs differentiated between days 5 and 11 *in vivo* (**Rebuttal Fig. 5c and 5d**).

By comparing OT-II cell numbers across lymphoid organs, we found that FTY720 treatment reduced both total OT-II cells and OT-II pTregs in the spleen on day 11, while having minimal effects on their abundance in the mLN and ILN (**Rebuttal Fig. 5e and 5f**). This pattern is consistent with the known function of FTY720 in restricting T-cell egress from lymph nodes but not from the spleen.

Together, these data indicate that OT-II cell differentiation into Foxp3⁺ pTregs can occur independently within the ILN and spleen without requiring migration from the mLN. The data also show that substantial Ror γ t⁺Foxp3⁺ differentiation *in vivo* occurs between days 5 and 11. These data can be found in the revised **Figure 3e, 3f**, and **Extended Data Figure 5a-f**.

Rebuttal figure 5. (a) Experimental design of OVA immunization with TGM1-IL-2(WT) and FTY720 treatment. **(b)** Quantification of CD4⁺ T cell percentages among total live cells in blood. **(c)** Representative FACS plots and **(d)** statistical analysis showing the expression of Ror γ t, CD25, and Foxp3 in OT-II cells from the indicated secondary lymphoid organs at the indicated time

points. (e) Quantification of total OT-II cell numbers in the indicated secondary lymphoid organs at the indicated time points. (f) Quantification of $Foxp3^+$, $Roryt^+Foxp3^+$ and $CD25^+Foxp3^+$ OT-II cell numbers in the indicated secondary lymphoid organs at the indicated time points.

#9 Extended data Fig. 4b – the authors suggest that tTreg cells arise from minor contamination in the transferred OT-II cells. What proportion of the transferred cells express OT-II va and vb chains at the time of analysis?

Response: Thank you for the question. Flow cytometry analysis showed that all donor $Thy1.1^+$ cells in recipient mice expressed the TCR $V\alpha 2$ and TCR $V\beta 5.1/5.2$ chains, confirming that they were OT-II cells (**Rebuttal Fig. 6a**). The “contamination” mentioned in the original manuscript refers to a very small population of tTreg OT-II cells. As shown in the post-sorting flow cytometry plots, although the sorted OT-II cells were almost entirely $CD44^-CD25^-Foxp3^-$, a minor fraction of $CD25^+Foxp3^+$ tTreg OT-II cells (less than 1%) remained and could be markedly expanded by IL-2(N88D) treatment *in vivo* (**Rebuttal Fig. 6b**). To avoid confusion, we have revised the corresponding sentence in **lines 294–295** of the revised manuscript as follows:

“Although IL-2(N88D) treatment generated a substantial $Foxp3^+$ population, these cells were predominantly $Helios^+ Roryt^-$, indicating they were likely expanded **OT-II** tTregs arising from minor contamination in the transferred **naïve** OT-II cells.”

Rebuttal figure 6. (a) Representative FACS plots showing the percentage of $TCR V\alpha 2^+$ and $TCR V\beta 5.1/5.2^+$ cells among donor $Thy1.1^+$ cells in the mLN of OVA-immunized recipient mice. **(b)** FACS plots showing $CD44$, $CD25$, and $Foxp3$ expression in sorted naïve OT-II cells prior to transfer.

Referee #2 (Remarks to the Author):

Building on the suite of designer IL-2R ligands that the Garcia group have advanced over recent years, Sun et al. describe a new set of dual-specific, AND ligands targeting the TGF β and IL-2 receptor complexes with the aim of driving pTreg cell development and function for application to immune-mediated disease. The authors appear to have succeeded, and in a well-written, logical exposition of a very nice data set show that using the bi-specific ligand produced by fusion of a three-domain helminth-derived TGF β mimic (TGM1) to wild-type IL-2, they can drive development of colonic-like pTreg cells in vivo that can prevent pulmonary inflammation in an OVA airway sensitization model. The protein engineering and in vitro characterization of the candidate molecules are a tour-de-force, and the immunology data are generally potent and highly interesting. There is a lot to unwrap here that is beyond the scope of a focused critique. Despite the remarkable findings, however, there are curious omissions that could strengthen the study substantially, particularly regarding the in vivo findings. Nevertheless, this is an important advance and should be of sufficient impact to warrant publication in Nature in revision.

Response: We thank the reviewer for the constructive comments. We have addressed all suggestions and questions point by point as follows.

Major Concerns:

#1 1. One of the potential assets but also unexpected features of the TGM1-IL-2 protein is its potent induction of ROR γ t+ pTreg cells in vivo. This is both a curiosity and potentially interesting finding. These are striking data and are well enforced by the scRNA-seq data provided. However, this is an unusual "colonic" phenotype to find in abundance in the secondary lymphoid tissues and it is odd that the authors made no effort to assess this cell population in the gut where these cells dominate—especially in view of the contention that TGM1-IL-2 does not induce much amplification of endogenous pTreg cell populations. These would be straightforward experiments and should be included in the analysis.

Response: In this study, we observed high expression of Ror γ t and other colonic Treg-associated genes, including c-MAF, CXCR6, and CCR6, in TGM1-IL-2-induced pTregs within peripheral lymphoid organs. Combined with the scRNA-seq data, these results indicate that TGF- β signaling delivered by TGM1-IL-2 is the primary driver of these gene expression in pTregs, consistent with its reported role in gut pTreg induction observed in studies with *Itgav* and *Itgbv* gene deletion²⁻⁶.

Notably, unlike microbiota- or food antigen-induced pTreg induction in the gut, where the antigen is enriched locally, we administered OVA or MOG antigens via intraperitoneal injection. Under this condition, the antigens are distributed systemically rather than being concentrated in the gut, allowing pTreg induction across multiple lymphoid organs rather than specifically in the mLN and gut. Consequently, we detected substantial OT-II or 2D2 pTregs in peripheral lymphoid organs, but few OT-II or 2D2 cells in the gut due to the low antigen presence there. This is why analyses of these cells in the gut were not included.

Additionally, as reported in studies of OT-II pTreg induction in response to oral OVA gavage, OT-II cells are primed and differentiate into Ror γ t+ pTregs in the mLN before migrating to the gut, where the antigens are enriched^{1,6}. Based on this, we reasoned that, as shown in our study, the

Roryt⁺ pTreg phenotype may not be restricted to the gut, but rather depends on the distribution of antigen and tolerance signals, such as TGF- β .

To further examine the phenotypes of the OT-II pTregs after their migration into intestinal tissues, we administered OVA in the drinking water for 5 days to promote migration of OT-II pTregs into the gut following their differentiation in response to intraperitoneal injection of OVA protein and TGM1-IL-2 (**Rebuttal Fig. 7a**). The results showed substantial Roryt⁺ OT-II pTregs in Peyer's patches (PP), small intestine (SI), and colon, resembling canonical colonic pTreg features, as shown by both their frequencies and numbers (**Rebuttal Fig. 7b-d**). However, these gut pTregs expressed low levels of Gata3 (**Rebuttal Fig. 7c-d**). These data can be found in the revised **Extended Data Figure 5g-i**.

Rebuttal figure 7. (a) Experimental design for OT-II pTreg induction followed by OVA administration in drinking water. **(b and c)** Representative flow cytometry plots showing Foxp3, Roryt, and Gata3 expression in OT-II cells from the indicated tissues. **(d and e)** Quantification of Foxp3⁺, Roryt⁺Foxp3⁺, and Gata3⁺Roryt⁺ OT-II cell frequencies and numbers in the indicated tissues.

#2 Moreover, the authors use published scRNA-seq data to draw parallels between this "colonic" pTreg cell phenotype and the TGM1-IL-2-induced cells (and might have included Sarah Teichman's published data here), but why not just extend the scRNA-seq to colonic pTregs isolated from their own mice? This would be the ideal control and comparator for their study.

Response: We appreciate the opportunity to clarify this issue. As noted, we included published scRNA-seq datasets from Sarah Teichman (PMID: 30737144) and Christophe Benoist (PMID: 33462454) to compare the enrichment of colonic Treg signatures in our OT-II pTreg transcriptomes. These comparisons provide strong evidence that OT-II pTregs induced by TGM1-IL-2 exhibit features characteristic of colonic pTregs. To enhance transparency, we have now included the gene list for Ror γ ⁺ colonic pTregs from the above study in **Extended Data Table 1** of the revised manuscript.

We fully agree that a direct comparison between the transcriptomes of OT-II pTregs and endogenous colonic Tregs from the same mice would be highly informative. However, our current scRNA-seq dataset did not include endogenous colonic Tregs, and performing an additional sequencing run at a different time could introduce substantial batch effects, potentially confounding the analysis. Nevertheless, in an attempt to address this point experimentally, we performed flow cytometric analyses comparing OT-II pTregs induced by TGM1-IL-2 with colonic Tregs from the same mice. The data showed that, similar to Foxp3⁺ Tregs in the colon, OT-II pTregs from mLN and spleen induced by TGM1-IL-2 (WT) expressed high levels of Ror γ t, c-MAF, CXCR6, and CCR6, with expression levels even higher than those observed in colonic Tregs (**Rebuttal Fig. 8**). These data can be found in the revised **Figure 7j**.

Rebuttal figure 8. Representative flow cytometry plots and quantification of Ror γ t, c-MAF, CXCR6, and CCR6 expression in the indicated cells.

#3 2. Given the potent pTreg (and eTreg) induction by delivery of TGM1-IL-2, it is surprising that the authors chose to limit study of the treatment efficacy of their protein using a tolerance-inducing pre-treatment protocol in the OT-II airway sensitization model. While the results are compelling, this model does not offer a rigorous test of the therapeutic use of the protein but more of a test of its tolerance-inducing capacity. Why not test the protein in another airway model (house dust mite Ag) or in gut inflammation models to determine if induction of pTregs occurs in the face of active

inflammation or even as a differentiation deviation schema (e.g. T cell transfer colitis, DSS colitis, EAE)?

Response: To address this question, we included two additional therapeutic models alongside the airway allergic inflammation model: an OVA-induced food allergy model (**Rebuttal Fig. 3**) and a MOG₃₅₋₅₅-induced EAE model (**Rebuttal Fig. 1 and 2**), as described in our response to Reviewer #1.

During the active disease phase, heightened inflammatory signals in the environment, such as those induced by adjuvants, strongly impair pTreg induction. In addition, IL-2 signaling provided by TGM1-IL-2 promotes antigen-specific T cell activation and expansion, which may further accelerate disease. Therefore, due to the complexity of signals and T cell status during the active disease phase and given the primary focus of this study on immune tolerance, we did not include efficacy data after disease onset. We plan to explore its therapeutic potential during active disease in future studies.

#4 Along the same lines, why not test the protein in a bystander repression setting, using the Ag-specific targeting to examine effects on polyclonally-driven disease? Would isolation of the induced pTregs treat disease in adoptive transfers where Ag is provided to support the pTregs? Especially in view of the dominant colonic phenotype of the pTregs induced, seems it would make sense to explore the function of these cells in the gut. Limiting the study to a Th2 sensitization model is tantalizing but ultimately only weakens the story.

Response: To address this question, we assessed the bystander suppressive effect of OT-II pTregs in a DSS-induced, polyclonal antigen-driven gut inflammation model. Specifically, following *in vivo* OT-II pTreg induction, mice were given DSS in drinking water for six days (**Rebuttal Fig. 9a**). To promote migration of OT-II pTregs into the gut and sustain their activation, 2.5 mg/mL OVA protein was provided in drinking water, and 5 mg of OVA protein was administered by oral gavage every other day (**Rebuttal Fig. 9a**).

The data showed that mice pre-treated with TGM1-IL-2 exhibited delayed and reduced weight loss compared to PBS-treated controls (**Rebuttal Fig. 9b and 9c**). TGM1-IL-2 treatment also resulted in longer colons and lower histological inflammation scores (**Rebuttal Fig. 9d and 9e**). Furthermore, TGM1-IL-2 reduced the proportion of IFN- γ -producing CD4⁺ T cells in the colon, while the proportion of IL-17a-producing CD4⁺ T cells remained unchanged (**Rebuttal Fig. 9f**). Together, these results indicate that TGM1-IL-2 treatment significantly ameliorates DSS-induced gut inflammation.

We further analyzed the properties of OT-II pTregs in the gut. The proportion of Foxp3⁺ OT-II cells among total CD4⁺ T cells was higher in the small intestine (SI) and colon compared to mLNs and Peyer's patches (PP), indicating effective migration of OT-II pTregs into the gut (**Rebuttal Fig. 9g**). In contrast, OT-II cells in the PBS group were almost entirely Foxp3⁻ conventional T cells (Tconv), and their frequency among total CD4⁺ T cells in the SI and colon was even lower than that in the TGM1-IL-2(WT) group (**Rebuttal Fig. 9g**). This suggests that the reduced gut inflammation was not due to suppression of Tconv differentiation by TGM1-IL-2(WT), but rather mediated by the suppressive function of OT-II pTregs. Moreover, OT-II pTregs induced by

TGM1-IL-2(WT) were predominantly $Roryt^+Gata3^-$ across mLNs, PP, SI, and colon (**Rebuttal Fig. 9h**). Together, these data indicate that TGM1-IL-2(WT)-induced OT-II pTregs efficiently migrate to the gut and suppress inflammation via bystander effects. These data can be found in the revised **Figure 7l-o**, and **Extended Data Figure 13**.

Rebuttal figure 9. (a) Experimental schematic showing in vivo induction of OT-II pTregs followed by DSS-induced colitis. (b) Weight change of mice in the indicated groups. (c) Individual weight changes of mice in the indicated groups. (d) Representative colon images and quantification of colon length in the indicated groups. (e) Representative H&E-stained colon sections. (f) Representative flow cytometry plots and quantification of IFN- γ^- and IL-17a-producing endogenous CD4⁺ T cells in the colon. (g) Representative FACS plots and quantification of Foxp3⁻ and Foxp3⁺ Thy1.1⁺ OT-II cells among total CD4⁺ T cells in the indicated tissues. (h) Representative FACS plots and quantification of Roryt⁺ and Gata3⁺ Foxp3⁺ OT-II cells in the indicated tissues.

Minor Concerns:

#5 1. It was difficult to determine exactly how the in vitro effector T cell polarization studies in Fig. 1 and Ext Data Fig. 2 were performed. This should be made more clear in the Methods and Fig. legends.

Response: As requested, we have added a detailed description of the *in vitro* experimental procedures to **lines 885–893** of the Methods section in the revised manuscript, as follows:

“For the *in vitro* mouse CD4⁺ T cell differentiation assay, 0.15×10^6 purified naïve CD4⁺ T cells were suspended in 200 μ L of complete medium (RPMI 1640 with GlutaMAX™ Supplement [Gibco, 61870036] supplemented with 10% fetal bovine serum [Gibco, A5256701], 10 mM HEPES [Gibco, 15630080], 1% sodium pyruvate [Gibco, 11360070], 1% penicillin–streptomycin [Gibco, 15140122], and 0.1% 2-mercaptoethanol [Gibco, 21985023]). Cells (200 μ L per well) were plated in flat-bottom 96-well plates (Thermo Scientific, 167008) pre-coated overnight with 5 μ g/mL InVivoMAb anti-mouse CD3 (BE0002, Bio X Cell) and 5 μ g/mL InVivoMAb anti-mouse CD28 (BE0015, Bio X Cell). Recombinant proteins were added at the indicated concentrations on day 0, and cells were cultured at 37°C for 4 days before analysis.”

#6 2. Inclusion of the TGM1-IL-2(DN) data using the REH variant are interesting but seem a digression from the main thrust of the study and Fig. 5 might be moved to the Extended Data set.

Response: We make one clarification to the reviewer, we used the RETR version of IL-2 that lacks common gamma chain binding and therefore binds to IL-2Rb but does not activate signaling⁹. Our logic for inclusion of this figure is as follows: We feel this engineered molecule uniquely shows the requirement, using a pharmacological approach, for co-signaling by pSMAD and pSTAT5, and therefore represents a valuable proof of concept experiment for the AND-gated properties of the molecule. While many previous studies have emphasized the importance of TGF- β signaling in pTreg induction *in vivo*, particularly in the gut, the role of IL-2 signaling has been less explored. In this study, we show that IL-2 is not only essential for CD4⁺ T cell activation and expansion during pTreg induction *in vivo*, but also critical for promoting Ror γ ⁺ pTreg identity and function. These findings provide new insight into the regulation of Ror γ ⁺ pTreg induction *in vivo*. Therefore, we believe it is important to retain the TGM1-IL-2(DN) data in the main figure.

#7 3. Similarly, the text describing the scRNA-seq findings is long and saps a bit of the energy of the main thrust of the paper. It could be toned down and some of the detail moved to Extended Data set.

Response: Thank you for your valuable suggestion. Accordingly, we have moved the scRNA-seq data on activated Treg signature enrichment and transcription factor activity analysis to the revised **Extended Data Figures 12 and 14** and have also toned down the description of less critical parts in the revised manuscript.

Reference

- 1 Fu, L. *et al.* PRDM16-dependent antigen-presenting cells induce tolerance to gut antigens. *Nature* **642**, 756-765 (2025). <https://doi.org:10.1038/s41586-025-08982-4>
- 2 Kedmi, R. *et al.* A RORgammat(+) cell instructs gut microbiota-specific T(reg) cell differentiation. *Nature* **610**, 737-743 (2022). <https://doi.org:10.1038/s41586-022-05089-y>
- 3 Rudnitsky, A. *et al.* A coordinated cellular network regulates tolerance to food. *Nature* **644**, 231-240 (2025). <https://doi.org:10.1038/s41586-025-09173-x>
- 4 Sun, I. H. *et al.* RORgammat eTACs mediate oral tolerance and Treg induction. *J Exp Med* **222** (2025). <https://doi.org:10.1084/jem.20250573>
- 5 Akagbosu, B. *et al.* Novel antigen-presenting cell imparts T(reg)-dependent tolerance to gut microbiota. *Nature* **610**, 752-760 (2022). <https://doi.org:10.1038/s41586-022-05309-5>
- 6 Cabric, V. *et al.* A wave of Thetis cells imparts tolerance to food antigens early in life. *Science* **389**, 268-274 (2025). <https://doi.org:doi:10.1126/science.adp0535>
- 7 Hsieh, C. S., Lee, H. M. & Lio, C. W. Selection of regulatory T cells in the thymus. *Nat Rev Immunol* **12**, 157-167 (2012). <https://doi.org:10.1038/nri3155>
- 8 Carpenter, A. C. & Bosselut, R. Decision checkpoints in the thymus. *Nat Immunol* **11**, 666-673 (2010). <https://doi.org:10.1038/ni.1887>
- 9 Mitra, S. *et al.* Interleukin-2 activity can be fine tuned with engineered receptor signaling clamps. *Immunity* **42**, 826-838 (2015). <https://doi.org:10.1016/j.immuni.2015.04.018>